# Rapid regulation of vesicle priming explains synaptic facilitation despite heterogeneous vesicle:Ca$^{2+}$ channel distances

Janus RL Kobbersmed[1,2], Andreas T Grasskamp[3], Meida Jusyte[3,4], Mathias A Böhme[3], Susanne Ditlevsen[1], Jakob Balslev Sørensen[2]*, Alexander M Walter[3,4]*

[1]Department of Mathematical Sciences, University of Copenhagen, København, Denmark; [2]Department of Neuroscience, University of Copenhagen, København, Denmark; [3]Molecular and Theoretical Neuroscience, Leibniz-Forschungsinstitut für Molekulare Pharmakologie, FMP im CharitéCrossOver, Berlin, Germany; [4]Einstein Center for Neuroscience, Berlin, Germany

*For correspondence:
jakobbs@sund.ku.dk (JBS);
awalter@fmp-berlin.de (AMW)

**Competing interests:** The authors declare that no competing interests exist.

**Abstract** Chemical synaptic transmission relies on the Ca$^{2+}$-induced fusion of transmitter-laden vesicles whose coupling distance to Ca$^{2+}$ channels determines synaptic release probability and short-term plasticity, the facilitation or depression of repetitive responses. Here, using electron- and super-resolution microscopy at the *Drosophila* neuromuscular junction we quantitatively map vesicle:Ca$^{2+}$ channel coupling distances. These are very heterogeneous, resulting in a broad spectrum of vesicular release probabilities within synapses. Stochastic simulations of transmitter release from vesicles placed according to this distribution revealed strong constraints on short-term plasticity; particularly facilitation was difficult to achieve. We show that postulated facilitation mechanisms operating via activity-dependent changes of vesicular release probability (e.g. by a facilitation fusion sensor) generate too little facilitation and too much variance. In contrast, Ca$^{2+}$-dependent mechanisms rapidly increasing the number of releasable vesicles reliably reproduce short-term plasticity and variance of synaptic responses. We propose activity-dependent inhibition of vesicle un-priming or release site activation as novel facilitation mechanisms.

## Introduction

At chemical synapses, neurotransmitters (NTs) are released from presynaptic neurons and subsequently activate postsynaptic receptors to transfer information. At the presynapse, incoming action potentials (APs) trigger the opening of voltage gated Ca$^{2+}$ channels, leading to Ca$^{2+}$ influx. This local Ca$^{2+}$ signal induces the rapid fusion of NT-containing synaptic vesicles (SVs) at active zones (AZs) (*Südhof, 2012*). In preparation for fusion, SVs localize (dock) to the AZ plasma membrane and undergo functional maturation (priming) into a readily releasable pool (RRP) (*Kaeser and Regehr, 2017*; *Verhage and Sørensen, 2008*). These reactions are mediated by an evolutionarily highly conserved machinery. The SV protein VAMP2/Synaptobrevin and the plasma membrane proteins Syntaxin-1 and SNAP25 are essential for docking and priming and the assembly of these proteins into the ternary SNARE complex provides the energy for SV fusion (*Jahn and Fasshauer, 2012*). The SNARE interacting proteins (M)Unc18s and (M)Unc13s (where 'M' indicates mammalian) are also essential for SV docking, priming and NT release (*Rizo and Südhof, 2012*; *Südhof and Rothman, 2009*), while Ca$^{2+}$ triggering of SV fusion depends on vesicular Ca$^{2+}$ sensors of the Synaptotagmin family (*Littleton and Bellen, 1995*; *Südhof, 2013*; *Walter et al., 2011*; *Yoshihara et al., 2003*).

**eLife digest** Cells in the nervous system of all animals communicate by releasing and sensing chemicals at contact points named synapses. The 'talking' (or pre-synaptic) cell stores the chemicals close to the synapse, in small spheres called vesicles. When the cell is activated, calcium ions flow in and interact with the release-ready vesicles, which then spill the chemicals into the synapse. In turn, the 'listening' (or post-synaptic) cell can detect the chemicals and react accordingly.

When the pre-synaptic cell is activated many times in a short period, it can release a greater quantity of chemicals, allowing a bigger reaction in the post-synaptic cell. This phenomenon is known as facilitation, but it is still unclear how exactly it can take place. This is especially the case when many of the vesicles are not ready to respond, for example when they are too far from where calcium flows into the cell. Computer simulations have been created to model facilitation but they have assumed that all vesicles are placed at the same distance to the calcium entry point: Kobbersmed et al. now provide evidence that this assumption is incorrect.

Two high-resolution imaging techniques were used to measure the actual distances between the vesicles and the calcium source in the pre-synaptic cells of fruit flies: this showed that these distances are quite variable – some vesicles sit much closer to the source than others.

This information was then used to create a new computer model to simulate facilitation. The results from this computing work led Kobbersmed et al. to suggest that facilitation may take place because a calcium-based mechanism in the cell increases the number of vesicles ready to release their chemicals.

This new model may help researchers to better understand how the cells in the nervous system work. Ultimately, this can guide experiments to investigate what happens when information processing at synapses breaks down, for example in diseases such as epilepsy.

Cooperative binding of multiple $Ca^{2+}$ ions to the SV fusion machinery increases the probability of SV fusion ($pV_r$) in a non-linear manner (*Bollmann et al., 2000*; *Dodge and Rahamimoff, 1967*; *Schneggenburger and Neher, 2000*).

A distinguishing feature of synapses is their activity profile upon repeated AP activation, where responses deviate between successive stimuli, resulting in either short-term facilitation (STF) or short-term depression (STD). This short-term plasticity (STP) fulfils essential temporal computational tasks (*Abbott and Regehr, 2004*). Postsynaptic STP mechanisms can involve altered responsiveness of receptors to NT binding, while presynaptic mechanisms can involve alterations in $Ca^{2+}$ signalling and –sensitivity of SV fusion (*von Gersdorff and Borst, 2002*; *Zucker and Regehr, 2002*). Presynaptic STD is often attributed to high $pV_r$ synapses, where a single AP causes significant depletion of the RRP. In contrast, presynaptic STF has often been attributed to synapses with low initial $pV_r$ and a rapid $pV_r$ increase during successive APs. This was often linked to changes in $Ca^{2+}$ signalling, for instance by rapid regulation of $Ca^{2+}$ channels (*Borst and Sakmann, 1998*; *Nanou and Catterall, 2018*), saturation of local $Ca^{2+}$ buffers (*Eggermann et al., 2012*; *Felmy et al., 2003*; *Matveev et al., 2004*), or the accumulation of intracellular $Ca^{2+}$ which may increase $pV_r$ either directly or via 'facilitation sensors' (*Jackman and Regehr, 2017*; *Katz and Miledi, 1968*). Alternatively, fast mechanisms increasing the RRP were proposed (*Fioravante and Regehr, 2011*; *Gustafsson et al., 2019*; *Pan and Zucker, 2009*; *Pulido and Marty, 2017*).

The coupling distance between $Ca^{2+}$ channels and primed SVs is an important factor governing $pV_r$ (*Böhme et al., 2018*; *Eggermann et al., 2012*; *Stanley, 2016*). Previous mathematical models describing SV fusion rates from simulated intracellular $Ca^{2+}$ transients have in many cases relied on the assumption of uniform (or near uniform) distances between SV release sites surrounding a cluster of $Ca^{2+}$ channels and such conditions were shown to generate STF (*Böhme et al., 2016*; *Meinrenken et al., 2002*; *Nakamura et al., 2015*; *Vyleta and Jonas, 2014*). However, alternative SV release site:$Ca^{2+}$ channel topologies have been proposed, including two distinct perimeter distances, tight, one-to-one connections of SVs and channels, or random placement of either the channels, the SVs, or both (*He et al., 2019*; *Böhme et al., 2016*; *Chen et al., 2015*; *Guerrier and Holcman, 2018*; *Keller et al., 2015*; *Shahrezaei et al., 2006*; *Stanley, 2016*; *Wong et al., 2014*). So far, the precise relationship between SV release sites and voltage gated $Ca^{2+}$ channels on the

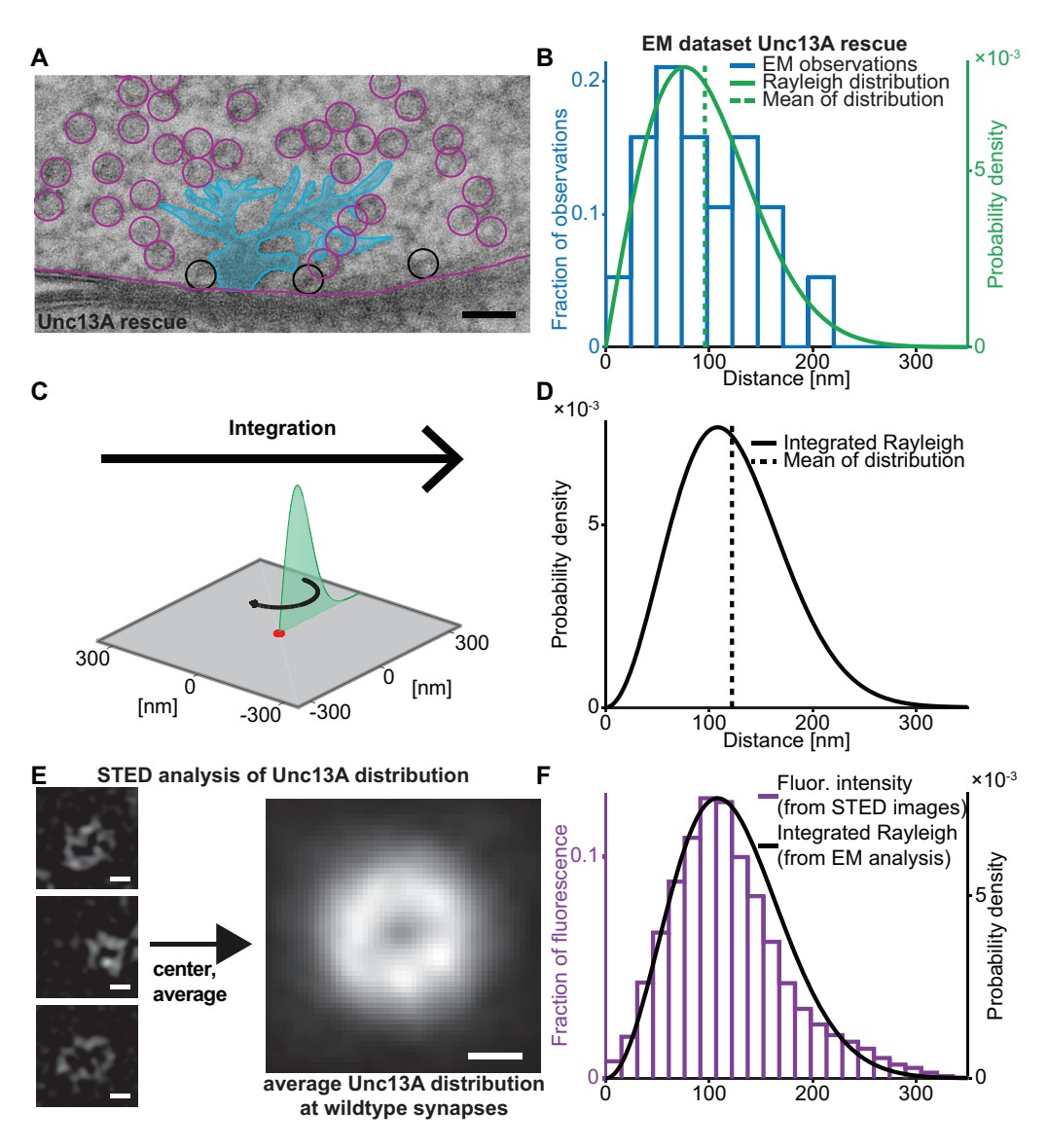

**Figure 1.** Deriving the spatial docked SV distribution. (A) Example EM image of an NMJ active zone (AZ) obtained from a 3rd instar *Drosophila* larva expressing the dominant Unc13A isoform after high pressure freeze fixation (Unc13A rescue: *elav-GAL4/+;;UAS-Unc13A-GFP/+;P84200/P84200*). The image captures a T-bar cross section. For clarity, the T-bar is colored in light blue, SVs are indicated with circles, the outline of the presynaptic plasma membrane is shown (magenta). Docked SVs are marked with black circles (non-docked in magenta). Black scale bar: 50 nm. (B) Histogram of the distances of docked SVs to the T-bar center obtained from EM micrographs 19 SVs observed in n = 10 EM cross-sections/cells from at least two animals, the same distance measurements had previously been used for the analysis depicted in Figure 5 of *Reddy-Alla et al. (2017)*. The solid green line is the fitted Rayleigh distribution ($\sigma$ = 76.5154 nm, mean is 95.9 nm, standard deviation, SD is 50.1 nm). (C) The one-dimensional Rayleigh distribution (green line) is integrated in order to estimate the docked SV distance distribution in the whole presynapse. (D) The integrated Rayleigh distribution is more symmetric, and the mean increases to 122.1 nm. SD is 51.5 nm. (E) The three left example images show wildtype (*w[1118]*) AZs stained against Unc13A and imaged on a STED microscope. The right hand image shows the average fluorescence signal for 524 individual centered AZ images from 16 different NMJs and more than three different animals (see Materials and methods for details). White scale bars: 100 nm. (F) Histogram of fluorescence intensities against distance from the AZ center, as derived from the average STED image plotted together with the integrated Rayleigh distribution derived from the EM analysis (replotted from panel D), showing a close agreement between the two approaches. Additional EM analysis of wildtype flies and the analysis of an independent STED experiment are compared to the data depicted here in *Figure 1—figure supplement 1*. Used genotype: Unc13A rescue (panel A, B), *w[1118]* (panel E, F). Materials and methods section 'Fly husbandry, genotypes and handling' lists all genotypes. Raw data corresponding to the depicted histograms can be found in the accompanying source data file (*Figure 1—source data 1*). Scripts used for analysis of average STED image and plotting of histograms in 1B and 1F can be found in accompanying source data zip file (*Figure 1—source data 2*).

*Figure 1 continued on next page*

*Figure 1 continued*

The online version of this article includes the following source data and figure supplement(s) for figure 1:

**Source data 1.** Source data for graphs in *Figure 1* and *Figure 1—figure supplement 1*.
**Source data 2.** Matlab codes used for data analysis, original images, and instructions for analysis depicted in *Figure 1* and *Figure 1—figure supplement 1*.
**Figure supplement 1.** EM + STED vesicle positions are consistent between independent datasets and overlapping with each other.

nanometre scale is unknown for most synapses, primarily owing to technical difficulties to reliably map their precise spatial distribution. However, (M)Unc13 proteins were recently identified as a molecular marker of SV release sites (*Reddy-Alla et al., 2017*; *Sakamoto et al., 2018*) and super-resolution (STED) microscopy revealed that these sites surround a cluster of voltage gated $Ca^{2+}$ channels in the center of AZs of the glutamatergic *Drosophila melanogaster* neuromuscular junction (NMJ) (*Böhme et al., 2016*; *Böhme et al., 2019*).

Here, by relying on the unique advantage of being able to precisely map SV release site:$Ca^{2+}$-channel topology we study its consequence for short-term plasticity at the *Drosophila* NMJ. Topologies were measured using electron microscopy (EM) following high pressure freeze fixation (HPF) or STED microscopy of Unc13 which both revealed a broad distribution of $Ca^{2+}$ channel coupling distances. Stochastic simulations were key to identify facilitation mechanisms in the light of heterogenous SV release site:$Ca^{2+}$ channel distances. Contrasting these simulations to physiological data revealed that models explaining STF through gradual increase in $pV_r$ (from now on called 'pV$_r$-based models') are inconsistent with the experiment while models of activity-dependent regulation of the RRP account for STP profiles and synaptic variance.

## Results

### Distances between docked SVs and $Ca^{2+}$ channels are broadly distributed

We first set out to quantify the SV release site:$Ca^{2+}$ channel topology. For this we analysed EM micrographs of AZ cross-sections and quantified the distance between docked SVs (i.e. SVs touching the plasma membrane) and the centre of electron dense 'T-bars' (where the voltage gated $Ca^{2+}$ channels are located *Fouquet et al. (2009)*; *Kawasaki et al. (2004)*; *Figure 1A*). In wildtype animals, this leads to a broad distribution of distances ('EM dataset wildtype', *Figure 1—figure supplement 1A*; *Böhme et al., 2016*; *Bruckner et al., 2017*). At the *Drosophila* NMJ, the two isoforms Unc13A and –B confer SV docking and priming, but the vast majority (~95%) of neurotransmitter release and docking of SVs with short coupling distances is mediated by Unc13A (*Böhme et al., 2016*). We therefore investigated the docked SV distribution in flies expressing only the dominant Unc13A isoform (Unc13A rescue, see Materials and methods for exact genotypes) which showed a very similar, broad distribution of distances as wildtype animals ('EM-dataset Unc13A rescue') (*Reddy-Alla et al., 2017*; *Figure 1A,B*). In both cases, distance distributions were well described by a Rayleigh distribution (*Figure 1B*, *Figure 1—figure supplement 1A*, solid green lines). The EM micrographs studied here are a cut cross-section of a three-dimensional synapse. To derive the relevant coupling distance distribution for all release sites (including the ones outside the cross-section), the Rayleigh distribution was integrated around a circle (*Figure 1C*), resulting in the following probability density function (pdf, see Materials and methods for derivation):

$$g(x) = \frac{\sqrt{2}}{\sqrt{\pi} \cdot \sigma^3} \cdot x^2 \cdot e^{-x^2/(2\sigma^2)}$$

These pdfs were more symmetrical than the ones from the cross-sections and peaked at larger distances (as expected from the increase in AZ area with increasing radius) (*Figure 1D*). The estimation of this pdf was very robust, resulting in near identical curves for the two EM datasets (*Figure 1—figure supplement 1B*).

We also used an independent approach to investigate the distribution of docked SV:$Ca^{2+}$ channel coupling distances without relying on the integration of docked SV observations from cross-sections:

since (M)Unc13 was recently described as a molecular marker of SV release sites (*Reddy-Alla et al., 2017*; *Sakamoto et al., 2018*) we investigated AZ images of wildtype NMJs stained against Unc13A (*Böhme et al., 2019*). Hundreds of individual AZ STED images (lateral resolution of approx. 40 nm) were aligned and averaged to obtain an average image of the AZ (*Figure 1E*), which revealed a ring-like distribution of the Unc13A fluorescence. In previous works we had established that the voltage gated $Ca^{2+}$ channels reside in the center of this ring (*Böhme et al., 2016*). As this average image already reflects the distribution throughout the AZ area (unlike for the EM data above where an integration was necessary) the distribution of coupling distances can directly be computed based on pixel intensities and their distance to the AZ centre. Two independent datasets where analysed, resulting in very similar average images and distance distributions ('wildtype STED dataset 1 and 2', *Figure 1—figure supplement 1*).

Remarkably, although the two approaches (EM and STED microscopy) were completely independent, the distributions of coupling distances quantified by either method coincided very well (*Figure 1F*, *Figure 1—figure supplement 1D*; note that the integrated Rayleigh distributions were determined from EM micrographs and integration; they were NOT fit to the Unc13A distribution), supporting the accuracy of this realistic release site topology. The compliance between SV docking positions and Unc13A distribution further indicates that SVs dock to the plasma membrane where priming proteins are available, and therefore the entire distribution of docked SVs is potentially available for synaptic release (*Imig et al., 2014*).

## Physiological assessment of short-term facilitation and depression at the *Drosophila* NMJ

Having identified the high degree of heterogeneity in the docked SV:$Ca^{2+}$ channel coupling distances, we became interested in how this affected synaptic function. We therefore characterized synaptic transmission at control NMJs (*Ok6-GAL4* crossed to *w[1118]*) in two electrode voltage clamp experiments. A common method to quantitatively evaluate synaptic responses and their STP behaviour is to vary the $Ca^{2+}$ concentration of the extracellular solution which affects AP-induced $Ca^{2+}$ influx (see below). We used this approach and investigated responses evoked by repetitive (paired-pulse) AP stimulations (10 ms interval). In line with classical studies (*Dodge and Rahamimoff, 1967*), our results display an increase of the evoked Excitatory Junctional Current (eEJC) responses to the first AP ($eEJC_1$ amplitudes) with increasing extracellular $Ca^{2+}$ (*Figure 2A,B*). STP was assessed by determining the paired-pulse ratio (PPR): the amplitude of the second response divided by first. The $eEJC_2$-amplitude was determined taking the decay of $eEJC_1$ into account (see insert in *Figure 2C*, *Figure 2—figure supplement 1A*). At low extracellular $Ca^{2+}$ (0.75 mM), we observed strong STF (with an average PPR value of 1.80), which shifted towards depression (PPR < 1) with increasing $Ca^{2+}$ concentrations (*Figure 2C,D*). Thus, the same NMJ displays both facilitation and depression depending on the extracellular $Ca^{2+}$ concentration, making this a suitable model synapse to investigate STP behaviour.

In panels B and D the mean $eEJC_1$ amplitudes and PPRs from six animals are shown and the error bars indicate standard deviation, SD (across all animals). We also examined the variation of repeated AP-evoked responses at the same NMJ between trials (10 s apart) at different extracellular $Ca^{2+}$-concentrations (*Figure 2E,F*). At low concentrations (0.75 mM), the probability of transmitter release is low, resulting in a low mean $eEJC_1$ amplitude with little variation (*Figure 2E,F*, *Figure 2—figure supplement 2* ). With increasing extracellular $Ca^{2+}$, the likelihood of SV fusion increased and initially so did the variance (e.g. at 1.5 mM extracellular $Ca^{2+}$). However, further increase in extracellular $Ca^{2+}$ (3 mM, 6 mM, 10 mM) led to a drop in variance (*Figure 2E*, *Figure 2—figure supplement 2*). *Figure 2F* depicts this average 'variance-mean' relationship from 6 cells (means of cell means and means of cell variances, error bars indicate SEM). When assuming a binomial model, this approach has often been used to estimate the number of release sites $n_{sites}$ and the size of the postsynaptic response elicited by a single SV ($q$) (*Clements and Silver, 2000*). In agreement with previous studies of the NMJ this relationship was well described by a parabola with forced intercept at y = 0 and $n_{sites}$ = 164 and q = 0.64 nA (*Figure 2F*, *Figure 2—figure supplement 2*; *Matkovic et al., 2013*; *Müller et al., 2012*; *Weyhersmüller et al., 2011*).

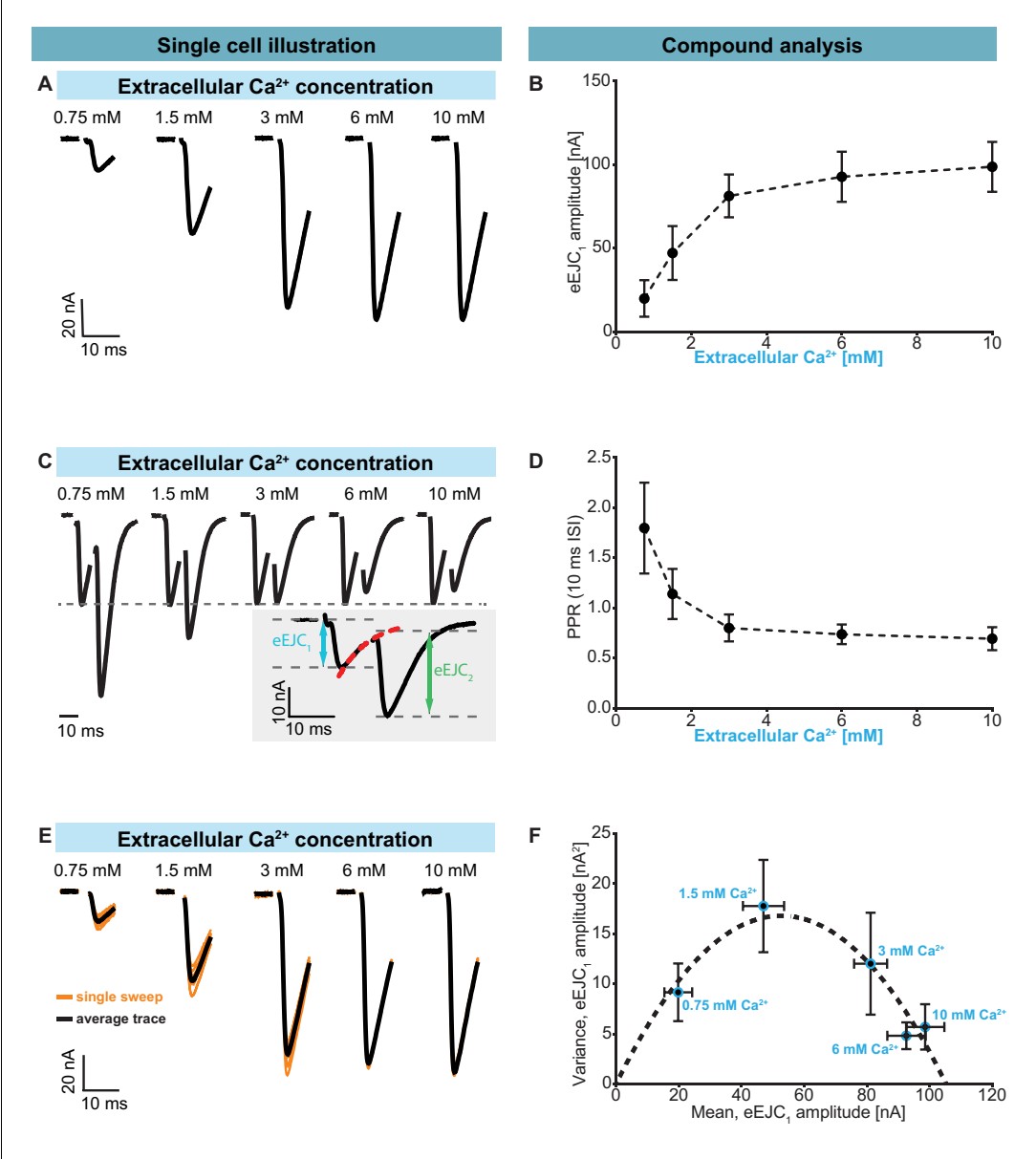

**Figure 2.** Characterization of short-term plasticity at the *Drosophila melanogaster* NMJ. Two-electrode voltage clamp recordings of AP-evoked synaptic transmission in muscle 6 NMJs (genotype: *Ok6-GAL4/+* (*Ok6-Gal4/II* crossed to *w[1118]*)). Left panel (**A, C, E**) shows example traces from one cell. Right panel (**B, D, F**) shows quantification across cells. (**A**) Representative eEJC traces from a single cell measured at different $Ca^{2+}$ concentrations (0.75–10 mM). (**B**) Average $eEJC_1$ amplitudes and SD from six animals as a function of extracellular $Ca^{2+}$ concentration. (**C**) Representative eEJC traces of paired pulse paradigm (10 ms inter-stimulus interval, normalized to $eEJC_1$) from single cell measured at different $Ca^{2+}$ concentrations (0.75–10 mM). While STF can be seen at the two lowest extracellular $Ca^{2+}$ concentrations (0.75 and 1.5 mM), the cell exhibits STD for extracellular $Ca^{2+}$ concentrations of 3 mM or more. Insert (gray background) shows calculation of $eEJC_2$. An exponential function was fitted to the decay to estimate the baseline for the second response (see *Figure 1—figure supplement 1* and Materials and methods for details). (**D**) Mean and SD of PPR values (6 cells from six animals) at different $Ca^{2+}$ concentrations. (**E**) Experiment to assess variance of repeated synaptic responses in a single cell. $eEJC_1$ traces in response to nine consecutive AP stimulations (10 s interval) are shown (orange lines) together with the mean $eEJC_1$ response (black line) at different extracellular $Ca^{2+}$ concentrations (0.75–10 mM, see Materials and methods). (**F**) Plot of mean $eEJC_1$ variance as a function of the mean $eEJC_1$ amplitude across 6 cells from six animals for each indicated $Ca^{2+}$ concentration. The curve shows best fitted parabola with intercept forced at (0,0) (Var = $-0.0061*<eEJC_1>^2+0.6375$ nA$*<eEJC_1>$, corresponding to $n_{sites} = 164$ and $q = 0.64$ nA when assuming a classical binomial model (*Clements and Silver, 2000*), see Materials and methods). For the variance-mean relationship of the single cell depicted in *Figure 2E*, please refer to *Figure 2—figure supplement 2*. Experiments were performed in *Ok6-Gal4/+* 3rd instar larvae, often used as a control genotype for experiments using cell-specific driver lines. Separate experiments were performed to ensure that this genotype showed similar synaptic responses and STP behavior as wildtype animals (*Figure 2—figure supplement 3*). Used genotype: *Ok6-Gal4/II* crossed to *w[1118]*. Materials and methods section 'Fly husbandry,

*Figure 2 continued on next page*

*Figure 2 continued*

genotypes and handling' lists all exact genotypes. Data points depict means, error bars are SDs across cells except in (F), where error bars show SEM. Raw data corresponding to the depicted graphs can be found in the accompanying source data file (*Figure 2—source data 1*). Scripts for analysis of recorded traces are found in accompanying source data zip file (*Figure 2—source data 2*). Raw traces from paired-pulse experiments summarized in *Figure 2* and *Figure 2—figure supplements 2* and *3* can be found in *Figure 2—source data 2*; *Figure 2—figure supplement 1—source data 1*; *Figure 2—figure supplement 3—source data 1*. Estimation of $eEJC_2$ amplitudes and fitting of a smooth mEJC function (used in simulations, see Materials and methods) are illustrated in *Figure 2—figure supplement 1*.

The online version of this article includes the following source data and figure supplement(s) for figure 2:

**Source data 1.** Raw data for experiments displayed in *Figure 2* and *Figure 2—figure supplement 2*.
**Source data 2.** Matlab code used for data analysis of electrophysiological traces.
**Source data 3.** Raw data which was used for depicted anaylsis in *Figure 2* and *Figure 2—figure supplement 3*.
**Figure supplement 1.** Illustration of analysis of experimental electophysiological data.
**Figure supplement 1—source data 1.** Raw data for mEJC recordings used for simulations in *Figure 2—figure supplement 1*.
**Figure supplement 2.** Illustration of fluctuation analysis (quantification across cells shown in *Figure 2F*) in a single representative cell.
**Figure supplement 3.** Electrophysiological comparison of synaptic transmission in wildtype (*w[1118], +/+*) (black) and *Ok6-Gal4/+* (orange) flies.
**Figure supplement 3—source data 1.** Raw data for experiments displayed in *Figure 2—figure supplement 3*.

## Simulation of AP-induced $Ca^{2+}$ signals

Having determined the distribution of coupling distances (*Figure 1*) and the physiological properties of the NMJ synapse (*Figure 2*), we next sought to compare how the one affected the other. There are two things two consider here. First of all, the SV release probability steeply depends on the 4th to 5th power of the local $Ca^{2+}$ concentration (*Neher and Sakaba, 2008*). Secondly, because of the strong buffering of $Ca^{2+}$ signals at the synapse, the magnitude of the AP-evoked $Ca^{2+}$ transients dramatically declines with distance from the $Ca^{2+}$ channel (*Böhme et al., 2018*; *Eggermann et al., 2012*). These two phenomena together make the vesicular release probability extremely sensitive to the coupling distance to the $Ca^{2+}$ channels. Because we find that this distance is highly heterogeneous among SVs within the same NMJ, the question arises how these two properties (heterogeneity of distances combined with a strong distance dependence of $pV_r$) functionally impact on synaptic transmission. Indeed, approaches by several labs to map the activity of individual NMJ AZs revealed highly heterogeneous activity profiles (*Akbergenova et al., 2018*; *Gratz et al., 2019*; *Muhammad et al., 2015*; *Peled and Isacoff, 2011*).

To quantitatively investigate the functional impact of heterogeneous SV placement, we wanted to use mathematical modelling to predict AP-induced fusion events of docked SVs placed according to the found distribution. A prerequisite for this is to first faithfully simulate local, AP-induced $Ca^{2+}$ signals throughout the AZ (such that the local transients at each docking site are known). We first determined the relevant AZ dimensions at the *Drosophila* NMJ, which, similarly to the murine Calyx of Held, is characterized by many AZs operating in parallel. We therefore followed previous suggestions from the Calyx using a box with reflective boundaries containing a cluster of $Ca^{2+}$ channels in the base centre (*Meinrenken et al., 2002*). The base dimensions (length = width) were determined as the mean inter-AZ distance of all AZs to their four closest neighbours (because of the 4-fold symmetry) from NMJs stained against the AZ-marker BRP (*Kittel et al., 2006*; *Wagh et al., 2006*; *Figure 3A*). To save computation time, we further simplified to a cylindrical simulation (where the distance to the $Ca^{2+}$ channel is the only relevant parameter) covering the same AZ area (*Figure 3B*, *Table 1*).

To simulate the electrophysiological experiments above, where the extracellular $Ca^{2+}$ concentration was varied (*Figure 2*), it was important to establish how the extracellular $Ca^{2+}$ concentration influenced AP-induced $Ca^{2+}$ influx. In particular, it is known that $Ca^{2+}$ currents saturate at high extracellular $Ca^{2+}$ concentrations (*Church and Stanley, 1996*). Unlike other systems, the presynaptic NMJ terminals are not accessible to electrophysiological recordings, so we could not measure the currents directly. We therefore used a fluorescence-based approach as a proxy. AP-evoked $Ca^{2+}$ influx was assessed in flies presynaptically expressing the $Ca^{2+}$-dependent fluorescence reporter GCaMP6m (*;P{y[+t7.7] w[+mC]=20XUAS-IVS-GCaMP6m}attP40/Ok6-GAL4*). Fluorescence increase was monitored upon stimulation with 20 APs (at 20 Hz) while varying the extracellular $Ca^{2+}$ concentration and showed saturation behaviour for high concentrations (*Figure 3—figure supplement 1*). This is consistent with a previously described Michaelis-Menten type saturation of fluorescence

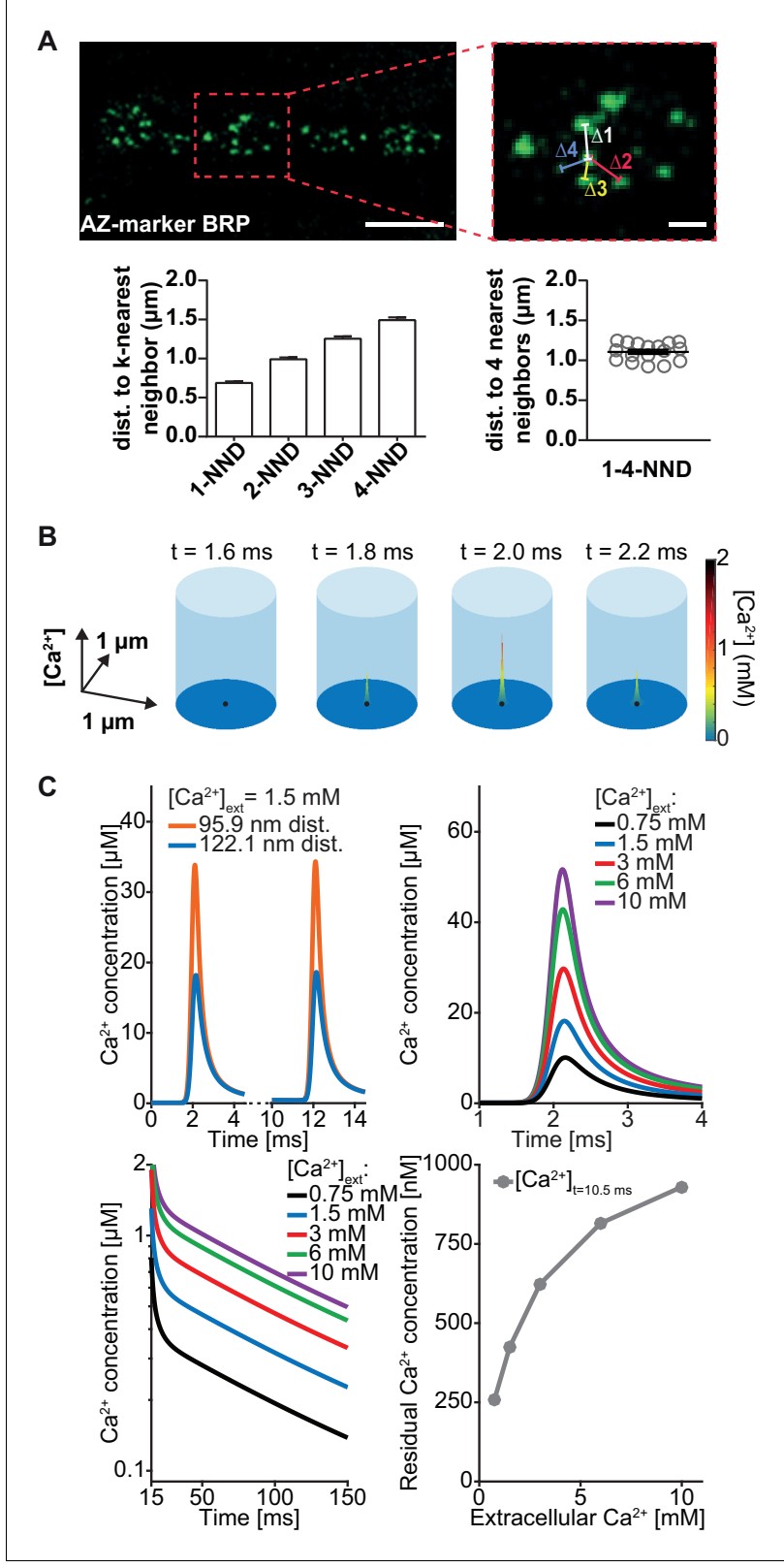

**Figure 3.** Simulation of AP-induced synaptic $Ca^{2+}$ profiles. (**A**) Estimation of the simulation volume and $Ca^{2+}$ simulations. The left hand image shows a confocal scan of a $3^{rd}$ instar larval NMJ stained against the AZ marker Bruchpilot (BRP) (genotype: *w[1118]; P{w[+mC]=Mhc-SynapGCaMP6f}3–5* (Bloomington Stock No. 67739). The right hand image shows a higher magnification of the indicated region. To determine the dimensions of the

*Figure 3 continued on next page*

*Figure 3 continued*

simulation volume, the average distance of each AZ to its closest four neighboring AZs (*k*-NND = $k^{th}$ nearest neighbor distance) was determined. The average inter-AZ distance to each of the closest four neighboring AZs (1- through 4-NND) is depicted on the left. Average and SEM of inter-AZ distances (1-4-NND) are depicted on the right. White scale bars: Left: 5 µm; right: 1 µm. (**B**) Example illustration of the $Ca^{2+}$ simulation. The simulation volume is a cylinder whose base area (radius 624 nm) is the same as a square with side length of the mean 1–4-NND. The local $Ca^{2+}$ concentration is shown at different time points following an AP-induced Gaussian $Ca^{2+}$ current (the area/height is a free parameter, see *Table 2*, the FWHM is 0.36 ms). The simulation started at t=0 ms, $Ca^{2+}$ influx was initiated at t=0.5 ms and peaked at t=2 ms. The $Ca^{2+}$ (point) source is located in the AZ center (black dot) and the $Ca^{2+}$ concentration is determined at 10 nm height from the plasma membrane. (**C**) Example simulation of the local $Ca^{2+}$ concentration profile in response to stimulation with a pair of APs (current was initiated at 0.5 and 10.5 ms and peaked at 2 and 12 ms). Simulations were performed using the best fit parameters of the single sensor model described below (see *Figure 4*, *Table 2*). Top left: $Ca^{2+}$ transients in response to the first AP at two distances: 95.9 nm and 122.1 nm (the mean of Rayleigh/integrated Rayleigh). Top right: AP-induced $Ca^{2+}$ transient at 122.1 nm for all experimental extracellular $Ca^{2+}$ concentrations. Bottom left: Semi-logarithmic plot of $Ca^{2+}$ decays toward baseline after the $2^{nd}$ transient (residual $Ca^{2+}$) at different extracellular $Ca^{2+}$ concentrations ($[Ca^{2+}]_{ext}$). Time constant of decay is $\tau$ = 111 ms. Bottom right: Residual $Ca^{2+}$ levels at 122.1 nm after 10.5 ms of simulation as a function of extracellular $Ca^{2+}$ concentrations. Data depicted in panel A were collected from 17 different animals. Used genotype: *w[1118]; P{w[+mC]=Mhc-SynapGCaMP6f}3–5* (Bloomington Stock No. 67739, panel A). Materials and methods section 'Fly husbandry, genotypes and handling' lists all exact genotypes. Values used for graphs can be found in the accompanying source data file (*Figure 3— source data 1*). GCaMP6m experiment is summarized in *Figure 3—figure supplement 1*. $Ca^{2+}$ signals for all optimised models (below) are summarised in *Figure 3—figure supplement 2*.

The online version of this article includes the following source data and figure supplement(s) for figure 3:

**Source data 1.** Average cell-wise mean fluorescence values and fit parameters of hill-curve fit on presynaptic GCaMP data, and NND values.

**Figure supplement 1.** Experiment to determine the dependence of AP-induced $Ca^{2+}$ influx on the extracellular $Ca^{2+}$ concentration.

**Figure supplement 2.** $Ca^{2+}$ profiles of all models using best fit parameters (reported in *Table 1*; *Table 2*).

---

responses of a $Ca^{2+}$-sensitive dye upon single AP stimulation at varying extracellular $Ca^{2+}$ concentrations at the Calyx of Held, where half-maximal $Ca^{2+}$ influx was observed at 2.6 mM extracellular $Ca^{2+}$ (*Schneggenburger et al., 1999*). This relationship was successfully used in the past to predict $Ca^{2+}$ influx in modeling approaches *Trommershäuser et al. (2003)*. In our measurements, we determined a half maximal fluorescence response at a very similar concentration of 2.68 mM extracellular $Ca^{2+}$ and therefore used this value as $K_{M,current}$ in a Michaelis-Menten equation (Materials and methods, *Equation 5*) to calculate AP-induced presynaptic $Ca^{2+}$ influx. The second parameter of the Michaelis-Menten equation, (the maximal $Ca^{2+}$ current charge, $Q_{max}$) was optimized for each model (*Figure 3—figure supplement 2*, for parameter explanations and best fit parameters see *Table 2*). We furthermore assumed that basal, intracellular $Ca^{2+}$ concentrations at rest were also slightly dependent on the extracellular $Ca^{2+}$ levels in a Michaelis-Menten relationship with the same dependency ($K_{M,current}$) and a maximal resting $Ca^{2+}$ concentration of 190 nM (resulting in 68 nM presynaptic basal $Ca^{2+}$ concentration at 1.5 mM external $Ca^{2+}$). With these and further parameters taken from the literature on $Ca^{2+}$ diffusion and buffering (see *Table 1*) the temporal profile of $Ca^{2+}$ signals in response to paired AP stimulation (10 ms interval) could be calculated at all AZ locations using the software CalC (*Matveev et al., 2002*; *Figure 3C*, *Figure 3—figure supplement 2*). This enabled us to perform simulations of NT release from vesicles placed according to the distribution described above.

## Stochastic simulations and fitting of release models

In the past, we and others have often relied on deterministic simulations based on numerical integration of kinetic reaction schemes (ordinary differential equations, ODEs). These are computationally effective and fully reproducible, making them well-behaved and ideal for the optimisation of parameters (a property that was also used here for initial parameter searches, see Materials and methods). However, NT release is quantal and relies on only a few (hundred) SVs, indicating that stochasticity plays a large role (*Gillespie, 2007*). Moreover, deterministic simulations always predict identical

**Table 1.** Parameters of Ca$^{2+}$ and buffer dynamics.

| Simulation volume | | |
|---|---|---|
| r | Radius of cylindric simulation volume | 623.99 nm |
| h | Height of cylindric simulation volume | 1 µm |
| n$_{grid}$ | Spatial grid points in CalC simulation | 71 × 101 (radius x height) |
| **Ca$^{2+}$** | | |
| Q$_{max}$ | Scaling of the total amount of Ca$^{2+}$ charge influx | Fitted (all models), see *Table 2* |
| D$_{Ca}$ | Diffusion coefficient of Ca$^{2+}$ (*Allbritton et al., 1992*) | 0.223 µm$^2$/ms |
| [Ca]$_{bgr}$ | Background Ca$^{2+}$ | $\frac{[Ca^{2+}]_{ext}}{[Ca^{2+}]_{ext}+K_{M,current}} \cdot 190$ nM |
| K$_{M,current}$ | Set to the same value as K$_{M,fluo}$ determined in GCaMP6 experiments | 2.679 mM |
| Ca$^{2+}$ uptake | Volume-distributed uptake (*Helmchen et al., 1997*) | 0.4 ms$^{-1}$ |
| **Buffer Bm ('fixed' buffer)** | | |
| D$_{Bm}$ | Diffusion coefficient | 0.001 µm$^2$/ms |
| K$_{D,Bm}$ | Equilibrium dissociation constant (*Xu et al., 1997*) | 100 µM |
| K$_{+,Bm}$ | Ca$^{2+}$ binding rate (*Xu et al., 1997*) | 0.1 (µM·ms)$^{-1}$ |
| K$_{-,Bm}$ | Ca$^{2+}$ unbinding rate: K$_{D,Bm}$·K$_{+,Bm}$ | 1 ms$^{-1}$ |
| Total Bm | Total concentration (bound+unbound) (*Xu et al., 1997*) | 4000 µM |
| **Buffer ATP** | | |
| D$_{ATP}$ | Diffusion coefficient (*Chen et al., 2015*) | 0.22 µm$^2$/ms |
| K$_{D,ATP}$ | Equilibrium dissociation constant (*Chen et al., 2015*) | 200 µM |
| K$_{+,ATP}$ | Ca$^{2+}$ binding rate (*Chen et al., 2015*) | 0.5 (µM·ms)$^{-1}$ |
| K$_{-,ATP}$ | Ca$^{2+}$ unbinding rate: K$_{D,ATP}$·K$_{+,ATP}$ | 100 ms$^{-1}$ |
| Total ATP | Total concentration (bound+unbound) (*Chen et al., 2015*) | 650 µM |
| **Resting Ca$^{2+}$** | | |
| K$_{M,current}$ | Michaelis Menten-constant of resting Ca$^{2+}$ (same as K$_{M,current}$ of Ca$^{2+}$ influx) | 2.679 mM |
| [Ca$^{2+}$]$_{max}$ | Asymptotic max value of resting Ca$^{2+}$ | 190 nM |

output making it impossible to analyse the synaptic variance between successive stimulations, which is a fundamental hallmark of synaptic transmission and an important physiological parameter (*Figure 2F*; *Scheuss and Neher, 2001*; *Vere-Jones, 1966*; *Zucker, 1973*). Stochastic simulations allow a prediction of variance which can help identify adequate models that will not only capture the mean of the data, but also its variance. To compare this, data points are now shown with error bars indicating the square root of the average variance between stimulations within a cell (*Figure 4C, E, 6E, G and 7E, G*). This is the relevant parameter since the model is designed to resemble an

'average' NMJ' and therefore cannot predict inter-animal variance. Finally, as we show here deterministic simulations cannot be compared to experimentally determined PPR values because of Jensen's inequality (full proof in Materials and methods, see *Figure 4—figure supplement 1*). Thus, stochastic simulations are necessary to account for SV pool sizes, realistic release site distributions, synaptic variance and STP. We thus implemented stochastic models of SV positions (drawn randomly from the distribution above) and SV $Ca^{2+}$ binding states based on inhomogeneous, continuous time Markov models with transition rates governing reaction probabilities (see Materials and methods for details).

We also needed to consider where new SVs would (re)dock once SVs had fused and implemented the simplest scenario of re-docking in the same positions. This ensures a stable distribution over time and agrees with the notion that vesicles prime into pre-defined release sites, which are stable over much longer time than a single priming/unpriming event (*Reddy-Alla et al., 2017*).

## A single-sensor model fails to induce sufficient facilitation and produces excessive variance

The first model we tested was the single-sensor model proposed by *Lou et al. (2005)*, where an SV binds up to 5 $Ca^{2+}$ ions, with each ion increasing its fusion rate or probability (*Figure 4A*, *Table 3*). Release sites were placed according to the distance distribution in *Figure 1D* and all sites were occupied by a primed SV prior to stimulation (i.e. the number of release sites equals the number of vesicles in the RRP). Sites becoming available following SV fusion were replenished from an unlimited vesicle pool, making the model identical to the one described by *Wölfel et al. (2007)*. $Ca^{2+}$ (un) binding kinetics were taken from *Wölfel et al. (2007) Table 3*, the values of the maximal $Ca^{2+}$ current charge ($Q_{max}$), the SV replenishment rate ($k_{rep}$) and the number of release sites ($n_{sites}$) were free parameters optimized to match the experimental data (see Materials and methods for details, best fit parameters in *Table 2*).

To be able to compare the output of this and all subsequent models to experimental data as depicted in *Figure 2* (postsynaptic eEJC measurements), the predicted fusion events were convolved with a typical postsynaptic response to the fusion of a single SV (mEJC, *Figure 2—figure supplement 1B*, see Materials and methods for more details). From the stochastic simulations (1000 runs each), we calculated the mean and variance of $eEJC_1$ amplitudes, and the mean and variance of PPRs at various extracellular $Ca^{2+}$ concentrations and contrasted these with the experimental data.

This single-sensor model was able to reproduce the $eEJC_1$ values (*Figure 4B,C*). Moreover, the model accounted for the STD typically observed at high extracellular $Ca^{2+}$ concentrations in the presence of rapid replenishment (*Hallermann et al., 2010*; *Miki et al., 2016*) (our best fit yielded $\tau \approx 6$ ms and reducing this rate led to unnaturally strong depression, *Figure 4E*, green curve+area). However, even despite rapid replenishment this model failed to reproduce the STF observed at low extracellular $Ca^{2+}$ (*Figure 4D,E*) and the variances predicted by this model were much larger than found experimentally (*Figure 4F,G*). The observation that $eEJC_1$ amplitudes were well accounted for, but STPs were not, may relate to the fact that this model was originally constructed to account for a single $Ca^{2+}$-triggered release event (*Lou et al., 2005*). As this model lacks a specialized mechanism to induce facilitation, residual $Ca^{2+}$ binding to the $Ca^{2+}$ sensor is the only facilitation method which appears to be insufficient (*Jackman and Regehr, 2017*; *Ma et al., 2015*; *Matveev et al., 2002*). This result differs from our previous study using this model where we had placed all SVs at the same distance to $Ca^{2+}$ channels which reliably produced STF (*Böhme et al., 2016*). So why does the same model fail to produce STF with this broad distribution of distances? To understand this we investigated the spatial distribution of SV release in simulations of the paired-pulse experiment at 0.75 mM extracellular $Ca^{2+}$ (*Figure 5*).

## In the absence of a facilitation mechanism, only part of the SV distribution is utilized

*Figure 5A* depicts two examples of synapses – seen from above – with SVs randomly placed according to the distance distribution in *Figure 1D*/5B. The synapse is shown immediately before $AP_1$, immediately after $AP_1$, immediately before $AP_2$ (i.e. after refilling) and immediately after $AP_2$ (the external $Ca^{2+}$ concentration was 0.75 mM). From this analysis it becomes clear that the $pV_r1$ caused by $AP_1$ essentially falls to zero around the middle of the SV distribution (*Figure 5B*, top panel). This

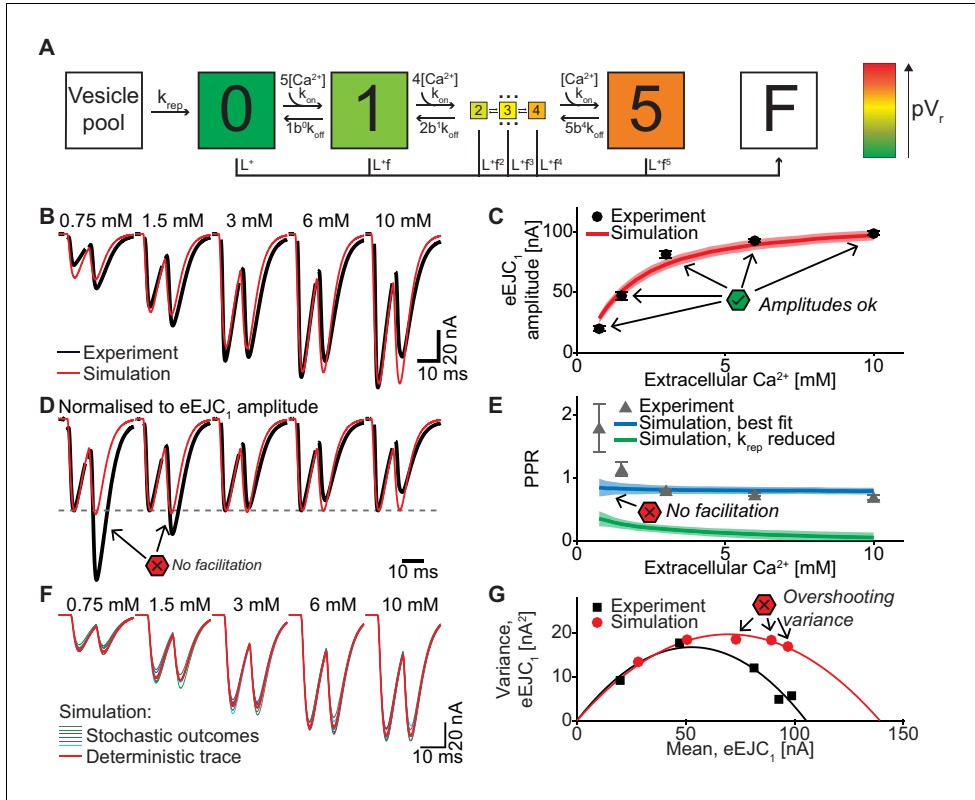

**Figure 4.** A single-sensor model reproduces the magnitude of transmission to single APs, but cannot account for STF and variances. (**A**) Diagram of the single-sensor model. Consecutive binding of up to 5 $Ca^{2+}$ ions to a vesicular $Ca^{2+}$ sensor increases the probability of SV fusion (transition to state F) indicated by the color of the state. Primed SVs can be replenished from an infinite Vesicle pool. (**B**) Experimental eEJC traces averaged over all cells (black) together with average simulated traces (red). (**C**) $eEJC_1$ amplitudes of experiment (black) and simulation (red). Error bars and colored bands show the standard deviations of data (see text) and simulations, respectively. Simulations reproduce $eEJC_1$ amplitudes well. (**D**) Average (over all cells), normalized eEJC traces of experiment (black) and simulation (red). Simulations obtained with this model lack facilitation, as indicated by the red symbols. (**E**) PPR values of experiment (gray) and best fit simulation (blue). Green curve show simulations with replenishment 100x slower than the fitted value illustrating the effect of replenishment on the PPR. Error bars and colored bands show standard deviation. Best fit simulations do not reproduce the facilitation observed in the experiment at low extracellular $Ca^{2+}$ concentrations. (**F**) Average simulated traces (red) and examples of different outcomes of the stochastic simulation (colors). (**G**) Plot of the mean synaptic variance vs. the mean $eEJC_1$ amplitudes, both from the experiment (black) and the simulations (red). The curves show the best fitted parabolas with forced intercept at (0,0) (simulation: Var = $-0.0041*<eEJC_1>^2+0.5669$ nA$*<eEJC_1>$, corresponding to $n_{sites}$ = 244 and q = 0.57 nA when assuming a classical binomial model (**Clements and Silver, 2000**), see Materials and methods). Simulations reveal too much variance in this model. Experimental data (example traces and means) depicted in panels B-E,G are replotted from **Figure 2A–D,F**. All parameters used for simulation can be found in **Tables 1–3**. Simulation scripts can be found in **Source code 1**. Results from simulations (means and SDs) can be found in the accompanying source data file (**Figure 4—source data 1**). Exploration of the difference between PPR estimations in deterministic and stochastic simulations are illustrated in **Figure 4—figure supplement 1**.

The online version of this article includes the following source data and figure supplement(s) for figure 4:

**Source data 1.** Simulation data for graphs in **Figure 4C, G, E** and **Figure 4—figure supplement 1**.
**Figure supplement 1.** Stochastic and deterministic simulations yield different PPR values.

means that only SVs close to the synapse center fuse, and these high-$pV_r$ SVs are depleted by $AP_1$. SV replenishment refills the majority (but not all) of those sites and thus $AP_2/pV_r2$ essentially draws on the same part of the distribution (**Figure 5B**, bottom panel). Because of this, and because the refilling is incomplete, this causes STD. Even with faster replenishment (which would be incompatible

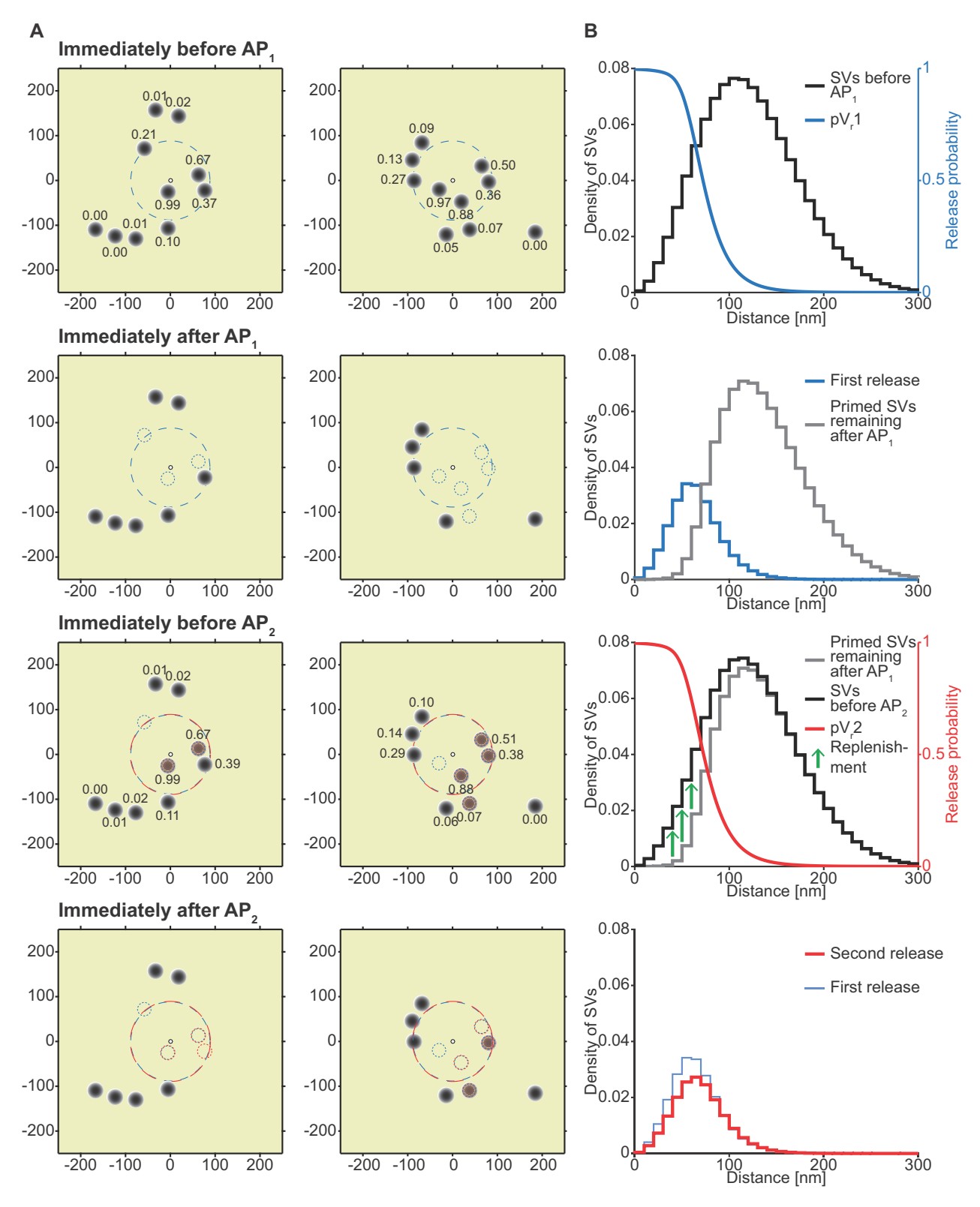

**Figure 5.** Analysis of the spatial dependence of SV fusion in the single-sensor model reveals a near-identical use of release sites during the two APs, thereby favoring STD. (**A**) Two examples of docked SVs stochastically placed according to the distribution described in *Figure 1D* and their behavior in the PPR simulation at 0.75 mM extracellular $Ca^{2+}$. For clarity, 10 SVs are shown per AZ (the actual number is likely lower) and only a central part of the AZ is shown. Top row: Prior to $AP_1$ SVs are primed (dark gray circles) and $pV_r1$ is indicated as numbers. The larger dashed, blue circle in the AZ center

*Figure 5 continued on next page*

*Figure 5 continued*

indicates pV$_r$1 = 0.25. Second row: After AP$_1$ some of the SVs have fused (dashed blue circles). Third row: Right before AP$_2$ some of the SVs that had fused in response to AP$_1$ have been replenished (orange shading), and pV$_r$2 is indicated as a number. The larger dashed, red circle indicates pV$_r$2 = 0.25. Bottom row: After AP$_2$ the second release has taken place. Small dashed circles indicate release from AP$_1$ and AP$_2$ (blue and red, respectively). The small increase in pV$_r$ caused by Ca$^{2+}$ accumulation cannot produce facilitation because of depletion of SVs. (**B**) The average simulation at the same time points as in (A). Histograms represent primed SVs (black and gray) as well as first and second release (blue and red) illustrating how release from AP$_1$ and AP$_2$ draw on the same subpopulation of SVs. The blue and red curves indicate the vesicular release probability as a function of distance during AP$_1$ (blue) and AP$_2$ (red). The green arrows show the repopulation of previously used sites via replenishment. AP$_2$ draws on the same portion of the SV distribution as AP$_1$ causing depression despite the fast replenishment mechanism. Parameters used for simulations can be found in *Tables 1–3*.

with the low PPR values at high extracellular Ca$^{2+}$, *Figure 4E*) this scenario would only lead to a modest increase of the PPR to values around 1. Therefore, our analysis reveals that large variation in Ca$^{2+}$ channel distances results in a specific problem to generate STF. Our analysis further indicates that with the best fit parameters of the single sensor model, the majority of SVs (those further away) is not utilized at all.

## A dual fusion-sensor model improves PPR values, but generates too little facilitation and suffers from asynchronous release and too much variance

The single-sensor model failed to reproduce the experimentally observed STF at low extracellular Ca$^{2+}$ concentrations because of the dominating depletion of SVs close to Ca$^{2+}$ channels, and the inability to draw on SVs further away. However, this situation may be improved by a second Ca$^{2+}$ sensor optimized to enhance the pV$_r$2 in response to AP$_2$. Indeed, in the absence of the primary Ca$^{2+}$ sensor for fusion, Ca$^{2+}$ sensitivity of synaptic transmission persists, which was explained by a dual sensor model (*Sun et al., 2007*). It was recently suggested that syt-7 functions alongside syt-1 as a Ca$^{2+}$ sensor for release (*Jackman et al., 2016*), and deterministic mathematical dual fusion-sensor model assuming homogeneous release probabilities (which implies homogeneous SV release site:Ca$^{2+}$ channel distances) was shown to generate facilitation (*Jackman and Regehr, 2017*). Similarly, stochastic modelling of NT release at the frog NMJ also showed a beneficial effect of a second fusion sensor for STF (*Ma et al., 2015*). We therefore explored whether a dual fusion sensor model could account for synaptic facilitation from realistic release site topologies.

The central idea of this dual fusion-sensor model is that while syt-1 is optimized to detect the rapid, AP-induced Ca$^{2+}$ transients (because of its fast Ca$^{2+}$ (un)binding rates, but fairly low Ca$^{2+}$ affinity), the cooperating Ca$^{2+}$ sensor is optimized to sense the residual Ca$^{2+}$ after this rapid transient (*Figure 3C*) (with slow Ca$^{2+}$ (un)binding, but high Ca$^{2+}$ affinity). The activation of this second sensor after (but not during) AP$_1$ could then enhance the release probability of the remaining SVs for AP$_2$ (*Figure 6A,B*). This is illustrated in *Figure 6B*, where k$_2$ (the on-rate of Ca$^{2+}$ binding to the slow sensor) is varied resulting in different time courses and amounts of Ca$^{2+}$ binding to the second sensor. Increasing the release probability is equivalent to lowering the energy barrier for SV fusion (*Schotten et al., 2015*). In this model both sensors regulate pV$_r$ and therefore additively lower the fusion barrier with each associated Ca$^{2+}$ ion (*Figure 6A*), resulting in multiplicative effects on the SV fusion rate. While the fast fusion reaction appears to have a 5-fold Ca$^{2+}$ cooperativity (*Bollmann et al., 2000*; *Burgalossi et al., 2010*; *Schneggenburger and Neher, 2000*), it is less clear what the Ca$^{2+}$ cooperativity of a second Ca$^{2+}$ sensor may be, although the fact that the cooperativity is reduced in the absence of the fast sensor (*Burgalossi et al., 2010*; *Kochubey and Schneggenburger, 2011*; *Sun et al., 2007*) could be taken as evidence for a Ca$^{2+}$ cooperativity < 5. We explored cooperativities 2, 3, 4, and 5 (cooperativities 2 and 5 are displayed in *Figure 6* and *Figure 6—figure supplement 1*). It is furthermore not clear whether such a sensor would be targeted to the SV (like syt-1 /-2), or whether it is present at the plasma membrane. Both scenarios are functionally possible and it was indeed reported that syt-7 is predominantly or partly localized to the plasma membrane (*Sugita et al., 2001*; *Weber et al., 2014*). A facilitation sensor on the plasma membrane would be more effective, which our simulations confirmed (not shown), because it would not be consumed by SV fusion, allowing the sensor to remain activated. We therefore present this version of the model

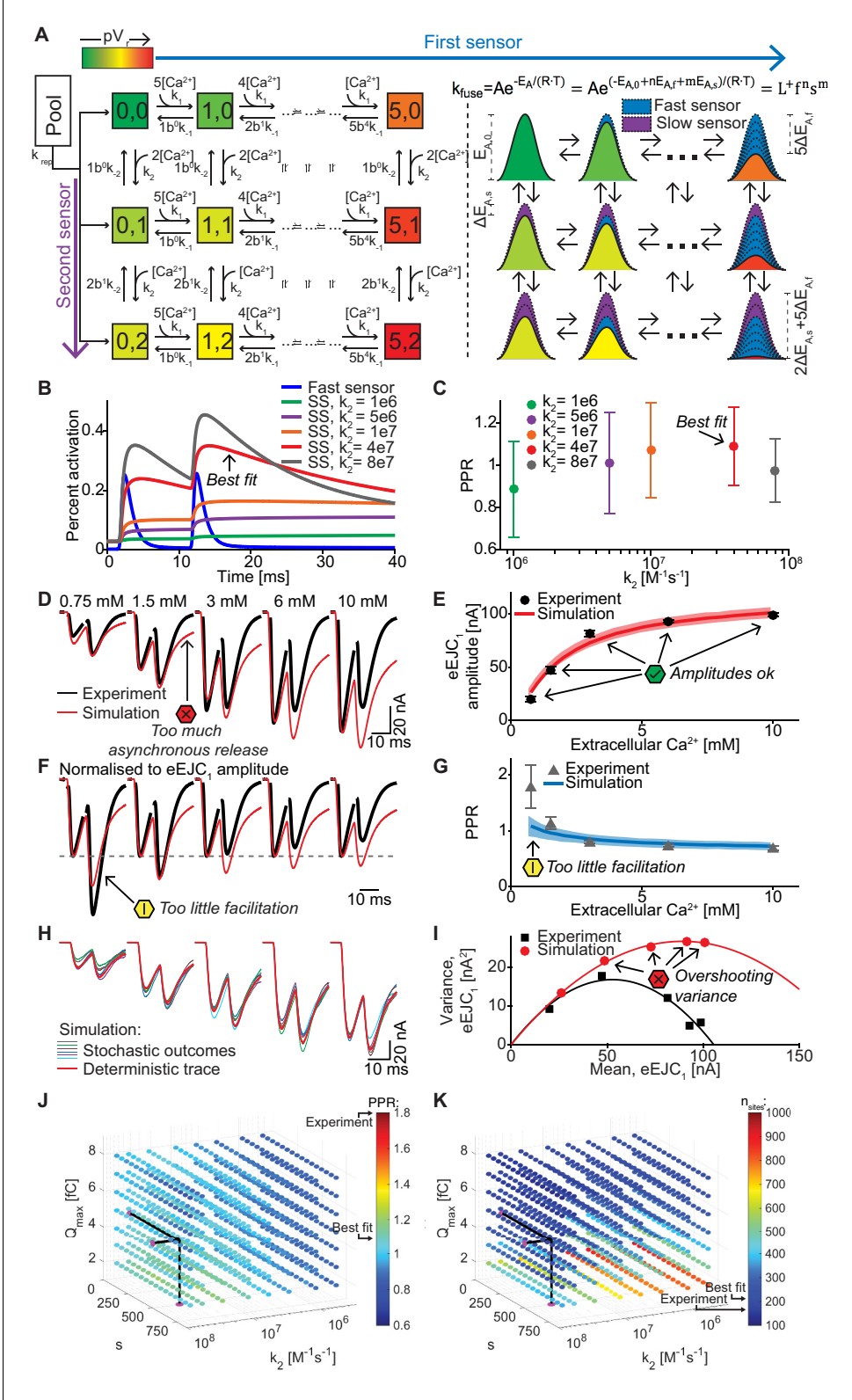

**Figure 6.** A dual fusion-sensor model of Ca²⁺ sensors cooperating for SV fusion improves STP behavior, but suffers from too little STF, asynchronous release and too much variance. (**A**) Diagram of the dual fusion-sensor model (left). A second Ca²⁺ sensor for fusion with slower kinetics can increase pVᵣ (indicated by color of each Ca²⁺ binding state). The second fusion sensor is assumed to act on the energy barrier in a similar way as the first sensor

*Figure 6 continued on next page*

*Figure 6 continued*

(right). The top right equation shows the relation between the fusion constant, $k_{fuse}$, and energy barrier modulation with $n$ and $m$ being the number of $Ca^{2+}$ bound to the first and second $Ca^{2+}$ sensor, respectively. $Ca^{2+}$ binding to the second sensor is described by similar equations as for the first sensor, but with different rate constants and impact on the energy barrier. (**B**) Simulation of $Ca^{2+}$ binding to the fast (blue) and slow (other colors) $Ca^{2+}$ sensor in simulations at 0.75 mM extracellular $Ca^{2+}$ with different $k_2$ values but with constant affinity (i.e. fixed ratio of $k_{-2}/k_2$). The binding is normalized to the maximal number of bound $Ca^{2+}$ to each sensor (5 and 2, respectively). For illustration purposes in this graph the fusion rate was set to 0 (because otherwise the fast sensor (blue line) would be consumed by SV fusion). $k_2 = 4e7\ M^{-1}s^{-1}$ (red trace) illustrates the situation for the optimal performance of the model (approximately best fit value). (**C**) PPR values in stochastic simulations with the same parameter choices as in (B) but allowing fusion. (**D**) Experimental eEJC traces (black) together with average simulated traces (red). Simulations show too much asynchronous release compared to experiments. (**E**) $eEJC_1$ amplitudes of experiment (black) and simulation (red). Error bars and colored bands show standard deviations of data and simulations, respectively. Simulations reproduce $eEJC_1$ amplitudes well. (**F**) Average, normalized eEJC traces of experiment (black) and simulation (red). Simulations show too little facilitation compared to experiment. (**G**) PPR values of experiment (gray) and simulation (blue). Error bars and colored bands show standard deviation. Simulations show too little facilitation compared to experiment. (**H**) Average simulated traces (red) and examples of different outcomes of the stochastic simulation (colors). (**I**) Plot of the mean synaptic variance vs. the mean $eEJC_1$ values, both from the experiment (black) and the simulations (red). Curves are the best fitted parabolas with forced intercept at (0,0) (simulation: $Var = -0.0034*<eEJC_1>^2 + 0.5992\ nA*<eEJC_1>$, corresponding to $n_{sites} = 294$ and q = 0.60 nA when assuming a classical binomial model (*Clements and Silver, 2000*), see Materials and methods). Simulations lead to too much variance at the highest $Ca^{2+}$ concentrations. (**J**) Parameter exploration of the second sensor varying the parameters $Q_{max}$, $k_2$, and s. Each ball represents a choice of parameters and the color indicates the average PPR value in stochastic simulations with 0.75 mM extracellular $Ca^{2+}$. None of the PPR values match the experiment (indicated by the black arrow). Black lines show the best fit parameters. (**K**) Same parameter choices as in (I). The colors indicate the number of RRP SVs in order to fit the $eEJC_1$ amplitudes at the five different experimental $Ca^{2+}$ concentrations. Black lines show the best fit parameters, and arrows show the experimental and best fit simulation values. Note that the best fit predicted more release sites than fluctuation analysis revealed in the experiment. Experimental data (example traces and means) depicted in panels D-G,I are replotted from *Figure 2A–D,F*. Parameters used for simulations can be found in *Tables 1–3*. Simulation scripts can be found in *Source code 1*. Results from simulations (means and SDs) can be found in the accompanying source data file (*Figure 6—source data 1*). Simulations of the dual fusion-sensor model with cooperativity 5 are summarized in *Figure 6—figure supplement 1*.

The online version of this article includes the following source data and figure supplement(s) for figure 6:

**Source data 1.** Simulation data for graphs in *Figure 6B, E, G, I* and *Figure 6—figure supplement 1B, D, E*.
**Figure supplement 1.** The dual fusion-sensor sensor model with cooperativity 5 (allowing the binding of 5 $Ca^{2+}$ to the second fusion sensor).

---

here. We used a second sensor with a $Ca^{2+}$ affinity of $K_D = 1.5\ \mu M$ (*Brandt et al., 2012*; *Jackman and Regehr, 2017*).

Like for the single-sensor model, all release sites were occupied with releasable vesicles ($n_{sites}$ equals the number of RRP vesicles) and their locations determined by drawing random numbers from the pdf. When fitting this model five parameters were varied: $Q_{max}$, $k_{rep}$, and $n_{sites}$ (like in the single-sensor model) together with $k_2$ ($Ca^{2+}$ association rate constant to the second sensor) and s (the factor describing the effect of the slow sensor on the energy barrier for fusion) (see *Table 2* for best fit parameters). The choice of $k_2$ had an effect on the PPR in simulations, confirming that the second sensor was able to improve the release following $AP_2$ (*Figure 6C*). *Figure 6D–I* show that the dual fusion-sensor model could fit the $eEJC_1$ amplitudes and the model slightly improved the higher PPR values at the low- and the lower PPR values at high extracellular $Ca^{2+}$ concentrations compared to the single sensor model (compare *Figures 4E* and *6G*). However, the model failed to produce the STF observed experimentally (the PPR values at 0.75 mM $Ca^{2+}$ were ~ 1.08 in the simulation compared to ~ 1.80 in the experiments). Another problem of the dual fusion-sensor model was that release became more asynchronous than observed experimentally (*Figure 6D*), which was due to the triggering of SV fusion in-between APs. Finally, predicted variances were much larger than the experimental values (*Figure 6I*).

In addition to the optimization, we systematically investigated a large region of the parameter space (*Figure 6J,K*), but found no combination of parameters that would be able to generate the

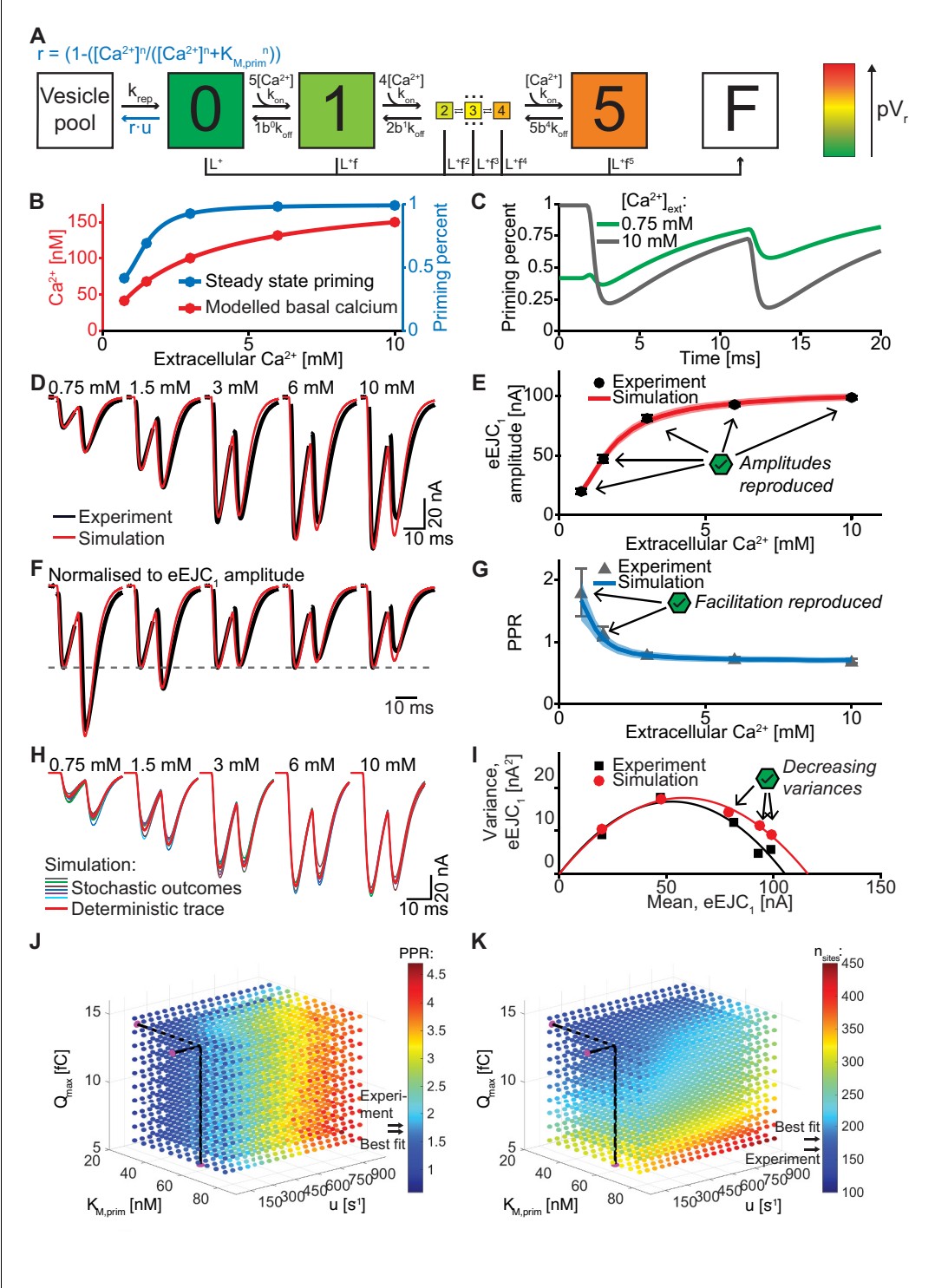

**Figure 7.** An unpriming model with Ca²⁺ dependent regulation of the RRP accounts for experimentally observed Ca²⁺ dependent eEJCs, STP and variances. (A) Diagram of the unpriming model. The rate of unpriming decreases with the Ca²⁺ concentration. All other reactions are identical to the single-sensor model (*Figure 4A*). (B) Assumed basal Ca²⁺ concentration at different extracellular Ca²⁺ concentrations (red curve) together with the steady-state amount of priming (blue). Increasing basal Ca²⁺ concentration increases priming. (C) The average fraction of occupied release sites as a function of time in simulations with 0.75 mM (green) and 10 mM (gray) extracellular Ca²⁺ concentration. Release reduced the number of primed SVs. At 0.75 mM Ca²⁺, the Ca²⁺-dependent reduction of unpriming leads to 'overfilling' of the RRP between AP₁ and AP₂, thereby inducing facilitation. (D) Average experimental eEJC traces (black) together with average simulated traces (red). (E) eEJC₁ amplitudes of experiment (black) and simulation (red). Error bars and colored bands show standard deviation. (F) Average, normalized eEJC traces of experiment (black) and simulation (red). (G) PPR values

*Figure 7 continued on next page*

*Figure 7 continued*

of experiment (gray) and simulation (blue). Error bars and colored bands show standard deviation. Simulations reproduce the experimentally observed facilitation. (H) Average simulated traces (red) and examples of different outcomes of the stochastic simulation (colors). (I) Plot of the mean synaptic variance vs. the mean $eEJC_1$ values, both from the experiment (black) and the simulations (red). The curves show the best fitted parabolas with forced intercept at (0,0) (simulation: $Var = -0.0053*<eEJC_1>^2 + 0.6090$ nA$*<eEJC_1>$, corresponding to $n_{sites} = 189$ and q = 0.61 nA when assuming a classical binomial model (*Clements and Silver, 2000*), see Materials and methods). (J) Similar to *Figure 6J*. Parameter exploration of the unpriming model varying $Q_{max}$, $k_{M,prim}$, and u (unpriming rate constant). Each ball represents a choice of parameters and the color indicates the PPR value. Black lines show the best fit parameters, and arrows show the experimental and best fit simulation values. (K) Same parameter choices as in (J). The colors indicate the optimal maximal number of SVs (i.e. number of release sites, $n_{sites}$) in order to fit the $eEJC_1$ amplitude at the five different $Ca^{2+}$ concentrations. A large span of PPR values (shown in (J)) can be fitted with a reasonable number of release sites (shown in (K)). Experimental data (example traces and means) depicted in panels D-G,I are replotted from *Figure 2A–D,F*. Parameters used for simulation can be found in *Tables 1–3*. Simulation scripts can be found in *Source code 1*. Results from simulations (means and SDs) can be found in the accompanying source data file (*Figure 7—source data 1*). Simulations of the unpriming model with cooperativity two are summarized in *Figure 7—figure supplement 1*. The site activation model (described later) is introduced and results are summarized in *Figure 7—figure supplement 3*. Simulations of the unpriming model with various inter-stimulus intervals are summarized in *Figure 7—figure supplement 2*.

The online version of this article includes the following source data and figure supplement(s) for figure 7:

**Source data 1.** Simulation data for graphs in *Figure 7E, G, I*; *Figure 7—figure supplement 1B, D, E*; *Figure 7—figure supplement 2D, F, H* and *Figure 7—figure supplement 3*.
**Figure supplement 1.** The unpriming model with cooperativity 2.
**Figure supplement 2.** A model with Ca2+-dependent release site activation accounts for experimentally observed eEJCs, STP and variances.
**Figure supplement 3.** Simulation based time course predictions of paired-pulse STF recovery for different interstimulus intervals across different Ca2+ concentrations (0.75–10 mM).

experimentally observed STF. Lowering the $Ca^{2+}$ influx (by decreasing $Q_{max}$) yielded a modest increase in PPR values (*Figure 6J*), but required a large number of release sites ($n_{sites}$) to match the $eEJC_1$ amplitudes (*Figure 6K*). Changing s had the largest effect when $k_2$ was close to the best fit value and moving away from this value decreased the PPRs, either by increasing the effect of the second sensor on $AP_1$ (when increasing $k_2$) or by decreasing the effect on $AP_2$ (when decreasing $k_2$), which both counteracts STF (*Figure 6B,J*).

Fitting the dual fusion-sensor model with a $Ca^{2+}$ cooperativity of 5 did not improve the situation (*Figure 6—figure supplement 1*, best fit parameters in *Table 2*): Although slightly more facilitation was observed, this model suffered from even larger variance overshoots (*Figure 6—figure supplement 1E*) and excessive asynchronous release (*Figure 6—figure supplement 1A,C*). We explored different $K_D$ values between 0.5 and 2 µM at cooperativities 2–5 in separate optimizations, but found no satisfactory fit of the data (results not shown). Thus, a dual fusion-sensor model is unlikely to account for STF observed from the realistic SV release site topology at the *Drosophila* NMJ. Note that this finding does not rule out that syt-7 functions in STF, but argues against a role in cooperating alongside syt-1 in a $pV_r$-based facilitation mechanism.

## Rapidly regulating the number of RRP vesicles accounts for $eEJC_1$ amplitudes, STF, temporal transmission profiles and variances

Since dual fusion-sensor models and other models depending on changes in $pV_r$ (see Discussion) are unlikely to be sufficient, we next investigated mechanisms involving an activity-dependent regulation of the number of participating release sites. For this we extended the single-sensor model by a single unpriming reaction (compare *Figures 4A* and *7A*). The consequence of reversible priming is that the initial release site occupation can be less than 100% (in which cases $n_{sites}$ can exceed the number of RRP vesicles). This enables an increase ('overfilling') of the RRP (/increase in site occupancy) during the inter-stimulus interval (consistent with reports in other systems *Dinkelacker et al., 2000*; *Gustafsson et al., 2019*; *Pulido et al., 2015*; *Smith et al., 1998*; *Trigo et al., 2012*). We assumed that $Ca^{2+}$ would stabilize the RRP/release site occupation by slowing down unpriming (*Figure 7A*). This made the steady-state RRP size dependent on the resting $Ca^{2+}$ concentration and the modest dependence of this on the extracellular $Ca^{2+}$ resulted in RRP enlargement with increasing extracellular $Ca^{2+}$ (*Figure 7B*), in agreement with recent findings on central synapses (*Malagon et al., 2020*). This model (like the dual fusion-sensor models depicted in *Figure 6* and *Figure 6—figure supplement 1*) includes two different $Ca^{2+}$ sensors, but the major difference is that these $Ca^{2+}$ sensors operate to regulate two separate sequential steps (priming and fusion). Indeed, this scenario aligns

**Table 2.** Best fit parameters of all models.

**Models presented in main figures**

| | Single-sensor model (*Figure 4*) | Dual fusion-sensor model, cooperativity 2 (*Figure 6*) | Unpriming model, cooperativity 5 (*Figure 7*) |
|---|---|---|---|
| $Q_{max}$ | 8.42 fC | 4.51 fC | 13.77 fC |
| $k_{rep}$ | 165.53 s$^{-1}$ | 159.30 s$^{-1}$ | 134.85 s$^{-1}$ |
| $n_{sites}$ | 216 | 211 | 180 |
| $k_2$ | | 4.10e7 M$^{-1}$s$^{-1}$ | |
| $s$ | | 510.26 | |
| $u$ | | | 236.82 s$^{-1}$ |
| $k_{M,prim}$ | | | 55.21 nM$^{-1}$ |
| Cost value (see Materials and methods) | 9.689 | 4.129 | 0.340 |

**Models presented in figure supplements**

| | Dual fusion-sensor model, cooperativity 5 (*Figure 6—figure supplement 1*) | Unpriming model, cooperativity 2 (*Figure 7—figure supplement 1*) | Site activation model (*Figure 7—figure supplement 3*) |
|---|---|---|---|
| $Q_{max}$ | 8.10 fC | 13.49 fC | 12.59 fC |
| $k_{rep}$ | 492.56 s$^{-1}$ | 106.59 s$^{-1}$ | 141.20 s$^{-1}$ |
| $n_{sites}$ | 112 | 203 | 189 |
| $k_2$ | 5.41e6 M$^{-1}$s$^{-1}$ | | |
| $s$ | 261.07 | | |
| $u$ | | 5207.70 s$^{-1}$ | |
| $k_{M,prim}$ | | 7.61 nM$^{-1}$ | |
| $\beta$ | | | 0.09 s$^{-1}$ |
| $\gamma$ | | | 194.77 s$^{-1}$ |
| $\delta$ | | | 10.70 s$^{-1}$ |
| Cost value (see Materials and methods) | 2.941 | 0.642 | 1.57 |

with reports of a syt-7 function upstream of SV fusion (*Liu et al., 2014*; *Schonn et al., 2008*). *Figure 7C* shows how the number of RRP vesicles develops over time in this model during a paired-pulse experiment for low and high extracellular Ca$^{2+}$ concentrations. In all cases, SV priming was in equilibrium prior to the first stimulus, indicated by the horizontal lines (0–2 ms, *Figure 7C*). Note that prior to AP$_1$ priming is submaximal (~41%) for 0.75 mM extracellular Ca$^{2+}$, but near complete (~99%) at 10 mM extracellular Ca$^{2+}$. At low extracellular Ca$^{2+}$ the elevation of Ca$^{2+}$ caused by AP$_1$ results in a sizable inhibition of unpriming, leading to an increase ('overfilling') of the RRP during the inter-stimulus interval. With this, more primed SVs are available for AP$_2$, causing facilitation (green line in *Figure 7C*). In contrast, at high extracellular Ca$^{2+}$ concentrations, the rate of unpriming is already low at steady state and the RRP close to maximal capacity (grey line in *Figure 7C*). At this high extracellular Ca$^{2+}$ concentration, AP$_1$ induces a larger Ca$^{2+}$ current (higher pV$_r$), resulting in strong RRP depletion, of which only a fraction recovers between APs (as in the other models, replenishment commences with a Ca$^{2+}$ independent rate $k_{rep}$). Because Ca$^{2+}$ acts in RRP stabilization, not in stimulating forward priming, this model (unlike the dual fusion-sensor models in *Figure 6* and *Figure 6—figure supplement 1*) did not yield asynchronous release in-between APs (*Figure 7D*). Thus, the two most important features of this model are the submaximal site occupation and an inhibition of unpriming by intracellular Ca$^{2+}$.

In this model we assumed a Ca$^{2+}$ cooperativity of n = 5 for the unpriming mechanism (we also explored n = 2, see *Figure 7—figure supplement 1*). The following parameters were optimized: $Q_{max}$, $n_{sites}$ and $k_{rep}$ (like in the single- and dual fusion-sensor models), together with $K_{M,prim}$, the Ca$^{2+}$ affinity of the priming sensor, and $u$, its Ca$^{2+}$ cooperativity. These values together define the

**Table 3.** Parameters of exocytosis simulation.

| Parameter | Explanation and reference | Value |
|---|---|---|
| *Common parameters* | | |
| $n_{sites}$ | Number of release sites (=maximal number of SVs) | Fitted (all models), see *Table 2* |
| $L^+$ | Basal fusion rate constant (*Kochubey and Schneggenburger, 2011*) | $3.5 \cdot 10^{-4}$ s$^{-1}$ |
| q | Amplitude of the mEJC. Estimated from variance-mean of data (see *Figure 2F*) | 0.6 nA |
| *Fast sensor (all models)* | | |
| $n_{max}$ | Cooperativity, fast sensor (*Lou et al., 2005*; *Schneggenburger and Neher, 2000*; *Wölfel et al., 2007*) | 5 |
| $k_1$ | Ca$^{2+}$ binding rate, first sensor (*Wölfel et al., 2007*) | $1.4 \cdot 10^8$ M$^{-1}$s$^{-1}$ |
| $k_{-1}$ | Ca$^{2+}$ unbinding rate, first sensor (*Wölfel et al., 2007*) | 4000 s$^{-1}$ |
| $b_f$ | Cooperativity factor, first sensor (*Lou et al., 2005*; *Wölfel et al., 2007*) | 0.5 |
| $k_f$ | Fusion rate constant of R(5,0) (fast sensor fully activated). (*Lou et al., 2005*; *Schneggenburger and Neher, 2000*; *Wölfel et al., 2007*) | 6000 s$^{-1}$ |
| f | $\left(\frac{k_f}{L^+}\right)^{\frac{1}{5}}$ | 27.978 |
| *Replenishment (all models)* | | |
| $k_{rep}$ | Replenishment rate constant | Fitted (all models), see *Table 2* |
| *Slow sensor (dual fusion-sensor model)* | | |
| $m_{max}$ | Cooperativity, second fusion sensor | 2 (5 in figure supplement) |
| $K_D$ | Dissociation constant, second fusion sensor (*Brandt et al., 2012*) | 1.5 µM |
| $k_2$ | Ca$^{2+}$ binding rate, second fusion sensor | Fitted (dual fusion-sensor model), see *Table 2* |
| $k_{-2}$ | Ca$^{2+}$ unbinding rate, second fusion sensor | $k_D \cdot k_2$ |
| $b_s$ | Cooperativity factor, second fusion sensor (=$b_f$) | 0.5 |
| s | Second fusion sensor analogue of f: factor on the fusion rate | Fitted (dual fusion-sensor model), see *Table 2* |
| *Unpriming (unpriming model)* | | |
| n | Cooperativity (exponent in unpriming rate equation) | 5 (2 in figure supplement) |
| u | Rate constant of unpriming | Fitted (unpriming model), see *Table 2* |
| $K_{M,prim}$ | Michaelis-Menten constant in expression of r | Fitted (unpriming model), see *Table 2* |
| *Site activation (site activation model)* | | |
| n | Cooperativity (exponent on [Ca$^{2+}$]) | 5 |
| α | Rate constant [I] to [D] | 1e6 s$^{-1}$ |
| β | Rate constant [D] to [I] | Fitted (site activation model), see *Table 2* |
| γ | Rate constant [D] to [A] | Fitted (site activation model), see *Table 2* |
| δ | Rate constant [A] to [D] | Fitted (site activation model), see *Table 2* |

Ca$^{2+}$-dependent unpriming rate (see *Table 2* for best fit parameters). The total number of fitted parameters (5) was the same as for the dual fusion-sensor models (*Figure 6* and *Figure 6—figure supplement 1*). *Figure 7D–I* present the results. It is clear that both eEJC$_1$ amplitudes and PPR values were described very well with this model at all extracellular Ca$^{2+}$ concentrations. In addition, the variance-mean relationship of the eEJC$_1$ was reproduced satisfactorily, except for a small variance

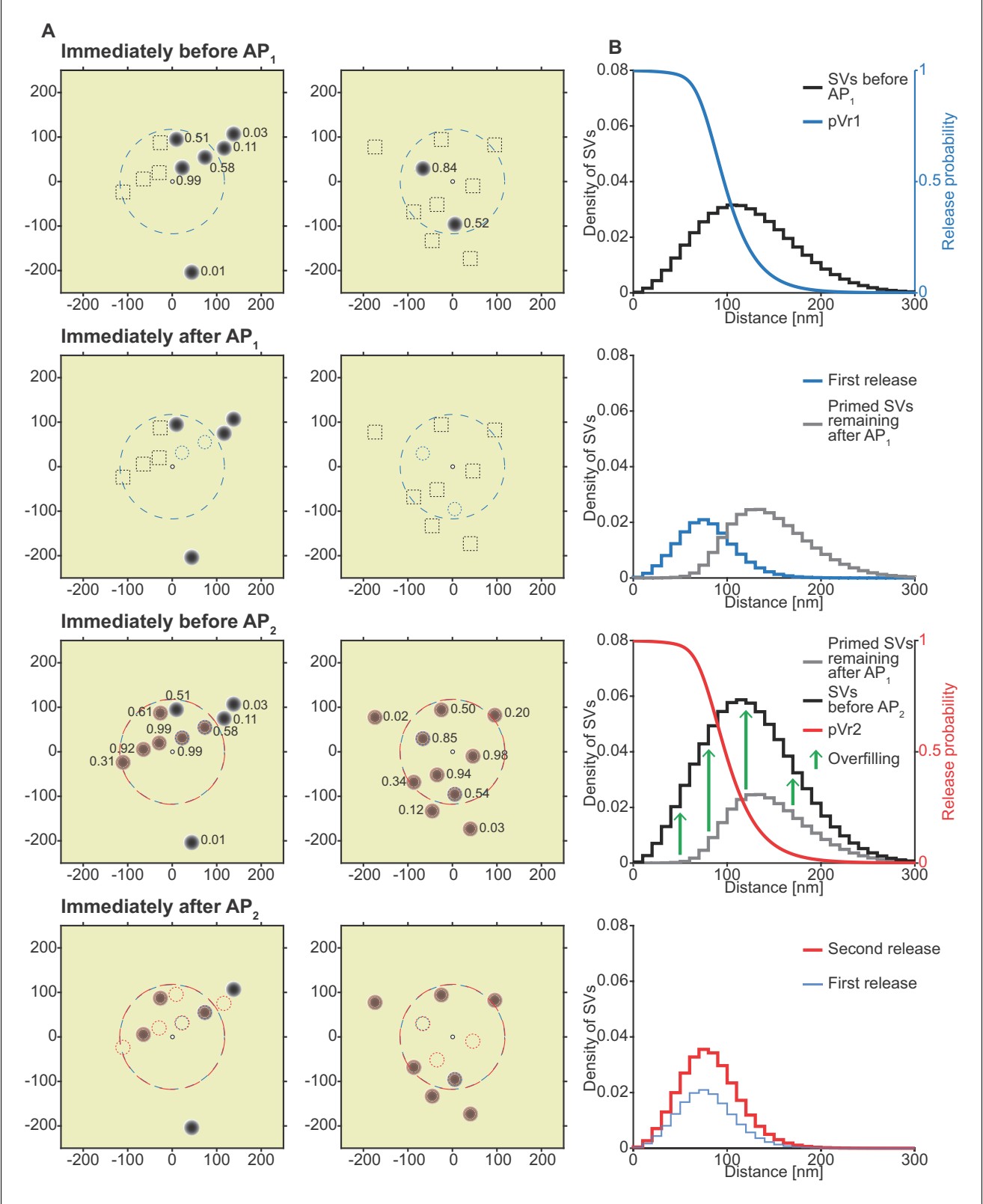

**Figure 8.** The unpriming model counteracts short-term depression by increasing the number of responsive SVs between stimuli and predicts a more efficient use of SVs throughout the synapse. (**A**) Two examples of docked SVs stochastically placed according to the distribution described in *Figure 1D* and their behavior in the PPR simulation at 0.75 mM extracellular $Ca^{2+}$ concentration. For clarity, 10 SVs are shown per AZ and only a central part of the AZ is shown. Top row: Prior to $AP_1$, only some release sites contain a primed SV (dark gray circles) and $pV_r1$ is indicated as a number. Initially empty

*Figure 8 continued on next page*

*Figure 8 continued*

release sites are indicated by dashed black squares. The larger dashed, blue circle in the AZ center indicates $pV_r1 = 0.25$. Second row: After $AP_1$ some of the SVs have fused (dashed blue circles). Third row: Right before $AP_2$ the initially empty sites as well as the sites with SV fusion in response to $AP_1$ have been (re)populated (orange shading). $pV_r2$ is indicated as a number. The larger dashed, red circle indicates $pV_r2 = 0.25$. Bottom row: After $AP_2$ the second release has taken place. Small, dashed circles indicate release from $AP_1$ and $AP_2$ (blue and red resp.). (B) The average simulation at the same time points as in (A). Histograms represent primed SVs (black and gray) as well as first and second release (blue and red) illustrating how release from $AP_1$ and $AP_2$ draw on a larger part of the SV distribution (compare to *Figure 5*) and how the increase in RRP size can induce facilitation. The blue and red curves indicate the vesicular release probability as a function of distance during $AP_1$ (blue) and $AP_2$ (red). Parameters used for simulations can be found in *Tables 1–3*.

overshoot for the highest extracellular $Ca^{2+}$ concentrations (*Figure 7I*, see Discussion). Fitting of the unpriming model with a $Ca^{2+}$ cooperativity of 2 also led to a good fit (*Figure 7—figure supplement 1*), although the variance overshoot was somewhat larger. We also explored the time-dependence of the facilitation by simulating PPR values for various inter-stimulus intervals at different extracellular $Ca^{2+}$ concentrations which could be investigated experimentally in the future to further refine parameters (*Figure 7—figure supplement 2*).

Different facilitating synapses exhibit a large range of PPR values, some larger than observed at the *Drosophila* NMJ (*Jackman et al., 2016*). Therefore, if this were a general mechanism to produce facilitation, we would expect it to be flexible enough to increase the PPR much more than observed here. To investigate the model's flexibility we systematically explored the parameter space by varying $Q_{max}$, $K_{M,prim}$, and $u$ (*Figure 7J,K*). Similar to *Figure 6J,K*, the colors of the balls represent the PPR value and the number of release sites needed to fit the $eEJC_1$ amplitudes. Consistent with a very large dynamic range of this mechanism, PPR values ranged from 0.85 to 3.90 (*Figure 7J,K*) and unlike the dual fusion-sensor model, PPR values were fairly robust to changes in $Ca^{2+}$ influx (note the different scales on *Figure 7J,K* and *Figure 6J,K*). Moreover, because this mechanism does not affect the $Ca^{2+}$ sensitivity of SV fusion, facilitation was achieved without inducing asynchronous release (*Figure 7D*).

We also investigated an alternative model based on $Ca^{2+}$-dependent release site activation. In this model, all sites are occupied by a vesicle, but some sites are inactive and fusion is only possible from activated sites. We assumed that site activation was $Ca^{2+}$-dependent. In order to avoid site activation during $AP_1$, which would again hinder STF and could contribute to asynchronous release, we implemented an intermediate delay state (*Figure 7—figure supplement 3A–B*) from which sites were activated in a $Ca^{2+}$-independent reaction. This could mean that priming occurs in two-steps, with the first step being $Ca^{2+}$-dependent. Similar to the unpriming model presented above, the modest increase of intracellular $Ca^{2+}$ with extracellular $Ca^{2+}$ yielded an RRP increase (/increase in active sites) (*Figure 7—figure supplement 3I*). This model agreed similarly well with the data as the unpriming model (*Figure 7—figure supplement 3C–H*). Thus, both mechanisms which modulate the RRP rather than $pV_r$ are fully capable of reproducing the experimentally observed $Ca^{2+}$-dependent $eEJC_1$ amplitudes, STF, release synchrony and variance. The unpriming model was preferred since it had fewer parameters and performed slightly better in optimisations than the site activation model.

## A release site facilitation mechanism utilizes a larger part of the SV distribution

Why do $n_{site}$/priming-based mechanisms (*Figure 7*, *Figure 7—figure supplement 1*, *Figure 7—figure supplement 3*) account for STF from the broad distribution of SV release site:$Ca^{2+}$ channel coupling distances, while the $pV_r$-based models (*Figures 4* and *6*, *Figure 6—figure supplement 1*) cannot? To gain insight into this, we analysed the spatial dependence of transmitter release in the unpriming model during the paired-pulse experiment (0.75 mM extracellular $Ca^{2+}$) in greater detail (*Figure 8*). Panel 8A, similarly to *Figure 5A*, shows example stochastic simulations (at external $Ca^{2+}$ concentration 0.75 mM, to illustrate facilitation). The best fit parameters of the unpriming model predicted a larger $Ca^{2+}$ influx (1.64-fold and 3.05-fold larger $Q_{max}$ value) than the single- and dual fusion-sensor models (*Table 2*). The larger $Ca^{2+}$ influx compensated for the submaximal priming of SVs (reduced release site occupancy) prior to the first stimulus by expanding the region where SVs are fused (*Figure 8B*). Comparing to *Figure 5B*, a much larger part of the SV distribution is utilized during the first stimulus. Following $AP_1$, vesicles prime into empty sites across the entire distribution,

allowing $AP_2$ to draw again from the entire distribution. During this time, the increased residual $Ca^{2+}$ causes overfilling of the RRP, that is more release sites are now occupied, giving rise to more release during $AP_2$. Notably, the $AP_2$-induced release again draws from the entire distribution. Thus, the unpriming model not only reproduces STF and synaptic variance, but also utilizes docked SVs more efficiently from the entire distribution compared to the single- and dual fusion-sensor model.

## Discussion

We here described a broad distribution of SV release site:$Ca^{2+}$ channel coupling distances in the *Drosophila* NMJ and compared physiological measurements with stochastic simulations of four different release models (single-sensor, dual fusion-sensor, $Ca^{2+}$-dependent unpriming and site activation model). We showed that the two first models (single-sensor and dual fusion-sensor), where residual $Ca^{2+}$ acts on the energy barrier for fusion and results in an increase in $pV_r$, failed to reproduce facilitation. The two latter models involve a $Ca^{2+}$-dependent regulation of participating release sites and reproduced release amplitudes, variances and PPRs. Therefore, the $Ca^{2+}$-dependent accumulation of releasable SVs is a plausible mechanism for paired-pulse facilitation at the *Drosophila* NMJ, and possibly in central synapses as well. In more detail, our insights are as follows:

1. The SV distribution was described by the single-peaked integrated Rayleigh distribution with a fitted mean of 122 nm. The distribution has a low probability for positioning of SVs very close to $Ca^{2+}$ channels (less than 1.5% within 30 nm) and is therefore reasonably consistent with suggestions of a SV exclusion zone of ~ 30 nm around $Ca^{2+}$ channels (*Keller et al., 2015*). Strikingly, almost exactly the same distribution was identified for the essential priming protein Unc13A (*Figure 1F*, *Figure 1—figure supplement 1D*), indicating that docked SVs are likely primed (*Imig et al., 2014*).

2. The broad distribution of SV release site:$Ca^{2+}$ channel distances particularly impedes $pV_r$-based facilitation mechanisms. Indeed, previous models that reproduced facilitation using $pV_r$-mechanisms typically placed SVs at an identical/similar distance to $Ca^{2+}$ channels, resulting in intermediate (and identical) $pV_r$ for all SVs (*Böhme et al., 2016*; *Böhme et al., 2018*; *Bollmann and Sakmann, 2005*; *Fogelson and Zucker, 1985*; *Jackman and Regehr, 2017*; *Matveev et al., 2006*; *Matveev et al., 2004*; *Tang et al., 2000*; *Vyleta and Jonas, 2014*; *Yamada and Zucker, 1992*). Here, having mapped the precise AZ topology, we show that the broad SV distribution together with the steep dependence of release rates on $[Ca^{2+}]$ creates a situation where $pV_r$ falls to almost zero for SVs further away than the mean of the distribution (*Figure 5*). As a result, most SVs either fuse during $AP_1$, or have $pV_r$ values close to zero, leaving little room for modulation of $pV_r$ to create facilitation. Such mechanisms (including buffer saturation, and $Ca^{2+}$ binding to a second fusion sensor) will act to multiply release rates with a number $> 1$. However, since SVs with $pV_r$ close to one have already fused during $AP_1$, and most of the remaining vesicles have $pV_r$ close to zero such a mechanism will be ineffective in creating facilitation. Thus, the broad distribution of SV release site:$Ca^{2+}$ channel distances makes it unlikely that $pV_r$-based mechanisms can cause facilitation.

3. The dual fusion-sensor model was explored as an example of a $pV_r$-based model. Two problems were encountered: The first problem was that the second sensor, due to its high affinity for $Ca^{2+}$, was partly activated in the steady state prior to the stimulus (*Figure 6B*). Therefore, it could not increase $pV_r2$ without also increasing $pV_r1$. This makes it inefficient in boosting the PPR. The second problem was kinetic: the second sensor should be fast enough to activate between two APs, but slow enough not to activate during $AP_1$. This is illustrated in *Figure 6B–C*, which shows the time course of activation of the two sensors and the corresponding PPR values for varying $Ca^{2+}$ binding rates of the second sensor. Since the sensor is $Ca^{2+}$-dependent, the rate inevitably increases during the $Ca^{2+}$ transient, leading to too much asynchronous release. In principle, the first problem could be alleviated by increasing the $Ca^{2+}$ cooperativity of the second sensor, which would make it easier to find parameters where the sensor would activate after but not before $AP_1$. We therefore tried to optimize the model with cooperativities of 3, 4, and 5 (*Figure 6—figure supplement 1* shows cooperativity 5), and indeed, the higher cooperativity made it possible to obtain slightly more facilitation. However, activation during the AP (the second problem) was exacerbated and caused massive and unphysiological asynchronous release. Thus, a secondary $Ca^{2+}$ sensor acting on the energy barrier for fusion is unlikely to account for facilitation in synapses with a broad distribution of SV release site:$Ca^{2+}$ channel distances.

4. We included stochasticity at the level of the SV distribution (release sites were randomly drawn from the distribution) and at the level of SV $Ca^{2+}$ (un)binding and fusion. This was essential since deterministic and stochastic simulations do not agree on PPR-values due to Jensen's inequality (for a stochastic process the mean of a ratio is not the same as the ratio of the means) (see Materials and methods and *Figure 4—figure supplement 1*). The effect is largest when the evoked release amplitude is smallest. Since small amplitudes are often associated with high facilitation, this effect is important and needs to be taken into account. Stochastic $Ca^{2+}$ channel gating on the other hand was not included, as this would increase simulation time dramatically. At the NMJ, the $Ca^{2+}$ channels are clustered (*Gratz et al., 2019*; *Kawasaki et al., 2004*), and most SVs are relatively far away from the cluster, a situation that was described to make the contribution of $Ca^{2+}$ channel gating to stochasticity small (*Meinrenken et al., 2002*). However, the situation will be different in synapses where individual SVs co-localize with individual $Ca^{2+}$ channels (*Stanley, 2016*).

5. Stochastic simulations made it possible to not only determine the mean $eEJC_1$ and PPR values, but also the standard deviation around these values upon repeated activation of the NMJ (indicated as lightly colored bands on the simulations in *Figure 4C,E*, *Figure 6E,G*, and *Figure 7E,G*), which can be compared to measurements (shown as black error bars in the same figure panels). This also enabled us to compare our model to experimental variance-mean data (*Figures 4G*, *6I* and *7I*), which we found was key to identify valid models. All models tested resulted in variance-mean dependences that were well approximated by a parabola with intercept 0. Note that such parabola agrees with the mean-variance relationship in a binomial distribution. However these simulations show that the assumption of heterogeneous release probability (and changing RRP size) can also lead to the experimentally observed parabolic variance-mean relationship. The single-sensor and dual fusion-sensor models resulted in overshooting variances (*Figures 4G* and *6I*), which became even worse in the case of higher cooperativity of the second fusion sensor (*Figure 6—figure supplement 1*). The right-hand intercept of the variance-mean relationship with the abscissa is interpreted as the product of the number of release sites ($n_{sites}$) and $q$ (the single SV quantum) and the tendency of these models to overshoot the variance is due to the fitting procedure increasing $n_{sites}$, while at the same time reducing $pV_r$, (by reducing $Q_{max}$, the maximal AP-induced $Ca^{2+}$ influx). The lower $pV_r$ increases the PPR by reducing the effect of depletion, but results in unrealistically high $n_{sites}$. Therefore, it was essential to contrast the models to experimental variance-mean data, which restrict $n_{sites}$. This revealed that $pV_r$-based facilitation mechanisms produced unrealistic variance-mean behavior. In this context, models involving a $Ca^{2+}$-dependent accumulation of releasable SVs fare much better, because only those can cause facilitation in the presence of realistic $n_{sites}$, resulting in very similar variance-mean behaviour to the experiment (*Figure 7I*, *Figure 7—figure supplement 1E*). The remaining slight overshoot for variances at high extracellular $Ca^{2+}$ concentrations could have technical/experimental reasons, because these experiments are of long duration, which might lead to run-down over time (which is not present in the model simulations) that causes a compression of the parabolic relationship along the abscissa (experiments were performed by increasing $Ca^{2+}$ concentrations).

6. We arrived at two models that can explain paired-pulse facilitation and variance-mean behaviour at the *Drosophila* NMJ. Both models include a $Ca^{2+}$-dependent increase in the number of participating (occupied/activated) release sites. In the $Ca^{2+}$-dependent unpriming model, forward priming happens at a constant rate, but unpriming is inversely $Ca^{2+}$-dependent, such that increases in residual $Ca^{2+}$ lead to inhibition of unpriming, thereby increasing release site occupation between stimuli (*Figure 7*). $Ca^{2+}$-dependent replenishment has been observed in multiple systems (*Dinkelacker et al., 2000*; *Smith et al., 1998*; *Stevens and Wesseling, 1998*; *Wang and Kaczmarek, 1998*). This has traditionally been implemented in various release models as a $Ca^{2+}$-dependent forward priming rate (*Man et al., 2015*; *Pan and Zucker, 2009*; *Voets, 2000*; *Weis et al., 1999*). In a previous secretion model in chromaffin cells, we had proposed a catalytic function of $Ca^{2+}$ upstream of vesicle fusion (*Walter et al., 2013*). However, in the context of STF such models would favour accelerated priming during the AP, which would counteract this facilitation mechanism and might cause asynchronous release, similar to the problem with the dual fusion-sensor model (*Figure 6*). In the model presented here this is prevented by including the $Ca^{2+}$ dependency on the unpriming rate. Consistent with this idea, recent data in cells and in biochemical experiments showed that the $Ca^{2+}$-dependent priming protein (M)Unc13 reduces unpriming (*He et al., 2017*; *Prinslow et al., 2019*). Another model that reproduced the electrophysiological data was the site activation model, where sites are activated $Ca^{2+}$-dependently under docked (but initially unprimed) SVs (*Figure 7—figure supplement 3*). In this case, we had to prevent rapid activation-and-fusion

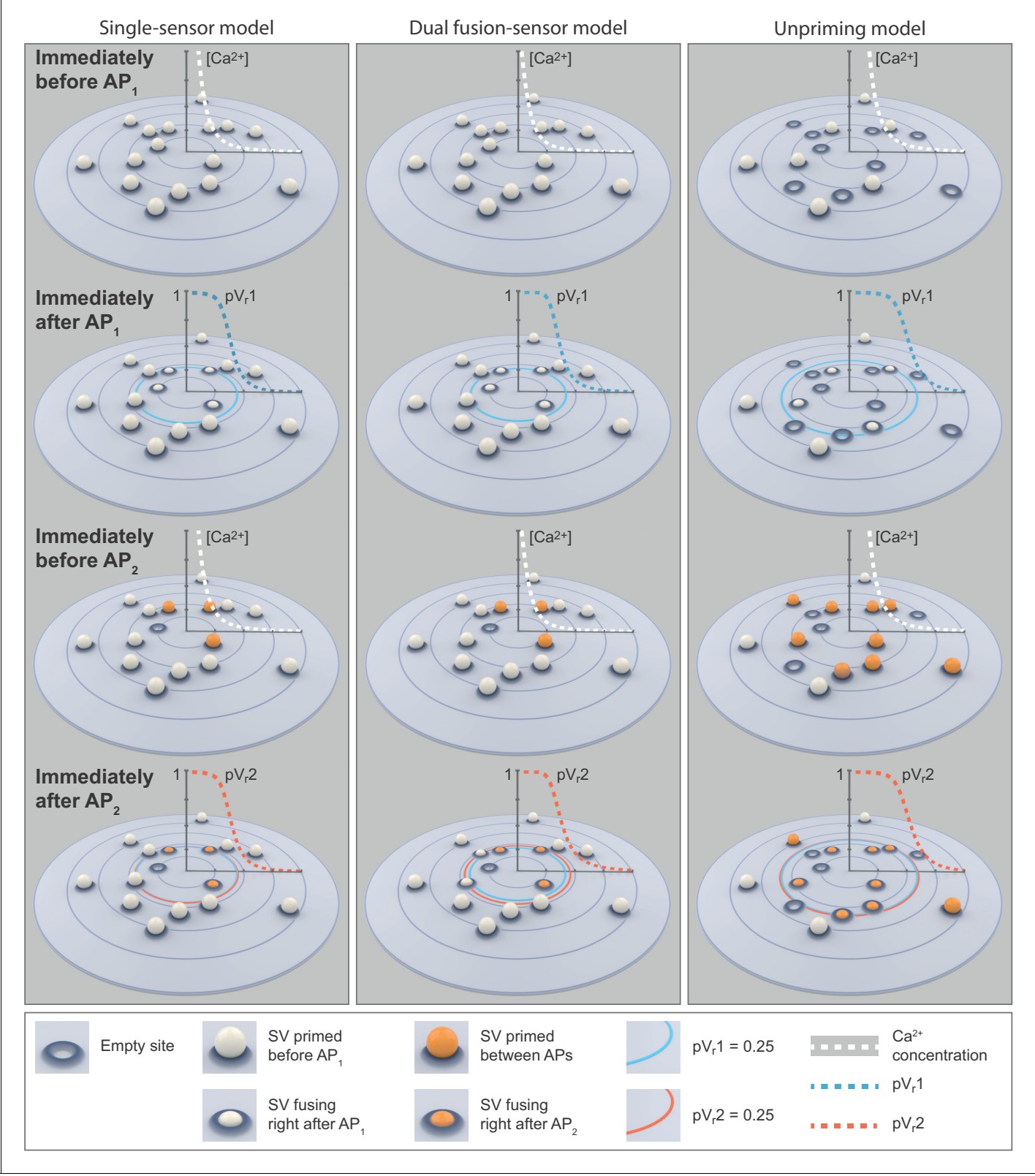

**Figure 9.** Cartoon illustrations of the single-sensor, the dual fusion-sensor, and the unpriming models during a paired-pulse simulation at 0.75 mM extracellular $Ca^{2+}$. Top row: SVs primed (white ball) prior to $AP_1$. In the single- and dual fusion-sensor models all release sites are occupied. In the unpriming model priming is in an equilibrium with unpriming and some release sites are empty. The dashed white graphs show the peak $Ca^{2+}$ concentration (simulation of optimal fits for each model) during the first transient as a function of distance to the $Ca^{2+}$ source. Second row: Some of the

*Figure 9 continued on next page*

*Figure 9 continued*

SVs fuse in response to AP$_1$. The dashed blue graphs show the pV$_r$1 as a function of distance. The large blue circles indicate pV$_r$1 = 0.25. In the unpriming model the larger Ca$^{2+}$ influx (according to the optimal fit) increases the area from which SVs fuse. Third row: Right before AP$_2$ some of the empty release sites have been repopulated or newly filled by priming (orange balls). The shift in the (un)priming equilibrium in the unpriming model makes the increase in the number of primed SVs substantially larger than in the other models. The dashed white graphs show the peak Ca$^{2+}$ concentration during the second transient as a function of distance to the Ca$^{2+}$ source. Bottom row: SV fusion in response to AP$_2$. The large dashed red graphs show pV$_r$2 as a function of distance to the Ca$^{2+}$ source. The blue and red circles indicate pV$_r$1 and pV$_r$2 of 0.25. In the dual fusion-sensor model, the second sensor increases pV$_r$ between stimuli, but the effect is small, even in the best fit of the model. These cartoons illustrate the mechanisms underlying our fitting results of the different models: The dual fusion-sensor model shows a small increase in second release compared to the single-sensor model, but only the unpriming model reproduces the experimentally observed facilitation. Parameters used for simulations can be found in *Tables 1–3*.

during the AP by including an extra, Ca$^{2+}$-independent transition, which introduces a delay before sites are activated (*Figure 7—figure supplement 3*). The two models are conceptually similar in that they either recruit new SVs to (always active) sites, or activate sites underneath dormant SVs. Those two possibilities are almost equivalent when measuring with electrophysiology, but they might be distinguished in the future using flash-and-freeze electron microscopy (*Chang et al., 2018*; *Watanabe et al., 2013*). Interestingly, Unc13 has recently been shown to form release sites at the *Drosophila* NMJ (*Reddy-Alla et al., 2017*). Therefore, the two models also correspond to two alternative interpretations of Unc13 action (to prevent unpriming, or form release sites).

In our model, all primed vesicles have identical properties, and only deviate in their distance to the Ca$^{2+}$ channel cluster (positional priming, *Neher and Brose, 2018*). Alternatively, several vesicle pools with different properties (molecular priming) could be considered, which might involve either vesicles with alternative priming machineries, or vesicles being in different transient states along the same (slow) priming pathway (*Walter et al., 2013*). In principle, if different primed SV states are distributed heterogenously such that more distant vesicles are more primed/releasable, such an arrangement might counteract the effects of a broad distance distribution, although this is speculative. Without such a peripheral distribution, the existence of vesicles in a highly primed/releasable state (such as the 'super-primed' vesicles reported at the Calyx of Held synapse), would result in pronounced STD, and counteract STF, which indeed has been observed (*Lee et al., 2013*; *Taschenberger et al., 2016*).

In this study electrophysiological recordings were performed on muscle 6 of the *Drosophila* larva which receives input from morphologically distinct NMJs containing big (Ib) and small (Is) synaptic boutons, which have been shown to differ in their physiological properties (*Atwood et al., 1993*; *He et al., 2009*; *Newman et al., 2017*). This could add another layer of functional heterogeneity in the postsynaptic responses analysed here (the EM and STED analyses shown here were focused on Ib inputs). Because our model does not distinguish between Is and Ib inputs, the estimated parameters represent a compound behaviour of all types of synaptic input to this muscle. Future investigations to isolate the contribution of the different input types (e.g. by genetically targeting Is/Ib-specific motoneurons using recently described GAL4 lines; *Pérez-Moreno and O'Kane, 2019*) could help distinguish between inputs and possibly further refine the model to identify parameter differences between these input types.

*Figure 9* summarizes the results for the single-sensor, dual fusion-sensor and unpriming models. Facilitation in single and dual fusion-sensor models depend on the increase in release probability from the first AP to the next (compare colored rings representing 25% release probability between row 2 and 4). However, the increase is very small, even for the dual fusion-sensor model, and to nevertheless produce some facilitation, optimisation finds a small Ca$^{2+}$ influx, which leads to an ineffective use of the broad vesicle distribution (and a too-high estimate of $n_{sites}$). In the unpriming model a higher fitted Ca$^{2+}$ influx ($Q_{Max}$) leads to a more effective use of the entire SV distribution, and facilitation results from the combination of incomplete occupancy of release sites before the first AP (row 1), combined with 'overshooting' priming into empty sites between APs (row 3).

Molecularly, syt-7 was linked to STF behaviour (*Jackman et al., 2016*), and our data does not rule out that syt-7 is essential for STF at the *Drosophila* NMJ. However, we show clearly that a pV$_r$-based facilitation mechanism (dual fusion-sensor model) cannot account for STF in synapses with

heterogeneous distances between release sites and Ca$^{2+}$ channels. Interestingly, syt-7 was also reported to function in vesicle priming and RRP replenishment (*Liu et al., 2014*; *Schonn et al., 2008*). Thus, future work will be necessary to investigate whether the function of syt-7 in STF might take place by Ca$^{2+}$-dependent inhibition of vesicle unpriming or release site activation.

Similar suggestions that facilitation results from a build-up of primed SVs during stimulus trains were made for the crayfish NMJ and mammalian synapses (*Gustafsson et al., 2019*; *Pan and Zucker, 2009*; *Pulido and Marty, 2018*). This is in line with our results, with facilitation arising from modulation of the number of primed SVs rather than pV$_r$. Our models are conceptually simple (e.g. all SVs are equally primed and distinguished only by distance to Ca$^{2+}$ channels, sometimes referred to as 'positional priming' *Neher and Sakaba, 2008*), and we improved conceptually on previous work by using estimated SV release site:Ca$^{2+}$ channel distributions, stochastic simulations and comparison to variance-mean relationships and we performed a systematic comparison of pV$_r$- and priming-based models. It has not been clear whether increases in primed SVs are also required for paired-pulse facilitation, or only become relevant in the case of 'tonic' synapses that build up release during longer stimulus trains (frequency facilitation *Neher and Brose, 2018*). Paired-pulse facilitation is a more wide-spread phenomenon in synapses than frequency facilitation, and we show here for the case of *Drosophila* NMJ that it also seems to require priming-based mechanisms. Thus, Ca$^{2+}$-dependent increases of the RRP during STP might be a general feature of chemical synapses.

# Materials and methods

**Key resources table**

| Reagent type (species) or resource | Designation | Source or reference | Identifiers | Additional information |
|---|---|---|---|---|
| Strain (*Drosophila melanogaster*) | w[1118] | Bloomington *Drosophila* Stock Center | | |
| Genetic reagent (*D. melanogaster*) | Ok6-GAL4/II | (*Aberle et al., 2002*) | PMID:11856529 | Ok6-Gal4/II crossed to w[1118] |
| Genetic reagent (*D. melanogaster*) | elav-Gal4/I | (*Lin and Goodman, 1994*) | PMID:7917288 | Used for elav-GAL4/+; ;UAS-Unc13A-GFP/+; P84200/P84200 |
| Genetic reagent (*D. melanogaster*) | UAS-Unc13A-GFP/III | (*Böhme et al., 2016*) | PMID:27526206 | Used for elav-GAL4/+;; UAS-Unc13A-GFP/+; P84200/P84200 |
| Genetic reagent (*D. melanogaster*) | ry$^{506}$; P{ry11} unc-13$^{P84200}$ / ci$^D$ | Kyoto Stock Center | FlyBase: FBst0300878 | Used for elav-GAL4/+;; UAS-Unc13A-GFP/+; P84200/P84200 |
| Genetic reagent (*D. melanogaster*) | w[1118]; P{w[+mC]=Mhc-SynapGCaMP6f}3–5 | (*Newman et al., 2017*) Bloomington *Drosophila* Stock Center | PMID:28285823 Bloomington Stock # 67739 | |
| Genetic reagent (*D. melanogaster*) | w[1118]; P{y[+t7.7] w[+mC]=20XUAS-IVS-665GCaMP6m} attP40/Ok6-GAL4 | Bloomington *Drosophila* Stock Center | Bloomington Stock # 42748 | |
| Antibody | Anti-Unc13A (guinea pig polyclonal) | (*Böhme et al., 2016*) | PMID:27526206 | Dilution: 1:500 |
| Antibody | Anti guinea pig STAR635 (goat polyclonal) | (*Böhme et al., 2016*) | PMID:27526206 | Dilution: 1:100 |
| Antibody | Anti Nc82 (mouse monoclonal) | Developmental Studies Hybridoma Bank | Antibody Registry ID: AB_2314866 | Dilution: 1:1000 |
| Antibody | Anti-mouse Cy5 (goat polyclonal) | Jackson Immuno Research | SKU: 115-175-072 | Dilution: 1:500 |
| Software, algorithm | LAS X software | Leica Microsystems | https://www.leica-microsystems.com | |
| Software, algorithm | LCS AF | Leica Microsystems | Leica Microsystems | |

*Continued on next page*

*Continued*

| Reagent type (species) or resource | Designation | Source or reference | Identifiers | Additional information |
|---|---|---|---|---|
| Software, algorithm | Image J | NIH | Version 1.48q/1.50 g; https://imagej.nih.gov/ij/ | |
| Software, algorithm | Imspector Software | Max Planck Innovation | Version 0.10 | |
| Software, algorithm | MATLAB | MathWorks | R2010b/R2016b | |
| Software, algorithm | Clampfit | Molecular Devices | Version 10.3 | |
| Software, algorithm | GraphPad Prism | GraphPad Software | Version 5.01/6.01 | |
| Software, algorithm | pClamp 10 | Molecular Devices | | |
| Software, algorithm | CalC | (*Matveev et al., 2002*) | PMID:12202362 Version 6.8.6 | |
| Other | Computer grid | Bioinformatics Center, University of Copenhagen | https://www1.bio.ku.dk/scarb/bioinformatics-centre/ | Used for simulations |
| Other | custom-built STED-microscope | (*Göttfert et al., 2017*) | PMID:23823248 | |
| Other | HPF machine (HPM100) | Leica Microsystems | https://www.leica-microsystems.com | |
| Other | AFS | Leica Microsystems | https://www.leica-microsystems.com | |
| Other | Ultramicrotome (RMC PowerTome XL; Reichert Ultracut S) | Leica Microsystems | https://www.leica-microsystems.com | |
| Other | Electrone microscope (TecnaiSpirit; FEI or Zeiss 900) | FEI; Zeiss | https://www.fei.com, https://www.zeiss.com | |

## Fly husbandry, genotypes and handling

Flies were kept under standard laboratory conditions as described previously (*Sigrist et al., 2003*) and reared on semi-defined medium (Bloomington recipe) at 25°C, except for GCaMP6m and synapGCaMP6f flies which were kept at room temperature, and *Ok6-GAL4/+* (*Figure 2*, *Figure 2—figure supplement 1*, *Figure 4* panel B-E and G, *Figure 6* panel D-G and I, *Figure 6—figure supplement 1*, *Figure 7D–G and I*, *Figure 7—figure supplement 1*, *Figure 7—figure supplement 3C–F,H*) which were kept at 29°C (for detailed genotypes see below). For experiments both male and female 3^rd instar larvae were used. The following genotypes were used:

*Figure 7 Ok6-GAL4/+* (*Ok6-Gal4/II* crossed to *w[1118]*; panel D-G, (I). *Figure 7—figure supplement 1*: *Ok6-GAL4/+* (*Ok6-Gal4/II* crossed to *w[1118]*). *Figure 7—figure supplement 3*: *Ok6-GAL4/+* (*Ok6-Gal4/II* crossed to *w[1118]*; panel C-F, (H).

The following stocks were used: *Ok6-GAL4/II* (*Aberle et al., 2002*), *UAS-Unc13A-GFP/III* (*Böhme et al., 2016*), *elav-Gal4/I* (*Lin and Goodman, 1994*). The following stock were obtained from the Bloomington *Drosophila* Stock Center: *P{w[+mC]=Mhc-SynapGCaMP6f}3–5/III* (*Newman et al., 2017*) and *w[1118]; P{y[+t7.7] w[+mC]=20XUAS-IVS-GCaMP6m}attP40*. The following stock was obtained from Kyoto Stock Center: *P84200/IV*.

## EM data acquisition and analysis

Sample preparation, EM image acquisition and the quantification of docked SV distances to the AZ center (center of the electron dense 'T-bar') are described in *Böhme et al. (2016)*; *Reddy-Alla et al. (2017)*. The Rayleigh distributions were fit to the distances of docked SVs to the T-bar pedestal center, which had been collected in two EM datasets; analyses of these datasets were published in two previous studies, (*Reddy-Alla et al., 2017*) for the histogram of distances depicted in *Figure 1A* and (*Böhme et al., 2016*) for the histogram of distances depicted in *Figure 1—figure supplement 1A*.

## Derivation of the realistic docked SV distribution from EM measurements

The distances between $Ca^{2+}$ channels and docked SVs in *Drosophila* NMJ obtained by EM was found to follow a Rayleigh distribution with best fit scale parameter σ = 76.51 nm (EM dataset 1) and σ = 74.07 nm (EM dataset 2). The fitting was performed with a MATLAB (MathWorks, version R2018b) function, *raylfit*, which uses maximum likelihood estimation. As these distances are found by EM of a cross-section of the active zone, we integrate this distribution around a circle to obtain the two-dimensional distribution of SVs in the circular space around the active zone.

The Rayleigh distribution has the following probability density function (pdf):

$$f(x) = \frac{x}{\sigma^2} e^{-x^2/2\sigma^2}, \ x > 0$$

The pdf of the SV distribution will then be a scaling of the following function

$$\hat{g}(x) = 2\pi x f(x) = 2\pi x \frac{x}{\sigma^2} e^{-x^2/2\sigma^2} \tag{1}$$

In order to find the pdf of the 2D SV distribution, we integrate $g$ to find the normalizing constant. By integration by parts we get

$$
\begin{aligned}
\int_0^\infty \hat{g}(x)\,dx &= \int_0^\infty 2\pi x \frac{1}{\sigma^2} x e^{-\frac{x^2}{2\sigma^2}}\,dx \\
&= 2\pi \left( \left[ -x e^{-\frac{x^2}{2\sigma^2}} \right]_0^\infty + \int_0^\infty e^{-\frac{x^2}{2\sigma^2}}\,dx \right) \\
&= 2\pi \int_0^\infty e^{-\frac{x^2}{2\sigma^2}}\,dx \\
&= 2\pi \tfrac{1}{2} \sigma \sqrt{2\pi}
\end{aligned}
$$

where the standard normal distribution was used in the last equality. Normalising (1) by this constant, we get the pdf of the distance distribution on a circular area in the active zone:

$$g(x) = \frac{\sqrt{2}}{\sqrt{\pi} \cdot \sigma^3} \cdot x^2 \cdot e^{-x^2/2\sigma^2}$$

## The SV distribution in simulations

In order to use the above SV distribution in simulations, we need to determine probabilities. $g(x)$ is a generalized gamma distribution with $a = \sqrt{2} \cdot \sigma$ a>0, p>0, d>0, $p = 2$ a>0, p>0, d>0, $d = 3$ a>0, p>0, d>0. The generalized gamma distribution with *a>0, p>0, d>0* has the following pdf:

$$h(x; a, d, p) = \frac{p}{a^d} \cdot \frac{x^{d-1} \cdot e^{-\left(\frac{x}{a}\right)^p}}{\Gamma(d/p)}$$

and cumulative density function (cdf):

$$H(x; a, d, p) = \frac{\gamma(d/p, (x/a)^p)}{\Gamma(d/p)}$$

where $\gamma$ is the lower incomplete gamma function, and $\Gamma$ is the (regular) gamma function. Both of these functions are implemented in MATLAB (MathWorks, version R2018b), which easily allows us to draw numbers from them.

Thus, the SV distribution has the following cdf:

$$G(x) = \frac{\gamma(1.5, (x^2/2\sigma^2))}{\Gamma(1.5)}$$

That is, given a uniformly distributed variable $q \in (0, 1)$, we can use inbuilt MATLAB functions to sample SV distances, *d*:

$$d = G^{-1}(q) = \sqrt{\gamma^{-1}(1.5, q \cdot \Gamma(1.5)) \cdot 2\sigma^2} \tag{2}$$

The implementation is as follows:

```
q = rand(1);
d = sqrt(2 * sigma^2 * gammaincinv(q, 1.5));
```

Note that in MATLAB the inverse incomplete gamma function with parameter *s* is scaled by $\Gamma(s)$, which is why we input $q$ and not $q/\Gamma(1.5)$.

## STED data acquisition and analysis

Sample preparation, Unc13A antibody staining, STED image acquisition and the isolation of single AZ images are described in *Böhme et al. (2019)* and in the following. Third-instar *w[1118]* larvae were put on a dissection plate with both ends fixed by fine pins. Larvae were then covered by 50 μl of ice-cold hemolymph-like saline solution (HL3, pH adjusted to 7.2 [*Stewart et al., 1994*]: 70 mM NaCl, 5 mM KCl, 20 mM MgCl$_2$, 10 mM NaHCO$_3$, 5 mM Trehalose, 115 mM D-Saccharose, 5 mM HEPES). Using dissection scissors a small cut at the dorsal, posterior midline of the larva was made from where on the larvae was cut completely open along the dorsal midline until its anterior end. Subsequently, the epidermis was pinned down and slightly stretched and the internal organs and tissues removed. For the 'STED dataset 2' shown in *Figure 1—figure supplement 1C,D*, animals were then incubated in a HL3 solution containing 0.5% DMSO for 10 min (this served as a mock control for another experiment not shown in this paper using a pharmacological agent diluted in DMSO). The dissected samples were washed 3x with ice-cold HL3 and then fixed for 5 min with ice-cold methanol. After fixation, samples were briefly rinsed with HL3 and then blocked for 1 hr in 5% native goat serum (NGS; Sigma-Aldrich, MO, USA, S2007) diluted in phosphate buffered saline (Carl Roth Germany) with 0.05% Triton-X100 (PBT). Subsequently dissected samples were incubated with primary antibodies (guinea-pig Unc13A 1:500; *Böhme et al., 2016*) diluted in 5% NGS in PBT overnight. Afterwards samples were washed 5x for 30 min with PBT and then incubated for 4 hr with fluorescence-labeled secondary antibodies (goat anti-guinea pig STAR635 (1:100) diluted in 5% NGS in PBT. For secondary antibody production STAR635 fluorophore (Abberior, Germany) was coupled to respective IgGs (Dianova, Germany). Samples were then washed overnight in PBT and subsequently mounted in Mowiol (Max-Planck Institute for Biophysical Chemistry, Group of Stefan Hell) on high-precision glass coverslips (Roth, Germany, LH24.1). Two-color STED images were recorded on a custom-built STED-microscope (*Göttfert et al., 2017*), which combined two pairs of excitation laser beams of 595 nm and 635 nm with one STED fiber laser beam at 775 nm. All STED images were acquired using Imspector Software (Max Planck Innovation GmbH, Germany). STED images were processed using a linear deconvolution function integrated into Imspector Software (Max Planck Innovation GmbH, Germany). Regularization parameter was 1e$^{-11}$. The point spread function (PSF for deconvolution was generated using a 2D Lorentz function with its half-width and half-length fitted to the half-width and half-length of each individual image. Single AZ images of 'STED dataset 1' (*Figure 1E,F*, *Figure 1—figure supplement 1C,D*) had previously been used for a different type of analysis defining AZ Unc13A cluster numbers; Wild-type in supplementary Figure 2a of *Böhme et al. (2019)*. In this study here, we wanted to obtain the average Unc13A distribution from all AZs (no distinction of AZ types). To get an average image of the Unc13A AZ distribution, we used a set of hundreds of 51 × 51 pixel images with a pixel size of 10 × 10 nm. We identified Unc13A clusters in each image using the fluorescence peak detection procedure described in *Böhme et al. (2019)* using MATLAB (version 2016b). Peak detection was performed as follows: In each deconvolved 51 × 51 pixel image of an Unc13A-stained AZ, a threshold of 25 gray values was applied below which no pixels were considered. Then, local maxima values were found by finding slope changes corresponding to peaks along pixel columns using the function *diff*. The same was done along rows for all column positions where peaks were found. The function *intersect* was then used to determine all pixel positions common in both columns and rows. A minimum distance of 50 nm between neighboring peaks was used to exclude the repeated detection of the same peak, and an edge of 10 nm around the image was excluded to prevent the detection of neighboring AZs. The center of mass of all peak *x,y*-coordinates found in a single image was then calculated as follows:

$$P_x = n^{-1} * \sum_{1}^{n} x_{obs}(n)$$

$$P_y = n^{-1} * \sum_{1}^{n} y_{obs}(n)$$

Here, n is the number of detected peaks, $(P_x, P_y)$ represents the center of mass $(x,y)$-coordinate, and $x_{obs}(n)$ and $y_{obs}(n)$ are the coordinates of the $n$-th detected peak. The image was then shifted such that this position $(P_x, P_y)$ would fall into the center pixel of the $51 \times 51$ AZ image. For this, we calculated the required shift ($d_x$ and $d_y$):

$$d_x = \frac{imgsize(x)}{2} - P_x$$

$$d_y = \frac{imgsize(y)}{2} - P_y$$

Here, imgsize($x,y$) refers to the pixel dimensions of the image in both $x$ and $y$ dimensions. The required shift $d_{x,y}$ was then applied to the image using *imtranslate*, which directly takes these shift values as an input. All shifted images were then averaged into a single compound average image of all AZs by taking the average of each individual pixel and linearly scaling the result in a range between 0 and 255. This resulted in a circular cloudy structure depicted in *Figure 1E*, *Figure 1—figure supplement 1C*. To obtain the distribution of fluorescence as a function of distance to the AZ center in the average picture, we determined the distance between the center of the image and the center of the pixel together with the fluorescence intensity in each pixel. The fluorescence intensity in each pixel was obtained by using the inbuilt MATLAB function 'imread', which outputs the intensities in a matrix with indexes corresponding to the pixel location in the picture. From the indexes ($x_p$, $y_p$) of each pixel (of size 10 nm), the distance (in nm) to the center was calculated by the following formula:

$$d(p) = \left( \sqrt{(x_p - 26)^2 + (y_p - 26)^2} \right) \cdot 10\text{nm}$$

We subtracted 26 from the pixel number, since the center pixel is the 26th pixel in x- and y-direction. These distances together with the intensity at each pixel provided the data for the histograms in *Figure 1F* and *Figure 1—figure supplement 1D*. The intensity values were normalized to the total amount of intensity making the y-axis of the histogram show percentage of the total amount of intensity.

## Calculation of mean distance to four nearest neighbors (1–4-NND)

Stage L3 larvae (n = 17; genotype: w[1118]; P{w[+mC]=Mhc-SynapGCaMP6f}3–5, Bloomington #67739) were fixed in ice-cold Methanol for 7 min and IHC-stained for BRP (mouse anti-Nc82, 1:1000; secondary AB: goat anti-mouse Cy5 1:500). Confocal images of the preparations were taken and processed as described in *Reddy-Alla et al. (2017)* for a different set of experiments not shown in this paper. Subsequently, the BRP channel was used to identify local fluorescence intensity maxima using the ImageJ-function 'Find Maxima' with a threshold setting between 10 and 20. The locations of maxima for each cell were then loaded into MATLAB (version 2016b) and the distances of each x,y-coordinate to all others were determined using the MATLAB function *pdist2*, resulting in a square matrix containing all possible inter-AZ distances. Each column of this matrix was then sorted in ascending order, and (as the distance of one AZ to itself is always 0) the mean of the 2nd to 5th smallest values across all AZs was determined and depicted as 1-NND through 4-NND in *Figure 3A*. The mean distance of the four nearest neighbouring AZs (1–4-NND) was calculated in each AZ (gray circles in *Figure 3A* bottom right) and the mean across AZs was used for quantification of the simulation volume (see below).

## Electrophysiological data acquisition and analysis

For both eEJC and mEJC (spontaneous release events,"miniature Excitatory Junctional Currents') recordings, two electrode voltage clamp (TEVC) recordings were performed from muscle 6 NMJs of abdominal segments A2 and A3 as reported previously (*Qin et al., 2005*). Prior to recordings, the larvae were dissected in haemolymph-like solution without Ca²⁺ (HL3, pH adjusted to 7.2

*Stewart et al., 1994*: 70 mM NaCl, 5 mM KCl, 20 mM MgCl$_2$, 10 mM NaHCO$_3$, 5 mM Trehalose, 115 mM D-Saccharose, 5 mM HEPES) on Sylgard (184, Dow Corning, Midland, MI, USA) and transferred into the recording chamber containing 2 ml of HL3 with CaCl$_2$ (concentrations used in individual experiments described below). TEVC recordings were conducted at 21°C using sharp electrodes (borosilicate glass with filament, 0.86×1.5×80 nm, Science Products, Hofheim, Germany) with pipette resistances between 20–30 MΩ, which were pulled with a P-97 micropipette puller (Sutter Instrument, CA, USA) and filled with 3 mM KCl. Signals were low-pass filtered at 5 KHz and sampled at 20 KHz. Data was obtained using a Digidata 1440A digitizer (Molecular devices, Sunnyvale, CA, USA), Clampex software (v10.6) and an Axoclamp 900A amplifier (Axon instruments, Union City, CA, USA) using Axoclamp software. Only cells with a resting membrane potential V$_m$ below −50 mV, membrane resistances R$_m$ above 4 MΩ and an absolute leak currents of less than 10 nA were included in the dataset.

## eEJC recordings

eEJC recordings were conducted at a membrane holding potential of −70 mV in TEVC mode. APs were evoked by giving 300 μs short depolarizing pulses (8 V) to respective innervating motoneuron axons using a suction electrode (pulled with DMZ-Universal Puller (Zeitz-Instruments GmbH, Germany) polished with the CPM-2 microforge (ALA Scientific, NY, USA)) and a stimulator (S48, Grass Technologies, USA).

For experiments shown in *Figure 2*, individual cells were recorded at an initial extracellular CaCl$_2$ concentration of 0.75 mM which was subsequently increased to 1.5 mM, 3 mM, 6 mM and 10 mM by exchanging and carefully mixing 1 ml of the bath solution with 1 ml HL3 of a higher CaCl$_2$ concentration (total concentrations of exchange solutions: 2.25 mM, 4.5 mM, 9 mM, 14 mM), ultimately adding up to the desired CaCl$_2$ concentration in the bath. At each titration step, cells were acclimated in the bath solution for 60 s and 10 repetitions of paired stimulating pulses (0.1 Hz, 10 ms interstimulus interval) were given. eEJC data shown in *Figure 2—figure supplement 3* was obtained by recording *Ok6-Gal /+* and *+/+* NMJs at 0.75 mM (*Figure 2—figure supplement 3A-D* ) and 1.5 mM (*Figure 2—figure supplement 3E-H* ) Ca$^{2+}$. A single test AP was given (followed by a 20 s intermission) and cells were stimulated once by two consecutive APs (10 ms inter-stimulus interval). In *Figure 2—figure supplement 3B, D, E, and G*, eEJC$_1$ and PPR averages are shown ± the estimated single-cell SD .

eEJC data was analyzed with our own custom-built MATLAB script (provided with the source data file, *Figure 2—source data 1*). After stimulation artifact removal, the eEJC$_1$ amplitude was determined as the minimum current value within 10 ms from the time of stimulation. To account for the decay only being partial before the second stimulus, we fitted a single exponential function to the eEJC decay from the time point of 90% of the amplitude to the time point of the second stimulus. The eEJC$_2$ amplitude was determined as the difference between the minimum after the second stimulus and the value of the fitted exponential at the time point of the second minimum (see insert in *Figure 2C* and *Figure 2—figure supplement 1A*). For analysis shown in *Figure 2*, the first stimulation per Ca$^{2+}$ concentration was excluded, as we noticed that the first trial often gave first eEJC responses that were higher than in the following trials. This may reflect the presence of a slow reaction by which SVs can be primed with an even higher release probability (possibly due to the 'superpriming' described at the murine Calyx of Held synapse *Lee et al., 2013*). However, as the var/mean analysis requires the existence of an equilibrium in-between stimuli which appears to have been reached between all of the succeeding stimuli, we decided to use only those for our analysis. For eEJC$_1$ amplitudes the average over all measurements and all cells (6 cells, nine measurements each) was calculated (*Figure 2B*). The PPR was calculated by dividing the second amplitude by the first throughout trials and averaging over all measurements and all cells (*Figure 2D*). In each cell, the variance of eEJC$_1$ and PPR was estimated (nine stimulations per Ca$^{2+}$ concentration) and the average variance (averaged across cells) was calculated at each extracellular Ca$^{2+}$ concentration. The error bars in *Figure 2B,D* are the SD (across all animals) at each extracellular Ca$^{2+}$ concentration. In *Figure 2F* the eEJC$_1$ averages and variances are ± SEM. A parabola with intersect y = 0 was fitted using the function *polyfitZero* (version 1.3.0.0 from MathWorks file exchange) in MATLAB. (Var = q*I-I$^2$/N, q being the quantal size, I the mean eEJC$_1$ amplitude and N number of release sites) (*Clements and Silver, 2000*).

## mEJC recordings

mEJC data was obtained from a separate set of experiments where mEJCs were recorded for 60 s in TEVC mode at 1.5 mM extracellular $Ca^{2+}$ and a holding potential of $-80$ mV for easier identification of miniature events. Because different holding potentials were used ($-80$ mV here compared to $-70$ mV for the data shown in *Figure 2*) it must be pointed out that these recordings were only used to determine the shape of the response for later convolution with SV fusion events predicted by the model (the mEJC amplitude was adjusted based on the variance-mean data collected at -70 mV, see below). For this, the average mEJC traces from five different cells were aligned to 50% of the rise and averaged. We then fitted the following formula to the data:

$$I_{mini}(t) = A \cdot \left(1 - e^{-\frac{(t-t_0)}{\tau_r}}\right) \cdot \left(B \cdot e^{-\frac{(t-t_0)}{\tau_{df}}} + (1-B) \cdot e^{-\frac{(t-t_0)}{\tau_{ds}}}\right)$$

$t_0$ is the onset, $A$ is the full amplitude (if there was no decay), $B$ is the fraction of the fast decay, and $\tau_r$, $\tau_{df}$, $\tau_{ds}$ are the time constants of the rise, fast decay, and slow decay respectively.

The best fit was

$$t_0 \approx 3.0 \ ms, \ A \approx 7.21 \ \mu A, B \approx 2.7e - 9,$$
$$\tau_r \approx 10.6928 \ s, \ \tau_{df} \approx 1.5 \ ms, \ \tau_{ds} \approx 2.8 \ ms$$

and is plotted together with the average experimental mini trace in *Figure 2—figure supplement 1B*. Note that $t_0$ is a time delay when this mEJC is implemented in the simulation and is therefore arbitrary. B is very small making the decay close to a single exponential. The maximum of this function is ~0.7 nA. However, as mentioned above, this function was rescaled to a value of 0.6 nA to match the mEJC amplitudes of the experiments conducted with a holding potential of -70 mV, that is the size of a single quantal event, q=0.6 nA, estimated from the variance-mean analysis (see *Figure 2F*).

## Presynaptic GCaMP recordings and analysis

Because the presynaptic terminals of the *Drosophila* larval NMJ are not readily accessible to electrical recordings of $Ca^{2+}$ currents, the saturation behaviour of $Ca^{2+}$ influx as a function of extracellular $Ca^{2+}$ concentrations was measured. We did so by engaging the fluorescent $Ca^{2+}$ indicator GCaMP6m (Genotype: w[1118]; P{y[+t7.7] w[+mC]=20XUAS-IVS-GCaMP6m}attP40, Flybase ID: FBti0151346), which we expressed presynaptically using OK6-Gal4 as a motoneuron-specific driver. Third instar larvae heterozygously expressing the indicator were used in experiments as follows. Dissection took place in $Ca^{2+}$-free, standard hemolymph-like solution HL-3 (in mM: NaCl 70, KCl 5, $MgCl_2$ 20, $NaHCO_3$ 10, Trehalose 5, Sucrose 115, HEPES 5, pH adjusted to 7.2) (*Stewart et al., 1994*). After dissection on a Sylgard-184 (Dow-Corning) block, larvae were transferred to the recording chamber containing HL-3 at varying $CaCl_2$ concentrations (see below). The efferent motoneuron axons were sucked into a polished glass electrode containing a chlorided silver-wire, which could be controlled via a mechanical micromanipulator (Narishige NMN25) and was connected to a pipette holder (PPH-1P-BNC, NPI electronics) via a patch electrode holder (NPI electronics), and connected to an S48 stimulator (Grass Technologies). Larvae were then recorded using a white-light source (Sutter DG-4, Sutter Instruments) and a GFP filter set with a Hamamatsu OrcaFlash 4.0v2 sCMOS (Hahamatsu Photonics) with a framerate of 20 Hz (50 ms exposure) controlled by µManager software (version 1.4.20, https://micro-manager.org) on an upright microscope (Olympus BX51WI) with a 60x water-immersion objective (Olympus LUMFL 60 × 1.10 w). Muscle 4 1b NMJs in abdominal segments 2 to 4 were used for imaging. Imaging was conducted over 10 s, and at 5 s, 20 stimuli were applied to the nerve at 20 Hz in 300µs 7V depolarization steps. This procedure was begun in the lowest $Ca^{2+}$ concentration (0.75 mM) and then repeated in the same larva at increasing $Ca^{2+}$ concentrations (in mM 1.5, 3, 6) by exchanging the extracellular solution. To achieve a situation with no $Ca^{2+}$ influx, a final recording was conducted where the bath contained HL-3 without $CaCl_2$ and instead 8.3 mM EGTA (this solution was made by diluting 2.5 ml of a 50 mM stock solution in $H_2O$ in 12.5 ml of HL3, resulting in a pH of 8.0). Because this results in a slight dilution (16%) of the components in the HL3, the same dilution was performed for the above described $Ca^{2+}$-containing solutions by adding 2.5 ml $H_2O$ to 12.5 ml of HL3 before $CaCl_2$ was added at above mentioned concentrations.

Analysis of 5 *Drosophila* 3rd instar Larvae was done after automated stabilization of *x,y*-movement in the recordings (8-bit multipage .TIF-stacks, converted from 16 bit) as described previously (*Reddy-Alla et al., 2017*), manually selecting a ROI around the basal fluorescent GCaMP signal, and reading out the integrated density (the sum of all pixel grey values) of the whole region over time. Background fluorescence was measured in a region of the same size and shape outside of the NMJ and subtracted (frame-wise) from the signal, separately for each single recording. The quantification was then performed individually for each $Ca^{2+}$ concentration, by subtracting the fluorescence 250 ms before the stimulation ($F_{t=4.75s}$) from the maximum fluorescence of the trace ($F_{max}$), yielding the change in fluorescence dF:

$$dF\left(Ca^{2+}\right) = F_{max} - F_{t=4,75s}$$

This was repeated for each cell and a Hill fit was performed on the individual values using Prism (version 6.07, GraphPad Software Inc):

$$F\left(\left[Ca^{2+}\right]_{ext}\right) = \frac{F_{end} * \left[Ca^{2+}\right]_{ext}^{m}}{\left(K_{M,fluo}\right)^{m} + \left[Ca^{2+}\right]_{ext}^{m}} + C \tag{3}$$

In the above equation, $F_{end}$ is the asymptotic plateau of the fluorescence increase. Furthermore, $[Ca^{2+}]_{ext}$ is the extracellular $Ca^{2+}$ concentration. $K_{M,fluo}$ (best fit value: 2.679 mM) is the concentration of extracellular $Ca^{2+}$ at which fluorescence was half of $F_{end}$. The exponent m indicates a cooperative effect of the extracellular $Ca^{2+}$ concentration on the fluorescence increase, which was constrained to a value of 2.43 (unitless) based on the described $Ca^{2+}$ cooperativity of GCaMP6m (*Barnett et al., 2017*). However, constraining this value only had a modest effect on the estimate of $K_{M,fluo}$ as leaving it as a free parameter yielded similar values for $K_{M,fluo}$ (3.054 mM) and m (1.887). The constant C added at the end of *Equation 3* allowed the baseline fluorescence to be different from zero. Results and best fit are summarized in (*Figure 3—figure supplement 1*).

## Proof that stochastic simulation of release is needed for PPR estimation

We here prove that stochastic simulations of neurotransmitter release provide a different average PPR value than the PPR value estimated in deterministic simulations. In the following, the stochastic variables $A_1$ and $A_2$ represent the amplitudes of the first and second release, respectively, capital 'E' denotes the mean of a stochastic variable (e.g. $EA_1$), and $a_1$ and $a_2$ represent the amplitudes of the first and second release in the deterministic simulations. In all cases of parameter sets that we tried, the average amplitudes from the stochastic simulations with 1000 repetitions differed < 0.5 nA from the deterministically determined amplitudes. Thus, we can assume that $EA_1 = a_1$ and likewise for the second release.

In deterministic simulations, the estimate of the PPR is

$$\bar{PPR} = \frac{a_2}{a_1} = \frac{EA_2}{EA_1}$$

On the other hand, stochastic simulations yield a sample of different PPR values, since repetitions of the simulation routine yield release varying from trial to trial. In that case, the estimated PPR is

$$\tilde{PPR} = E\left(\frac{A_2}{A_1}\right) \tag{4}$$

This resembles the way the PPR is estimated in experiments.

Using Jensen's Inequality and the fact that the function f(x)=1/x is strictly convex, we get

$$\frac{1}{EA_1} < E\left(\frac{1}{A_1}\right) = E\left(A_1^{-1}\right)$$

Applying this to (4) we get

$$\begin{aligned}
\tilde{PPR} &= \mathrm{E}\left(\frac{A_2}{A_1}\right) \\
&= \mathrm{E}\left(A_1^{-1}A_2\right) \\
&= Cov\left(A_1^{-1}, A_2\right) + \mathrm{E}\left(A_1^{-1}\right)\mathrm{E}(A_2) \\
&> Cov\left(A_1^{-1}, A_2\right) + \frac{\mathrm{E}A_2}{\mathrm{E}A_1} \\
&= Cov\left(A_1^{-1}, A_2\right) + \bar{PPR}
\end{aligned}$$

Thus, the average stochastically simulated PPR do not necessarily converge to the deterministic estimate with increasing repetitions (note that in general it is true that the mean of a non-linear function of two random variables is not equal to the non-linear function evaluated in the means). An example is shown in *Figure 4—figure supplement 1*, where the single-sensor model was simulated with varying amounts of Ca$^{2+}$ influx (by varying Q$_{max}$). The most left blue point, for example, is significantly higher than the deterministic estimate (p=4e-16, one-sample t-test). This motivates the use of stochastic simulations for correct estimation of the PPR.

## Simulation flow

All MATLAB procedures for simulation of the models can be found in *Source code 1*.

All simulations (deterministic and stochastic, see below) consisted of the same four basic steps, which we describe in detail here.

1. Given a set of parameters, we first ran deterministic Ca$^{2+}$ simulations in space and time in the presynapse at the desired extracellular Ca$^{2+}$ concentrations.
2. A set of SV distances was drawn from the generalized gamma distribution. The set of SV distances provided the points at which to read the intracellular Ca$^{2+}$ concentrations for the exocytosis simulation.
3. The simulation of the models for Ca$^{2+}$ binding and exocytosis was performed for each SV position with the Ca$^{2+}$ transients giving rise to the changing reaction rates.
4. The outcome of the exocytosis simulation were convolved with a mEJC which yielded the eEJC.

For each new set of parameters, steps 1–4 were repeated. For stochastic simulations, steps 2–4 were repeated 1000 times except for the parameter exploration in *Figures 6J–K* and *7J–K*, where we ran 200 repetitions per parameter set. The many repetitions allowed a good estimate of both mean and variance of the models. In all cases, the mean amplitudes from the stochastic simulations with 1000 repetitions differed < 0.5 nA from the deterministically determined amplitudes.

## Ca$^{2+}$ simulation

Simulation of Ca$^{2+}$ signals in the presynapse was performed with the program CalC version 6.8.6 developed and maintained by Victor Matveev (*Matveev et al., 2002*). After this work was initiated, a bug affecting simulations of multiple Ca$^{2+}$ channels in the same topology was found and a new version of CalC was released. This update had no effect on the simulations used in this study.

Intracellular Ca$^{2+}$ concentrations were simulated in space and time in a cylinder-shaped volume. The cylinder allowed us to assume spatial symmetry which reduced simulation time significantly. Borders of the simulation volume were assumed to be reflective to mimic diffusion of Ca$^{2+}$ from adjacent AZs (*Meinrenken et al., 2002*) and a volume-distributed uptake mechanism was assumed.

From measurements of the distance between an AZ and its four nearest neighbors (*Figure 3A*) we estimated the distance between centers of active zones to be 1.106 µm, leading to the assumption that the AZ spans a square on the membrane with area of 1.223 µm$^2$. In order for the cylindrical simulation volume to cover an area of the same size, the radius was set to 0.624 µm. The height of the simulation volume was set to 1 µm making the simulation volume 1.223 µm$^3$. Increasing the height further had no effect on the Ca$^{2+}$ transients.

The total amount of charge flowing into the cell was assumed to relate to extracellular Ca$^{2+}$ in a Michaelis-Menten-like way (as previously described by *Schneggenburger et al., 1999*; *Trommershäuser et al., 2003*) such that

$$Q = \frac{Q_{max} \cdot [Ca]_{ext}}{K_{M,current} + [Ca]_{ext}} \tag{5}$$

$K_{M,current}$ was set to the value of 2.679 mM as determined for $K_{M,fluo}$ in the GCaMP6m experiments (see above). $Q_{max}$ was fitted during the optimizations of the models.

We simulated a 10 ms paired pulse stimulus initiated after 0.5 ms of simulation. The Ca²⁺ currents for the two stimuli were simulated for 3 ms each and assumed to be Gaussian with FWHM = 360 µs and peak 1.5 ms after initiation. That is:

$$I_{Ca} = \begin{cases} Q \cdot \frac{1}{\sigma \cdot \sqrt{2\pi}} e^{-\frac{(t-2)^2}{2\sigma^2}}, & for\ t\in[0.5,\ 3.5] \\ Q \cdot \frac{1}{\sigma \cdot \sqrt{2\pi}} e^{-\frac{(t-12)^2}{2\sigma^2}}, & for\ t\in[10.5,\ 13.5] \\ 0, & else \end{cases}$$

with $\sigma = \frac{0.360}{2\sqrt{2 \cdot \ln(2)}} = 0.153$.

The CalC simulation output were data files that contained the spatio-temporal intracellular Ca²⁺ profile at the height of 10 nm from the plasma membrane. In exocytosis simulations, these concentrations were interpolated at the SV distances in the x,y-plane and at time points with MATLAB's built-in interpolate functions when computing the reaction rates of the system at a given time point.

The resting Ca²⁺ concentration was assumed to relate to the extracellular Ca²⁺ concentration in a similar way as during stimulation, such that

$$[Ca^{2+}]_{basal} = [Ca^{2+}]_{max} \cdot \frac{[Ca^{2+}]_{ext}}{K_{M,current} + [Ca^{2+}]_{ext}} \tag{6}$$

with $[Ca^{2+}]_{max} = 190\ nM$

For designation and value of Ca²⁺ parameters, see **Table 1**.

## SV distribution drawing

In all simulations we had to determine where to place release site. This was done by using the cdf of the SV distance distribution derived above (**Equation 2**).

For deterministic simulations, which were used in the fitting routine of the models (see below), the unit interval was divided into 180 bins of the form

$$\left(\frac{k-1}{180}, \frac{k}{180}\right),\ \ k = 1,\ 2\dots 180.$$

The midpoints were the percentiles giving rise to distances at which we read the Ca²⁺ simulation. This approach provided an approximation of the SV distribution. In accordance with our assumption that the AZs work in parallel the 180 distances gave rise to 180 independent different systems of ODEs with 1/180 of the total amount of SVs in each system. The results were then added together as a good approximation of the mean of the stochastic simulations with random SV distance drawings.

In each run of the stochastic simulations, we drew $n$ random numbers from the unit interval, $n$ being the number of SVs, and computed the distances based on the formula derived above.

## Rate equations of the simulated models

The models are summarized in **Figures 4A**, **6A** and **7A**, and **Figure 7—figure supplement 3A,B**. In the following equations the single-sensor, dual fusion-sensor, and unpriming models are all described. The site activation model is a combination of the equations for the single-sensor model and the site activation equations described below. The red text denotes terms that are unique to the dual fusion-sensor model, blue text indicates unpriming, which is unique to the unpriming model. Parameters are described below. For designation and value of parameters, see **Tables 2,3**.

Rate equations of the single-sensor model, dual fusion-sensor model and unpriming model:

$$\frac{d[R(0,0)]}{dt} = k_{rep}[P0] - (r \cdot u + 5[Ca^{2+}]k_1 + 2[Ca^{2+}]k_2 + L^+)$$
$$[R(0,0)] + k_{-1}[R(1,0)]$$
$$+ k_{-2}[R(0,1)]$$

$$\frac{d[R(1,0)]}{dt} = -(4[Ca^{2+}]k_1 + k_{-1} + 2[Ca^{2+}]k_2 + L^+f)[R(1,0)]$$
$$+5[Ca^{2+}]k_1[R(0,0)] + 2b_f k_{-1}[R(2,0)]$$
$$+ k_{-2}[R(1,1)]$$

$$\frac{d[R(2,0)]}{dt} = -(3[Ca^{2+}]k_1 + 2b_f k_{-1} + 2[Ca^{2+}]k_2 + L^+f^2)[R(2,0)]$$
$$+4[Ca^{2+}]k_1[R(1,0)] + 3b_f^2 k_{-1} \cdot [R(3,0)]$$
$$+k_{-2} \cdot [R(2,1)]$$

$$\frac{d[R(3,0)]}{dt} = -(2[Ca^{2+}]k_1 + 3b_f^2 k_{-1} + 2[Ca^{2+}]k_2 + L^+f^3)[R(3,0)]$$
$$+3[Ca^{2+}]k_1[R(2,0)] + 4b_f^3 k_{-1} \cdot [R(4,0)]$$
$$+k_{-2} \cdot [R(3,1)]$$

$$\frac{dR[(4,0)]}{dt} = -([Ca^{2+}]k_1 + 4b_f^3 k_{-1} + 2[Ca^{2+}]k_2 + L^+f^4)[R(4,0)]$$
$$+2[Ca^{2+}]k_1[R(3,0)] + 5b_f^4 k_{-1} \cdot [R(5,0)]$$
$$+k_{-2} \cdot [R(4,1)]$$

$$\frac{d[R(5,0)]}{dt} = -(2[Ca^{2+}]k_2 + 5b_f^4 k_{-1} + L^+f^5)[R(5,0)] + [Ca^{2+}]k_1[R(4,0)]$$
$$+k_{-2} \cdot [R(4,1)]$$

$$\frac{d[R(0,1)]}{dt} = k_{rep}[P1] - (5[Ca^{2+}]k_1 + [Ca^{2+}]k_2 + k_{-2} + L^+s)$$
$$[R(0,1)] + k_{-1} \cdot [R(1,1)]$$
$$+2[Ca^{2+}]k_2[R(0,0)] + 2b_s k_{-2} \cdot [R(0,2)]$$

$$\frac{d[R(1,1)]}{dt} = -(4[Ca^{2+}]k_1 + k_{-1} + [Ca^{2+}]k_2 + k_{-2} + L^+fs)[R(1,1)]$$
$$+5[Ca^{2+}]k_1[R(0,1)]$$
$$+2b_f k_{-1} \cdot [R(2,0)] + 2[Ca^{2+}]k_2[R(1,0)] + 2b_s k_{-2} \cdot [R(1,2)]$$

$$\frac{d[R(2,1)]}{dt} = -(3[Ca^{2+}]k_1 + 2b_f k_{-1} + [Ca^{2+}]k_2 + k_{-2} + L^+f^2 s)[R(2,1)]$$
$$+4[Ca^{2+}]k_1[R(1,1)]$$
$$+3 \cdot b_f^2 \cdot k_{-1}[R(3,1)] + 2[Ca^{2+}]k_2[R(2,0)] + 2b_s k_{-2} \cdot [R(2,2)]$$

$$\frac{d[R(3,1)]}{dt} = -(2[Ca^{2+}]k_1 + 3b_f^2 k_{-1} + [Ca^{2+}]k_2 + k_{-2} + L^+f^3 s)[R(3,1)]$$
$$+3[Ca^{2+}]k_1[R(2,1)] + 4b_f^3 k_{-1} \cdot [R(4,1)]$$
$$+ 2[Ca^{2+}]k_2[R(3,0)] + 2b_s k_{-2}$$
$$\cdot [R(3,2)]$$

$$\frac{d[R(4,1)]}{dt} = -([Ca^{2+}]k_1 + 4b_f^3 k_{-1} + [Ca^{2+}]k_2 + k_{-2} + L^+f^4 s)[R(4,1)]$$
$$+2[Ca^{2+}]k_1[R(3,1)]$$
$$+5b_f^3 k_{-1} \cdot [R(5,1)] + 2[Ca^{2+}]k_2[R(4,0)] + 2b_s k_{-2}[R(4,2)]$$

$$\frac{d[R(5,1)]}{dt} = -(5b_f^4 k_{-1} + [Ca^{2+}]k_2 + k_{-2} + L^+f^5 s)[R(5,1)] + [Ca^{2+}]k_1[R(4,1)]$$
$$+ 2[Ca^{2+}]k_2[R(5,0)] + 2b_s k_{-2} \cdot [R(5,2)]$$

$$\frac{d[R(0,2)]}{dt} = k_{rep}[P2] - (5[Ca^{2+}]k_1 + 2b_s k_{-2} + L^+s^2)[R(0,2)] + k_{-1}[R(1,2)]$$
$$+[Ca^{2+}]k_2[R(0,1)]$$

$$\frac{d[R(1,2)]}{dt} = -(4[Ca^{2+}]k_1 + k_{-1} + 2b_s k_{-2} + L^+fs^2)[R(1,2)] + 5[Ca^{2+}]k_1[R(0,2)]$$
$$+ 2b_f k_{-1}[R(2,2)] + [Ca^{2+}]k_2[R(1,1)]$$

$$\frac{d[R(2,2)]}{dt} = -\left(3[Ca^{2+}]k_1 + 2b_f k_{-1} + 2b_s k_{-2} + L^+ f^2 s^2\right)[R(2,0)] + 4[Ca^{2+}]k_1[R(1,2)] \\ + 3b_f^2 k_{-1}[R(3,0)] + [Ca^{2+}]k_2[R(2,1)]$$

$$\frac{d[R(3,2)]}{dt} = -\left(2[Ca^{2+}]k_1 + 3b_f^2 k_{-1} + 2b_s k_{-2} + [Ca^{2+}]k_2 + L^+ f^3 s^2\right)[R(3,2)] \\ + 3[Ca^{2+}]k_1[R(2,2)] + 4b_f^3 k_{-1} \cdot [R(4,2)] + [Ca^{2+}]k_2[R(3,1)]$$

$$\frac{d[R(4,2)]}{dt} = -\left([Ca^{2+}]k_1 + 4b_f^3 k_{-1} + 2b_s k_{-2} + [Ca^{2+}]k_2 + L^+ f^4 s^2\right)[R(4,2)] \\ + 2[Ca^{2+}]k_1[R(3,2)] + 5b_f^3 k_{-1} \cdot [R(5,2)] + [Ca^{2+}]k_2[R(4,1)]$$

$$\frac{d[R(5,2)]}{dt} = -\left(5b_f^4 k_{-1} + 2b_s k_{-2} + L^+ f^5 s^2\right)[R(5,2)] + [Ca^{2+}]k_1 \\ [R(4,2)] + [Ca^{2+}]k_2[R(5,1)]$$

$$\frac{d[F]}{dt} = L^+([R(0,0)] + f[R(1,0)] + f^2[R(2,0)] + f^3[R(3,0)] + f^4[R(4,0)] + f^5[R(5,0)] \\ + [sR(0,1)] + fs[R(1,1)] + f^2 s[R(2,1)] + f^3 s[R(3,1)] + f^4 s[R(4,1)] \\ + f^5 s[R(5,1)] + [s^2 R(0,2)] + fs^2[R(1,1)] + f^2 s^2[R(2,1)] + f^3 s^2[R(3,1)] \\ + f^4 s^2[R(4,1)] + f^5 s^2[R(5,1)])$$

$$\frac{d[P0]}{dt} = L^+([R(0,0)] + f[R(1,0)] + f^2[R(2,0)] \\ + f^3[R(3,0)] + f^4[R(4,0)] + f^5[R(5,0)]) \\ + k_{-2}[P1] - 2k_2[Ca^{2+}][P0] - k_{rep}[R(0,0)] + r \cdot u[R(0,0)]$$

$$\frac{d[P1]}{dt} = L^+([sR(0,1)] + fs[R(1,1)] + f^2 s[R(2,1)] + f^3 s[R(3,1)] + f^4 s[R(4,1)] \\ + f^5 s[R(5,1)]) - k_{-2}[P1] + 2k_2[Ca^{2+}][P0] + 2b_s k_{-2} \cdot [P2] - k_{rep}[R(0,1)]$$

$$\frac{d[P2]}{dt} = L^+([R(0,0)] + f[R(1,0)] + f^2[R(2,0)] + f^3[R(3,0)] + f^4[R(4,0)] + f^5[R(5,0)] \\ + [sR(0,1)] + fs[R(1,1)] + f^2 s[R(2,1)] + f^3 s[R(3,1)] + f^4 s[R(4,1)] \\ + f^5 s[R(5,1)] + [s^2 R(0,2)] + fs^2[R(1,1)] + f^2 s^2[R(2,1)] + f^3 s^2[R(3,1)] \\ + f^4 s^2[R(4,1)] + f^5 s^2[R(5,1)]) + 2k_2[P1] - 2b_s k_{-2}[Ca^{2+}][P2] - k_{rep}[R(0,2)]$$

$$r = 1 - \frac{[Ca^{2+}]^n}{[Ca^{2+}]^n + K_{M,unprim}^n}$$

In the single-sensor and site activation models, $k_2 = k_{-2} = u = 0$, and $s = 1$. This excludes all reactions exclusive for the dual fusion-sensor and unpriming models. Similarly, $u = 0$ in the dual fusion-sensor model and $k_2 = k_{-2} = 0$ and $s = 1$ in the unpriming model.

$[R(n,m)]$ denotes the $Ca^{2+}$ binding state of a SV with $n$ $Ca^{2+}$ ions bound to the first sensor and $m$ $Ca^{2+}$ ions bound to the second fusion sensor. Note that in the single-sensor, site activation and unpriming models, $m$ is always zero (since there is no second fusion sensor), and the states are denoted with a single number in *Figures 4A* and *6A* and *Figure 7—figure supplement 3*. $[F]$ counts the cumulative number of fused SVs. $[P0]$ is not shown in the figures, but are part of the equations denoting the number of empty sites. That is, in the single-sensor and unpriming models $\frac{d[P0]}{dt}$ has a positive part equal to $\frac{d[F]}{dt}$ and a negative part equal to the rate of replenishment. In the dual fusion-sensor model, there are three states of empty sites, $[P0]$, $[P1]$, $[P2]$. These corresponded to the different states of $Ca^{2+}$ binding to the second fusion sensor of the empty sites since we assumed the second sensor to be located on the plasma membrane. Note that these equations describe the second sensor with cooperativity 2, which is described in Results. We also optimized cooperativities 3, 4, and 5. The equations can easily be extended to these cases, since the rate equations of the second fusion sensor are of the same form as for the first sensor. In the unpriming model (*Figure 7A*) we assumed unpriming to take place from state *[R(0)]* with a $Ca^{2+}$-dependent rate.

For the individual reactions, we can express the rates of $Ca^{2+}$ (un)binding, fusion, and replenishment of a single SV in a more general form. This is useful in the stochastic simulation method

introduced later. In the following, we denote the general form of the rate for each possible reaction in the models described above.

The expressions in brackets denote the states involved in the reaction.

$$
\begin{aligned}
[R(n,m)] &\rightarrow [R(n-1,m)] &:& nk_{-1}b^{n-1} \\
[R(n,m)] &\rightarrow [R(n+1,m)] &:& (n_{max}-n)[Ca^{2+}]k_1 \\
[R(n,m)] &\rightarrow [R(n,m-1)] &:& mk_{-2}b^{m-1} \\
[R(n,m)] &\rightarrow [R(n,m+1)] &:& (m_{max}-m)[Ca^{2+}]k_2 \\
[R(n,m)] &\rightarrow [F] &:& L^+s^m f^n \\
[P0] &\rightarrow [R(0,0)] &:& k_{rep}
\end{aligned}
\tag{7}
$$

with $n_{max}$ and $m_{max}$ denoting the cooperativity of the first and second fusion sensors, respectively. Equations in line 3 and 4 in (7) were only non-zero in the dual fusion-sensor model.

### Rate equation of the site activation model

In the site activation model (*Figure 7—figure supplement 3*), all reactions regarding Ca$^{2+}$ (un)binding and replenishment was as in the one-sensor model. In addition we assumed a mechanism acting on the release sites independently of the Ca$^{2+}$ binding of the SV. All sites regardless of the SV status were either activated (A state) or not (D or I states). This mechanism is proposed as a facilitation mechanism, which necessitates its primary effect to be on the second stimulus rather than the first. We were therefore forced to implement the D state, which is a temporary 'delay' state making sure the mechanism does not increase first release. The changing of [A] and [I] states at 0.75 and 10 mM extracellular Ca$^{2+}$ are shown in (*Figure 7—figure supplement 3I*).

The site activation mechanism has the following rate equations:

$$
\begin{aligned}
\frac{d[A]}{dt} &= -\delta[A] + \gamma[D] \\
\frac{d[D]}{dt} &= -(\beta+\gamma)[D] + \alpha[Ca^{2+}]^n[I] + \delta[A] \\
\frac{d[I]}{dt} &= -\alpha[Ca^{2+}]^n[I] + \beta[D]
\end{aligned}
$$

where $\alpha$, $\beta$, $\delta$, $\gamma > 0$ are rate parameters.

The deterministic implementation of the site activation model included 3 sets of ODEs, one for each state in the site activation model. Each set consisted of the equations of the one-sensor model as well as transitions between states of equal Ca$^{2+}$ binding in the 3 sets of ODEs (e.g. from R(0,D) to R(0,A)) (*Figure 7—figure supplement 3B*).

In the stochastic simulations the site activation rates were included in the propensity vector like any other reaction. Whenever a site activation reaction occurred, a release site vector consisting of $n_{sites}$ elements was updated. For each site, the fusion rate was multiplied by 0, when the site state was I or D.

### Steady-state estimation

Prior to simulation, the Ca$^{2+}$ binding states of all SVs were assumed to be in equilibrium. We can determine the steady state iteratively by setting

$$
[R(0,0)]_{init} = 1
$$

$$
[R(n+1,m)]_{init} = \frac{(n_{max}-n)[Ca^{2+}]k_1}{(n+1)k_{-1}b^n}[R(n,m)]_{init}
$$

$$
[R(n,m+1)]_{init} = \frac{(m_{max}-m)[Ca^{2+}]k_2}{(m+1)k_{-2}b^m}[R(n,m)]_{init}
$$

This can be reduced to the non-iterative expression:

$$[R(n,m)]_{init} = \frac{\left(\prod_{i=1}^{n}(n_{max}+1-i)\right)\cdot[Ca^{2+}]^n\cdot k_1^n}{n!\cdot b^{\sum_{j=1}^{n}(j-1)}\cdot k_{-1}^n}\cdot\frac{\left(\prod_{i=1}^{m}(m_{max}+1-i)\right)\cdot[Ca^{2+}]^m\cdot k_2^m}{m!\cdot b^{\sum_{j=1}^{m}(j-1)}\cdot k_{-2}^m}$$

$$= \left(\frac{\frac{n_{max}!}{(n_{max}-n)!}\cdot[Ca^{2+}]^n k_1^n}{n!\cdot b^{\frac{n(n-1)}{2}}\cdot k_{-1}^n}\right)\cdot\left(\frac{\frac{m_{max}!}{(m_{max}-m)!}[Ca^{2+}]^m k_2^m}{m!\cdot b^{\frac{m(m-1)}{2}}\cdot k_{-2}^m}\right)$$

Note that for n = 0, the first parenthesis is 1, while m = 0 implies that the second parenthesis is 1, making this solution valid also in the absence of a second fusion-sensor. We ignored the very small fusion rate. In the steady-state of the unpriming model, the number of SVs in [R(0,0)] must furthermore be in equilibrium with the number of empty states:

$$[P] = \frac{r\cdot u}{k_{rep}+r\cdot u}\cdot[R(0,0)]$$

After finding this steady-state, the solution is scaled to match the desired number of SVs by multiplying all states with a constant, such that the sum of all [R(n,m)] and [P] equals the number of SVs. The steady-state of the site activation was determined before simulation by calculating the fraction of states being in [A], [D], or [I]. This was done by calculating

$$[D] = 1, \quad [A] = \frac{\delta}{\gamma}[D] = \frac{\delta}{\gamma}, \quad [I] = \frac{\beta}{[Ca^{2+}]^n\alpha}[D] = \frac{\beta}{[Ca^{2+}]^n\alpha}$$

and normalizing to sum to 1. This determined the steady state fraction of activation of sites. In the stochastic simulations, the SVs were randomly assigned initial states according to the probabilities of the different states in the steady-state.

## Deterministic exocytosis simulation

All deterministic exocytosis simulations of the above equations were carried out with the inbuilt MATLAB ODE solver *ode15s*.

## Stochastic exocytosis simulation

All stochastic exocytosis simulations as well as simulation data handling were carried out in MATLAB with custom-written scripts (included in *Source code 1*). For the simulation itself we used a modified version of the Gillespie Algorithm (*Gillespie, 2007*), which included a minimal time step since reaction rates change quickly with the changing intracellular $Ca^{2+}$ concentration. The minimal step was $\mu$ = 1e-6 s. In the algorithm, the time from the current simulation time point, $t$, until the next reaction, $\tau$, is determined, the reaction is carried out and the new simulation time point is set to t+$\tau$. Whenever the simulation yielded $\tau>\mu$, the simulation time point was set to t+$\mu$, no reaction was carried out and the propensities of the model were updated at the new time point. This is a valid method of obtaining a better estimate because the waiting time until next reaction is exponentially distributed.

The implementation of the algorithm takes advantage of the general form of the rate equations in (7). Instead of calculating matrices of states and reaction rates, we have a vector, $V$, of length $n_{sites}$, where each element represents the status of one SV/site. The SV state of a docked SV on the $k^{th}$ site in state [R(n,m)] is denoted by the two-digit number

$$V_k = m\cdot 10 + n$$

If the site was empty (due to initial submaximal priming or SV fusion) we assigned $V_k = 100$. Using *Equation 7*, the rates of any primed SV are

$$r_k = \begin{pmatrix} m\cdot k_{-2}\cdot b^{m-1} \\ n\cdot k_{-1}\cdot b^{n-1} \\ L^+ f^n s^m \\ (n_{max}-n)\cdot[Ca^{2+}]\cdot k_1 \\ (m_{max}-m)\cdot[Ca^{2+}]\cdot k_2 \\ r\cdot b \end{pmatrix}$$

The sum of these rates of all SVs yield the summed propensities of the system, $a_0$, which is the basis of the calculation of $\tau$, whereas the cumulative sum is used for determination of which SV undergoes a reaction (*Gillespie, 2007*). When a SV undergoes a reaction, we find the index of the reaction occurring, $j$, by using the cumulative sum of $r_k$ in the same way as in the standard implementation of the Gillespie Algorithm (*Gillespie, 2007*). Putting $\hat{j} = j - 3$ allows us to easily update the status of the SV, since

$$V_k = V_k + 1_{(\hat{j} \neq 3)} \cdot sign(\hat{j}) \cdot 10^{|\hat{j}|-1} + 1_{(\hat{j}=0) \vee (\hat{j}=3)} \cdot (100 - V_k)$$

In parallel with this a vector of fusions is updated, such that at every time point, the next element in the fusion vector is set to 1 if a fusion took place, and 0 else.

## Parallel computing

Many repetitions of time consuming stochastic simulations had to be performed, and many sets of ODEs were solved for each choice of parameters. Therefore, simulations were carried out on the computer grid on The Bioinformatics Center, University of Copenhagen. This allowed running repetitions in parallel with MATLAB's *Parallel Computing toolbox* using between 5 and 100 cores depending on the simulation job.

## Calculating the postsynaptic response

In order to calculate the eEJC, we needed a vector of the SV fusions at different time points. Both deterministic and stochastic simulations yielded the vectors *time_outcome* and *fuse_outcome*, which is a pair of vectors of the same length but with changing time steps. For the sampling we generated a time vector, *time_sample,* with a fixed time step of 1 μs. From here, the determining of the SV fusion times differ between deterministic and stochastic simulations.

In the deterministic simulations, we simulated a sample of distances, *bins*, as described earlier. Each bin gave rise to a set of ODEs, which could be simulated independently, and the *fuse_outcome* is continuously changing based on the rates. In MATLAB the interpolation for bin $k$ was done as follows:

$$fuse\_interp_k = interp1(time\_outcome, fuse\_outcome, time\_sample)$$

*fuse_interp$_k$* contained the cumulative fused SVs over time in a single bin sampled at the time points of the vector *time_sample*. These were summed to find the total number of fused SVs:

$$fuse\_interp = \sum_{k=1}^{n_{bins}} fuse\_interp_k$$

Therefore the SVs fused per time step were be the difference between neighboring values in the *fuse_interp* vector:

$$fusion\_vec = [0, diff(fuse\_interp)]$$

This vector was the basis for the computation of the eEJC.

In the stochastic simulations, the *fuse_outcome* vector contains discrete SV fusions at certain time points. We therefore sample the SV fusions by assigning them to the nearest time points on the *time_sample* vector. That is, each fusion time was rounded to the nearest microsecond, thereby giving rise to the *fusion_vec*, which in the stochastic case contained whole numbers of SV fusions at different time points.

In both deterministic and stochastic simulations the mEJC was generated as a vector, *mEJC_vec,* with the same time step as the *time_sample* and *fuse_vec*. This allows us to calculate the eEJC with MATLAB's convolve function, *conv*, such that

$$eEJC = conv(fuse\_vec, mEJC\_vec)$$

where *fusion_vec* is a vector with the same time step, each element being the number of SV fusions at each time point.

## Analysis of simulated eEJCs

The $eEJC_1$ amplitude was determined as the minimum current of the eEJC within the time interval $(0,10)$ ms. Similar to the analysis of experimental eEJC data, we fitted an exponential function to the decay for estimation of the base value for the second response (see *Figure 2—figure supplement 1A*). The $eEJC_2$ amplitude was the difference between the second local minimum and the fitted exponential function extrapolated to the time point of the second local minimum (as described for the analysis of electrophysiology experiments).

## Fitting routine

Because deterministic simulations cannot predict PPR values (due to Jensen's inequality, see above), but stochastic simulations cannot be fitted to data, we first ran deterministic simulations comparing the simulated first and second absolute eEJC amplitudes to the experimental amplitudes (not the PPR, see Materials and methods). Afterwards we ran stochastic simulations with the optimised parameters in order to compare PPRs and variances to experimental results. To determine the optimal parameters for the deterministic simulations at the five experimental extracellular $Ca^{2+}$ concentrations, the models were fitted to the two peak amplitudes, $eEJC_1$ and $eEJC_2$, by minimizing the following cost value:

$$cost\big(eEJC_{1,sim}, eEJC_{2,sim}\big) = \sum_{k=1}^{5}\left(\frac{\big(eEJC_{1,sim,k} - eEJC_{1,exp,k}\big)^2}{eEJC_{1,exp,k}} + \frac{\big(eEJC_{2,sim,k} - eEJC_{2,exp,k}\big)^2}{eEJC_{2,exp,k}}\right)$$

where we sum over the five different experimental $Ca^{2+}$ concentrations. Note that in deterministic simulations, $eEJC_1$ and $eEJC_2$ amplitudes are precise estimates of average amplitudes in stochastic simulations allowing us to do deterministic optimizations.

When fitting the models, we used the inbuilt MATLAB function *fminsearch,* which uses the Nelder-Mead Simplex Search, to minimize the above cost function. The cost calculation in each iteration was a two-step process taking advantage of the fact that the total number of SVs scales the $eEJC_1$ and $eEJC_2$ values in the deterministic simulations. For each choice of parameters the simulation was run with 180 sites (the initial number of sites is arbitrary, but matched the number of distance bins), and the optimal number of sites were determined afterwards. Thus, a given set of parameters gave rise to amplitudes $eEJC_{1,init}$ and $eEJC_{2,init}$ from simulations with 180 sites. After that we determined $c_{sites} \in R^+$ such that $cost\big(c_{sites}eEJC_{1,init}, c_{sites}eEJC_{2,init}\big)$ was minimized. The number of sites in the given iteration was therefore $180 \cdot c_{sites}$ and the cost of that particular iteration was

$$cost\big(eEJC_{1,sim}, eEJC_{2,sim}\big) = cost\big(c_{sites}eEJC_{1,init}, c_{sites}eEJC_{2,init}\big)$$

In this way the optimization algorithm did not have to include $n_{sites}$ in the parameter search algorithm, which reduced the number of iterations significantly.

In the stochastic simulations, the number of SVs was set to $180 \cdot c_{sites}$ rounded to nearest integer.

## Acknowledgements

JRL Kobbersmed was supported by a pregraduate Scholarship in the Medical Sciences from the Independent Research Fund Denmark (application by JB Sørensen) and by the Data Science Laboratory, Faculty of Science, University of Copenhagen. AT Grasskamp was supported by a NeuroCure Ph.D. fellowship funded by the NeuroCure cluster of excellence within the International Graduate Program Medical Neurosciences (Charité Universitätsmedizin Berlin, Germany). AM Walter is a member of the Einstein Center for Neurosciencies Berlin, M Jusyte was supported by an Einstein Center for Neurosciences Ph.D. fellowship. This work was supported by grants from the Deutsche Forschungsgemeinschaft to AM Walter (Emmy Noether program project number 261020751, TRR 186 project number 278001972). We thank Matthijs Verhage and Niels Cornelisse for helpful comments on our manuscript. We would like to thank Victor Matveev for helpful comments on the use of his $Ca^{2+}$ simulator software. We thank Barth van Rossum, Leibniz-Forschungsinstitut für Molekulare Pharmakologie (FMP), for the illustrations in *Figure 9*. We also thank The Bioinformatics Centre, University of Copenhagen, for the use of their servers for heavy computations. Stocks obtained from

the Bloomington *Drosophila* Stock Center (NIH P40OD018537) and from the Kyoto Stock Center were used in this study.

## Additional information

### Funding

| Funder | Grant reference number | Author |
|---|---|---|
| Deutsche Forschungsge-meinschaft | Emmy Noether Programme, Project Number 261020751 | Alexander M Walter |
| Deutsche Forschungsge-meinschaft | Project Number 278001972 - TRR 186 | Alexander M Walter |
| Independent Research Fund Denmark | Pregraduate scholarship (8141-00007B) | Jakob Balslev Sørensen Janus RL Kobbersmed |
| NeuroCure Cluster of Excel-lence | NeuroCure Fellowship | Andreas T Grasskamp |
| Einstein Stiftung Berlin | Einstein Center for Neuroscience | Meida Jusyte Alexander M Walter |
| University of Copenhagen | Data Science Laboratory | Janus RL Kobbersmed |
| Lundbeck Foundation | R277-2018-802 | Jakob Balslev Sørensen |

The funders had no role in study design, data collection and interpretation, or the decision to submit the work for publication.

### Author contributions

Janus RL Kobbersmed, Formal analysis, Investigation, Visualization, programming and simulation; Andreas T Grasskamp, Formal analysis, Investigation, Visualization, participating in programming and simulation, imaging, image analysis; Meida Jusyte, Formal analysis, Investigation, Visualization, electrophysiological recordings; Mathias A Böhme, Investigation, STED microscopy; Susanne Ditlevsen, Jakob Balslev Sørensen, Alexander M Walter, Conceptualization, Supervision

### Author ORCIDs

Janus RL Kobbersmed  https://orcid.org/0000-0003-0313-6205
Andreas T Grasskamp  https://orcid.org/0000-0002-5895-6529
Meida Jusyte  https://orcid.org/0000-0001-9948-871X
Mathias A Böhme  https://orcid.org/0000-0002-0947-9172
Susanne Ditlevsen  http://orcid.org/0000-0002-1998-2783
Jakob Balslev Sørensen  https://orcid.org/0000-0001-5465-3769
Alexander M Walter  https://orcid.org/0000-0001-5646-4750

### Decision letter and Author response

Decision letter https://doi.org/10.7554/eLife.51032.sa1
Author response https://doi.org/10.7554/eLife.51032.sa2

## Additional files

### Supplementary files

• Source code 1. MATLAB scripts used for simulations.

• Transparent reporting form

### Data availability

All data and software codes generated and used during this study are included in the manuscript and supporting files. Source data is included for all figures.

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
