## [Decision Letter]

**Acceptance summary:**

The paper presents a thorough combination of EM and computational modelling to explain the short term plasticity of synaptic release.

**Decision letter after peer review:**

Thank you for submitting your article "Rapid regulation of vesicle priming explains synaptic facilitation despite variable vesicle:Ca^2+^ channel distances" for consideration by *eLife*. Your article has been reviewed by three peer reviewers, one of whom is a member of our Board of Reviewing Editors, and the evaluation has been overseen by Ronald Calabrese as the Senior Editor. The following individuals involved in review of your submission have agreed to reveal their identity: Victor Matveev (Reviewer #1).

The reviewers have discussed the reviews with one another and the Reviewing Editor has drafted this decision to help you prepare a revised submission. While *eLife* aims to prevent long review cycles and substantial additional work, the reviewers have identified quite a number of distinct issues. In subsequent discussions these were all deemed relevant to support the study's conclusion. However, a number of issues can be addressed by providing a deeper discussion of the limitations and assumptions of the study.

Summary:

This work combines experimental recordings with mathematical modeling to investigate mechanisms of Ca^2+^-exocytosis coupling and facilitation (STF) at *Drosophila* NMJ. The distinguishing feature of this study is that vesicle-channel distances were carefully mapped, allowing the authors to investigate the implications of non-uniform channel-vesicle distance on NT release at this NMJ. The main conclusion of this work is that many traditional models cannot account for the STF observed at this NMJ, given the observed distribution of channel-vesicle distances. The reason is that traditional models explain STF through changes in the vesicle release probability, but the latter is very high close to the Ca^2+^ channel, and very low far from the Ca^2+^ channel. Therefore, very few vesicles would feel the effect of small Ca^2+^ accumulation. The authors propose and test an alternative model which explains the observed properties of neurotransmitter release at this terminal. This model posits that vesicle undergo priming prior to becoming fusion-ready. Although similar STF models have been recently considered by others (as cited in this work), the presented model is more comprehensive, implements important stochastic effects, and proposes inhibition of de-priming rather than acceleration of priming as a key target of Ca^2+^ action. An alternative "site activation" model is also presented, and there is some parameter sensitivity analysis included.

The hypothesis presented for STF and Ca-exocytosis coupling is in general interesting, and the model comparisons presented contribute to the understanding of this problem.

Essential revisions:

1.1) Equation 2: The whole Ca^2+^ current calibration is very hard to follow, and is my main concern regarding methods description. In Equation 2, shouldn't K_M_ be raised to power "h"? Also, this expression gives the relationship between fluorescence and intracellular Ca^2+^, or was the fluorescence recorded with extracellular GCaMP? Further, why is K_M_ determined from GCaMP6 calibration? Isn't K_M_ an innate property of the Ca channel conductivity (and possibly Ca-dependent inactivation), having nothing to do with the indicator dye? The entire Ca^2+^ current calibration has to be explained much more clearly, I could not follow it.

1.2) Unless I misunderstood the modeling/methods, the described stochastic simulation process, while technically correct, appears needlessly complicated and computationally expensive. Any Markov Chain with a single absorbing state and deterministic propensities ([Ca] is deterministic here) is described exactly by its master equation ODE, and the first passage time probability density (vesicle release time density) is directly and exactly computed from this ODE system as the transition rate to the final absorbing state. Therefore, Gillespie SSA simulations are not required, and the only Monte Carlo step involves drawing the vesicle release time from this exact probability distribution, without having to simulate/resolve any intermediate reaction times (once vesicle is released, one may have to recompute the FPT density, but that's not expensive). This should not affect the results, but would greatly improve the simulation efficiency, accuracy, and simplify the parameter sensitivity analysis. Since this should not affect the results, I would not request any significant modifications, but this could be somehow reflected in Materials and methods and checked (unless I misunderstand something and the transition rates are in fact stochastic).

1.3) Subsection “Rate equations of the simulated models”: this part of Materials and methods is particularly hard to read, and could be improved just by a simple re-structuring. It is awkward to explain parameters before the actual equations are shown. The rate dependence on [Ca^2+^] in subsection “Rate equations of the simulated models” seems strange: shouldn't there be a power of "n" in both terms in the denominator?

1.4) Discussion paragraph ten: when discussing alternative STF scenarios beyond "pVr-based" models, it could be appropriate to specifically distinguish between models where all vesicles have the same properties, and models with distinct vesicle pools with different properties (such as the super-primed pool models cited in various parts of the manuscript). Of course, this distinction is complicated, since vesicle pool heterogeneity could result from vesicles being in different transient states along the same slow priming processes, but this could also be pointed out… If the authors find it appropriate, they could also mention here the putative "highly Ca^2+^-sensitive pools" at endocrine cells, which I find interesting and potentially relevant (reviewed in Pedersen and Sherman (2009) PNAS 106:7432-7436).

1.5) It would be valuable to briefly comment on the facilitation decay time constant predicted by their model, since facilitation decay provides a powerful additional constraint on the potential model mechanism of STF. Even if experimental data on STF decay is not available in this case, the decay time constant could be presented as a prediction to be verified in future experimental work. The authors could simply quote the relevant priming/unpriming time scale determining STF decay to avoid additional simulation. I note that matching facilitation decay time course puts a strong constraint on the single-sensor and dual-sensor models, and is an extra issue with such models among those mentioned in this work (see e.g. Matveev et al., 2002, cited in this manuscript).

1.6) Discussion paragraph nine: since the site activation model is not described in detail in Results or Discussion, but is one of two presented new models reproducing experimental observations, it would be helpful to describe it slightly more in this part of the manuscript, if not earlier (1-2 more sentences would suffice).

1.7) When the dual-sensor model is first introduced, it would be appropriate to more prominently cite the related work of Sun et al., 2007, (which the authors cite elsewhere). Further, when discussing various STF models and prior work in this direction, I suggest citing the study of Ma et al., 2015, which is one of the most detailed models of STF and involves a fully stochastic approach: https://www.ncbi.nlm.nih.gov/pubmed/25210157.

2.1) While the characterization of the vesicle distribution is well done. It remains possible that the vesicles are not drawn independently from the distribution. In an extreme case, the number of vesicles could be fixed, or they could repulse each other, and yet the same distribution could still be found. I wonder if this can be discussed.

2.2) The role of heterogeneity in release and averaging across experiments (alluded to in Discussion paragraph six) is not clear to me. It would be good to know that the authors have convinced themselves that such heterogeneity does not over-estimate variance or otherwise change the results.

3.1) The EM analysis was conducted in mutant animals. Exact genotypes were not provided, but based on the information in Reddy-Alla et al., 2017 where the data were originally published I think they are working with Unc13 null animals expressing UAS-Unc13A via a Gal4 driver. This data is being used to drive models of normal synaptic function, so it's critical that this experiment is conducted in wild-type animals with normal levels of Unc13.

3.2) The authors use the data presented in Figure 1D to place synaptic vesicles relative to Ca^2+^ channels in their models. It's difficult to discern the number of animals, NMJs and active zones from the numbers provided: "n=19 observations in 10 EM cross sections/cells." This information should be clearly stated. Assuming an observation is a docked vesicle and a cross section is an active zone, the 3D placement of vesicles was derived from the observation of 19 docked synaptic vesicles in single sections of 10 active zones. Given that the central thesis that vesicles are heterogeneously distributed, so few observations cannot confidently capture the biological range.

This also suggests less than two docked vesicles per cross section, which is somewhat lower than docked vesicle/active zone numbers at wild-type *Drosophila* type Ib synapses previously reported by multiple groups, including the authors' prior work. This could be the genotype (see below) and/or the fact that single sections do not accurately capture the full complement of docked vesicles at an active zone. For example, vesicles that appear close to, but not in contact, with the membrane in one slice, may be in clear contact with the membrane in an adjacent plane. 3D EM approaches, which have been done at the *Drosophila* NMJ, would provide much better estimates of synaptic vesicle topology and obviate the need for deriving 3D estimates based on limited information.

3.3) It is well documented that the hundreds of active zones at *Drosophila* NMJs are of different developmental stages with very different morphologies and release properties, so the distribution of synaptic vesicles observed at a handful of active zones can't accurately represent the NMJ as a whole. This will be very challenging to address experimentally, but should at least be considered in their interpretations and addressed in the Discussion.

3.4) The study seems to involve a mixture of analysis of two motorneuron subtypes with different structural and functional properties without considered this in the modeling. Though not always stated, it looks like the EM, STED and GCaMP experiments were conducted at type Ib synapses, while the electrophysiology measured the compound response to both Ib and Is motor inputs. Can they specifically measure and model type Ib?

3.5) I think wild-type animals were investigated in the STED and electrophysiology experiments, but the genotypes are not noted. All genotypes should be clearly labeled in figures. Additionally, since the data is being re-used here, this manuscript should provide all relevant methods rather than referring readers to the earlier publication. The STED results appear to be based on three animals from a single previously published experiment. Have any technical replicates of this experiment been performed to control for experimental variability?

3.6) There are very few references to other *Drosophila* labs working in this area. Multiple labs have conducted relevant work on the distribution of synaptic vesicles, Ca^2+^ channels, and heterogeneous release properties at this synapse that should be cited.

---

## [Author Response]

Essential revisions:1.1) Equation 2: The whole Ca^2+^ current calibration is very hard to follow, and is my main concern regarding methods description. In Equation 2, shouldn't K_M_ be raised to power "h"? Also, this expression gives the relationship between fluorescence and intracellular Ca^2+^, or was the fluorescence recorded with extracellular GCaMP? Further, why is K_M_ determined from GCaMP6 calibration? Isn't K_M_ an innate property of the Ca channel conductivity (and possibly Ca-dependent inactivation), having nothing to do with the indicator dye? The entire Ca^2+^ current calibration has to be explained much more clearly, I could not follow it.

The reviewer is correct that the parameter K_M_ should also be raised to the power of h. We have corrected this. Thank you for pointing this out. The relevant Ca^2+^ concentration is that of the extracellular medium, we also did not state this clearly in the equation.

We are sorry that it was unclear from the text what the reasoning of the experiment was. This is because we introduced an analysis of how Ca^2+^ influx depends on the extracellular Ca^2+^ concentration *before* explaining why this at all would be relevant. We have now restructured the text and swapped the sequence of Figures 2 and 3. This way we first present the electrophysiological recordings (to which the mathematical models are later contrasted) at different extracellular Ca^2+^ concentrations. This then delivers the explanation of why the dependency of Ca^2+^ influx a as a function of extracellular Ca^2+^ concentrations should be studied in Figure 3.

We now also extended the main text to clearly explain the premise of this experiment, which was (as the reviewer points out) to quantify the saturation behavior of the Ca^2+^ current as a function of the extracellular Ca^2+^ concentration. The reviewer is entirely correct that this is an intrinsic property of the Ca^2+^ channels. As such, it would be more direct to conduct an electrophysiology experiment to measure these currents, but this is not possible in this preparation. We therefore used an intracellular Ca^2+^ fluorescence measurement as a proxy. GCaMP6m was expressed within the presynapse and its relative fluorescence change in response to action potential activation (10 AP at 20 Hz) z) was measured. This only allows us to measure relative changes in Ca^2+^ inflow. We now point out more clearly that this experiment cannot derive absolute Ca^2+^ concentration because the system cannot be calibrated in such a way. The fluorescence values detect residual Ca^2+^ after the AP train. The relative change in fluorescence can give an indication of the relative magnitude of Ca^2+^ influx as a function of extracellular Ca^2+^ levels. We indeed see that the fluorescence increase saturates (as expected from the saturation of Ca^2+^ currents) and this is well described by a hill curve. We were most interested in the “IC50/KM” value of the curve, because we used this value to approximate the saturation of the Ca^2+^ current in Equation 4. The second relevant parameter, “h”, describes the cooperativity of the fluorescence increase as a function of extracellular Ca^2+^. Because we reasoned that this should be a property of the GCaMP sensor, rather than the channel (this was described in Barnett et al., 2017) this parameter was fixed to the reported value of h = 2.43. The best fit parameter value of KM in this case was 2.679 mM, which is near identical to the KM value (2.6 mM) previously reported for the Ca^2+^ influx at the mammalian calyx of Held synapse (Schneggenburger and Neher, 1999). Both the IC50 and the Michaelis Menten constant report on the half-saturation point. So as a matter of fact, our results are primarily confirmatory in that the Ca^2+^ channels at the *Drosophila* NMJ show a largely similar saturation behavior.

We also checked whether fixing the parameter “h” had a major effect on the estimation of KM from our experiments. This was not the case, as allowing additional variation of h during the fit resulted only in a marginally different estimate of the parameters (h=1.887 and K_M_=3.054 mM). With the added explanation, we are now confident that the aim and rationale and the methodology is clear.

1.2) Unless I misunderstood the modeling/methods, the described stochastic simulation process, while technically correct, appears needlessly complicated and computationally expensive. Any Markov Chain with a single absorbing state and deterministic propensities ([Ca] is deterministic here) is described exactly by its master equation ODE, and the first passage time probability density (vesicle release time density) is directly and exactly computed from this ODE system as the transition rate to the final absorbing state. Therefore, Gillespie SSA simulations are not required, and the only Monte Carlo step involves drawing the vesicle release time from this exact probability distribution, without having to simulate/resolve any intermediate reaction times (once vesicle is released, one may have to recompute the FPT density, but that's not expensive). This should not affect the results, but would greatly improve the simulation efficiency, accuracy, and simplify the parameter sensitivity analysis. Since this should not affect the results, I would not request any significant modifications, but this could be somehow reflected in Materials and methods and checked (unless I misunderstand something and the transition rates are in fact stochastic).

In principle this is correct, however, to calculate the solution of the master equation ODE can only be done numerically, due to the time varying Ca^2+^-dependent transition rates. Then the FPT density has to be calculated from this numerical distribution. Moreover, this distribution would have to be calculated repeatedly, for each randomly drawn SV:channel distance, and after each fusion, where first a random time for replenishment has to be drawn, then the numerical solution of the ODE has to be found to calculate the FPT density. We doubt that this will improve simulation efficiency or accuracy.

1.3) Subsection “Rate equations of the simulated models”: this part of Materials and methods is particularly hard to read, and could be improved just by a simple re-structuring. It is awkward to explain parameters before the actual equations are shown. The rate dependence on [Ca^2+^] in subsection “Rate equations of the simulated models” seems strange: shouldn't there be a power of "n" in both terms in the denominator?

We agree on this point and the section has been restructured. The equations were moved up before the parameter explanations. We added: “Parameters are described below. For designation and value of parameters, see Table 2 and 3.”

The reviewer is of course right about the power of ‘n’ on the terms in the denominator. We double-checked that this was correctly implemented in the scripts used for simulations and the typo in the manuscript has been corrected.

1.4) Discussion paragraph ten: when discussing alternative STF scenarios beyond "pVr-based" models, it could be appropriate to specifically distinguish between models where all vesicles have the same properties, and models with distinct vesicle pools with different properties (such as the super-primed pool models cited in various parts of the manuscript). Of course, this distinction is complicated, since vesicle pool heterogeneity could result from vesicles being in different transient states along the same slow priming processes, but this could also be pointed out… If the authors find it appropriate, they could also mention here the putative "highly Ca^2+^-sensitive pools" at endocrine cells, which I find interesting and potentially relevant (reviewed in Pedersen and Sherman (2009) PNAS 106:7432-7436).

This is a good point, and we have inserted such a discussion now, distinguishing between vesicle pools with different intrinsic properties, and vesicles with the same intrinsic properties, but different distances to the Ca^2+^-channels. Regarding the highly Ca^2+^-sensitive pool, we are well aware of the work and have cited it repeatedly. However, it is based on work in chromaffin cells and beta cells, which is now 15 years old. We would therefore prefer not to go into it here, as we fear it might sidetrack the text.

1.5) It would be valuable to briefly comment on the facilitation decay time constant predicted by their model, since facilitation decay provides a powerful additional constraint on the potential model mechanism of STF. Even if experimental data on STF decay is not available in this case, the decay time constant could be presented as a prediction to be verified in future experimental work. The authors could simply quote the relevant priming/unpriming time scale determining STF decay to avoid additional simulation. I note that matching facilitation decay time course puts a strong constraint on the single-sensor and dual-sensor models, and is an extra issue with such models among those mentioned in this work (see e.g. Matveev et al., 2002, cited in this manuscript).

This is a very good suggestion. We now mention this reference in this context in the text. We have also followed the reviewer’s suggestion to predict the time-course of facilitation of this model which is now included as a supplementary item (Figure 7—figure supplement 3). These predictions can then later be tested experimentally by varying the ISI in PPR experiments.

1.6) Discussion paragraph nine: since the site activation model is not described in detail in Results or Discussion, but is one of two presented new models reproducing experimental observations, it would be helpful to describe it slightly more in this part of the manuscript, if not earlier (1-2 more sentences would suffice).

Thank you for pointing this out. We have now added a more elaborate description of the site activation model in the Results section at the end of the paragraph “Rapidly regulating the number of participating release sites accounts for eEJC_1_ amplitudes, STF, temporal transmission profiles and variances”.

1.7) When the dual-sensor model is first introduced, it would be appropriate to more prominently cite the related work of Sun et al., 2007, (which the authors cite elsewhere). Further, when discussing various STF models and prior work in this direction, I suggest citing the study of Ma et al., 2015, which is one of the most detailed models of STF and involves a fully stochastic approach: https://www.ncbi.nlm.nih.gov/pubmed/25210157.

Thanks for pointing this out. We have added statements and references accordingly.

2.1) While the characterization of the vesicle distribution is well done. It remains possible that the vesicles are not drawn independently from the distribution. In an extreme case, the number of vesicles could be fixed, or they could repulse each other, and yet the same distribution could still be found. I wonder if this can be discussed.

There are two issues here: one is what the vesicle distribution looks like, the other is how this distribution comes about. The first question we can solve (and we believe we have solved it), the second question is open. However, for the work we present the question how the distribution comes about (by independent random localization of each vesicle, or affected by repulsive interaction between vesicles) does not affect the initial positioning of vesicles – the random drawing we employ will ensure a distribution of vesicles, which closely reflects the identified distribution, even if the mechanism that created this distribution does not involve random positioning. The only place where the mechanism creating the distribution becomes a question is during recovery, i.e. when some vesicles have fused as a result of the first AP and new vesicles need to prime. It is possible that vesicle positions are again drawn independently from the distribution, alternatively vesicles could selectively prime into the now empty positions (‘slots’). We have chosen the latter possibility for several reasons: (1) It is the only mechanism that would have any chance of producing facilitation in pVr-based models. The vesicles fusing during the first AP are localized much closer to the Ca^2+^-channels than the average vesicle, and priming into newly selected random positions will therefore shift the distribution of vesicles away from the Ca^2+^-channels, making facilitation an impossibility. As we conclude that facilitation using pVr-based models is not likely, we had to make sure we were not stacking the deck against this possibility; (2) If vesicles prime into new slots, the distribution of vesicles after full recovery would have shifted after the action potential, and then would have to shift back, which is a complicated mechanism for which there is no evidence; (3) We believe it is generally accepted that in synapses vesicles prime into pre-existing ‘slots’ on the plasma membrane, created by priming proteins (including Unc13s). And we have shown that these proteins are extremely stable (in FRAP experiments we previously established a recovery time constant of ~6 hours, Reddy-Alla et al., 2017). Therefore, it is effectively the slots we have to draw according the distribution, and not the vesicle positions, although the position of vesicles of course follows from the distribution of slots.

2.2) The role of heterogeneity in release and averaging across experiments (alluded to in Discussion paragraph six) is not clear to me. It would be good to know that the authors have convinced themselves that such heterogeneity does not over-estimate variance or otherwise change the results.

We have now made the text clearer. In this kind of experiment there are two sources of variance: there is the variance between responses within the same cell (within-cell variance), and there is variance between different cells (between-cell variance). An advantage of having a stochastic model is that we can compare it not only to the mean response, but also to the variance. The relevant variance is the within-cell variance, as it is measured experimentally in variance-mean experiments, where a single cell is stimulated repetitively. Therefore, we have displayed the data as the mean (between cells) plus-minus the mean within-cell variance in the figures where simulation output is compared to experiments (Figures 4, 6, 7). In fact, this results in a very good match between model and data, both for the means and the variances, for the depriming model (e.g. Figure 7E, G). This does not over-estimate variance. Indeed, a single model should be able to reproduce the within-cell variance, but it cannot encompass the between-cell variance, since we would need one model per cell.

We also tried to clarify the statement referred to by replacing the second part of the statement with the following sentence: “Note that such parabola agrees with the mean-variance relationship in a binomial distribution. However these simulations show that the assumption of heterogeneous release probability can also lead to the experimentally observed parabolic variance-mean relationship.”

3.1) The EM analysis was conducted in mutant animals. Exact genotypes were not provided, but based on the information in Reddy-Alla et al., 2017 where the data were originally published I think they are working with Unc13 null animals expressing UAS-Unc13A via a Gal4 driver. This data is being used to drive models of normal synaptic function, so it's critical that this experiment is conducted in wild-type animals with normal levels of Unc13.

We are sorry that this was not clearly stated and now provide all details in a section of the Materials and methods where all genotypes are listed by figures so this information can easily be found. The reviewer is correct that this is a previously published dataset and the flies are indeed Unc13 null animals expressing the Unc13A isoform under an elav-Gal4 neuronal driver. The rationale here was to isolate the contribution of Unc13A, because the second *Drosophila* isoform Unc13B only marginally (<5%) contributes to synaptic transmission. As Unc13B localizes at larger distances from the Ca^2+^ channels and likely functions at docking at those distant positions (which is hard to map in the EM images) we thought that this would be more adequate. Also, the second approach to map the docking sites using STED microscopy only maps the distribution of the Un13A isoform (it is an isoform specific antibody targeting the N-terminal region of the protein). The reviewer is correct that the re-expression of the protein using the elav-Gal4 driver could potentially lead to a higher expression than the endogenous promoter. However, we would like to point out, that our focus here is to study the distribution of docking sites rather than their number (see also point 3.2). Therefore, the higher levels of Unc13A would not affect the conclusions of our study as long as the protein shows a similar distribution. Nonetheless, the reviewer makes a valid point that a corresponding analysis should also be performed in wildtype animals. We therefore now added this analysis to our study which is also based on a previously published dataset (measurements of docked SVs in wildtype *Drosophila* larvae from Böhme et al., 2016.) The result and the fitted Rayleigh distribution are shown in Figure 1—figure supplement 1A.

When comparing the relevant (2D) integrated Rayleigh distribution from which vesicles are placed in our simulation in both cases (wildtype vs. Unc13A rescue, Figure 2—figure supplement 2B) it becomes evident that the distributions are near identical. This rules out that the altered Unc13A levels severely affect the docked vesicle distribution.

This alleviates the reviewer’s concern that the genotype influenced our analysis of docked SV positions. Please note that we did not re-run the optimization of all models with the second distribution. This would be pointless, because the distributions are essentially identical and any effect on the best fit parameters would be marginal and it would not be justified to spend weeks of calculations on a computer grid for this purpose.

3.2) The authors use the data presented in Figure 1D to place synaptic vesicles relative to Ca^2+^ channels in their models. It's difficult to discern the number of animals, NMJs and active zones from the numbers provided: "n=19 observations in 10 EM cross sections/cells." This information should be clearly stated. Assuming an observation is a docked vesicle and a cross section is an active zone, the 3D placement of vesicles was derived from the observation of 19 docked synaptic vesicles in single sections of 10 active zones. Given that the central thesis that vesicles are heterogeneously distributed, so few observations cannot confidently capture the biological range.

We are sorry that this information was not provided here. We indeed only referred to the original publication. Of course it makes sense to immediately make the information available and we have now added this to the figure legend. We also provide all source data and provide the relevant information in the transparent reporting form. The analysis for the Unc13A re-expression dataset is based on 10 AZs from at least two animals. The analysis of the EM data from wildtype flies (Figure 1—figure supplement 1A) is based on 11 AZs from 5 animals.

We disagree with the reviewer regarding the second point (that the number of observations is too low to conclude on the distribution of distances) and the argument against this is provided below. In this study, we find that the distribution of distances is broad and this can be safely concluded from the number of observations we have. We used two different methods to test how reliable the finding of a broad vesicle distance distribution from the EM data is: by using confidence intervals from the fitting of the Rayleigh distribution presented in the paper (based on the likelihood function) and by bootstrapping from the data set at hand (without assuming anything about the underlying distribution).

Likelihood approach:

In the paper, we fitted a Rayleigh distribution to the data set using the maximum likelihood function, which yielded σ=76.52 nm as the best fit parameter.

From the likelihood function we can estimate 95% confidence intervals of σ and the means and SDs of the found distributions (Author response table 1).

**Author response table 1. resptable1:** 

	95% CI, lower bound	Best fit	95% CI, higher bound
Parameter value (σ)	62.53 nm	76.52 nm	98.61 nm
Mean of Rayleigh	78.37 nm	95.90 nm	123.59 nm
SD of Rayleigh	40.97 nm	50.13 nm	64.60 nm
Mean of integrated Rayl	99.76 nm	122.08 nm	157.36 nm
SD of integrated Rayl	42.12 nm	51.52 nm	66.40 nm

Using the three different σ-values from above (best fit and the lower and upper bound of the 95% CI), the integrated Rayleigh distribution from which we draw SV distances in our simulations looks as shown in Author response image 1:

Since the STD varies with the choice of parameter in a monotone way and since all these three distributions are relatively broad, any choice of parameter in the confidence interval will yield a broad distribution. Thus, when fitting a Rayleigh distribution to the data, we get a broad distribution of distances, also when taking the uncertainty of the fit into account.

Bootstrap analysis from the data

To estimate the mean and SD of the underlying distribution of SV distances without making any assumptions on the nature of this distribution we bootstrapped from the data set at hand (Figure 1B, n=19). We drew 19 observations with replacement from the data set 10000 times and estimated the mean and the variance in each set of observations. The estimated mean is the average sampled mean, and the estimated SD is the square root of the average sampled variance. The confidence intervals are the 2.5% and 97.5% quantiles. In the case of the SD, the confidence interval is the square root of the quantiles of the variances.

The histogram in Author response image 2 shows the sampled mean with 95% confidence interval marked:

In this way, we use the knowledge we have about the data (our data set of 19 observations) to estimate the underlying mean and SD without making any assumptions about the distribution. Author response table 2 compares the mean and SD from bootstrapping to the mean and SD found by fitting the Rayleigh distribution (i.e. not the integrated distribution, since we only estimate the mean and SD of the distribution underlying our 1D sampling):

**Author response image 2. respfig2:** The histogram in Author response image 2 shows the sampled mean with 95 % confidence interval marked.

**Author response table 2. resptable2:** 

	95% CI, lower bound	Estimated mean/SD of distribution	95% CI, higher bound
Mean of Rayleigh	78.37 nm (CI from likelihood)	95.90 nm	123.59 nm (CI from likelihood)
SD of Rayleigh	40.97 nm (CI from likelihood)	50.13 nm	64.60 nm (CI from likelihood)
Mean from bootstrap	72.74 nm (CI from quantiles)	94.85 nm	118.32 nm (CI from quantiles)
SD from bootstrap	37.37 nm (CI from quantiles)	51.52 nm	65.21 nm (CI from quantiles)

Thus, both methods of estimation indicate that the underlying distribution is in fact broad. The first method provides a confidence interval for the distribution fitted, whereas the bootstrap estimates mean and SD (and confidence intervals) from the number of observations in the dataset without assuming the underlying distribution to be a Rayleigh distribution (i.e. it is entirely non-parametric). Since both methods yield a broad distribution (large SD) and since the integration of the distribution will lead to a larger SD, both methods points toward the SV distance distribution being broad.

As the reviewer pointed out, the broadness of the distribution is at the core of the problem investigated in this paper, and we think this analysis justifies our assumptions on this matter. Besides, the distribution is confirmed in an independent EM dataset (Figure 1—figure supplement 1A) and in independent STED analyses (Figure 1E,F, Figure 1—figure supplement 1C,D).

This also suggests less than two docked vesicles per cross section, which is somewhat lower than docked vesicle/active zone numbers at wild-type *Drosophila* type Ib synapses previously reported by multiple groups, including the authors' prior work. This could be the genotype (see below) and/or the fact that single sections do not accurately capture the full complement of docked vesicles at an active zone. For example, vesicles that appear close to, but not in contact, with the membrane in one slice, may be in clear contact with the membrane in an adjacent plane. 3D EM approaches, which have been done at the *Drosophila* NMJ, would provide much better estimates of synaptic vesicle topology and obviate the need for deriving 3D estimates based on limited information.

We realize that the image previously shown in Figure 1A was maybe not representative and included a new one. We would like to point out that we are not really interested in the absolute number of docked vesicles in this study (in fact, this is a free parameter in the model). The focus was primarily on their distribution. Nevertheless, the number of docked SVs does in fact align with previous estimates by us and others: the Unc13A re-expression dataset reports ~2 docked SV per active zone, the wildtype dataset ~3. Estimates by other groups are in this range, see e.g. Bruckner et al., 2016 (http://doi.org/10.1083/jcb.201601098) where the O’Conner-Giles lab estimated ~2.5 docked SVs/AZ in thin EM sections and 2-3 docked SVs/AZ from EM tomography images. One should note, however, that the definition of docking often depends on docking definitions and on the resolution in the EM images, and as such is not always directly comparable. However, as pointed out above, since this number is a free parameter for our modeling; it is not a concern that some SVs docked at this AZ may be missed (that would be docked in other sections).

Using EM-tomography could be another method to address the docked SV distribution, but as such an approach relies heavily on computational approaches for image reconstruction, we feel that this is less direct. EM-tomography might make it possible to identify more docked vesicles in a given presynapse; however, this will not change the distribution of docking sites. Besides, we show a complementary approach by our STED analysis, which we feel is stronger because it uses another technique but delivers a similar estimate of SV docking sites with molecular information.

3.3) It is well documented that the hundreds of active zones at *Drosophila* NMJs are of different developmental stages with very different morphologies and release properties, so the distribution of synaptic vesicles observed at a handful of active zones can't accurately represent the NMJ as a whole. This will be very challenging to address experimentally, but should at least be considered in their interpretations and addressed in the Discussion.

We fully agree that the heterogeneity across AZs is a very important factor and indeed inspired us to this project. Therefore it seems that this point may partly rely on a misunderstanding, as it is exactly the main premise of our study to quantitatively investigate the consequence of heterogeneous SV placemen. Note that in our model we make no distinction between whether the variability is present within single AZs, or is a variability between AZs, therefore the model captures both. Thus, our approach is actually suitable to address exactly the type of heterogeneity between AZs that the reviewer is pointing to. We feel our approach is much more accurate than previous ones by taking into account this heterogeneity and by furthermore doing so in quantitative manner (e.g. captured by a single parameter of the Rayleigh distribution) and showing its consequence for function. It is correct that apart from to this there are additional sources of heterogeneity and we now extended on the discussion of these sources (e.g. Ib/Is type boutons, heterogeneous priming).

3.4) The study seems to involve a mixture of analysis of two motorneuron subtypes with different structural and functional properties without considered this in the modeling. Though not always stated, it looks like the EM, STED and GCaMP experiments were conducted at type Ib synapses, while the electrophysiology measured the compound response to both Ib and Is motor inputs. Can they specifically measure and model type Ib?

We agree that many things can be learned here regarding additional sources of heterogeneity (see also point 3.3). We followed the reviewer’s suggestion and added this to in the Discussion. Indeed, both connection types likely contribute to the electrophysiological measurements (though it is currently not exactly known to which extent). Also, the STED analysis focusses on AZs within Ib type boutons and it could be interesting to systematically investigate the different bouton types using STED microscopy in the future. The GCaMP data were obtained from type Ib synapses, but measurements were merely used to capture the saturation behavior of Ca^2+^ currents as a function of extracellular Ca^2+^ concentrations (see also point 1.1). We found exactly the same relationship as described for the murine calyx of Held synapse. This makes sense, because this should be a fundamental property of the Ca^2+^ channels. We therefore feel that it is less likely this property would be markedly different between the two input types as it was so similar across species. Nevertheless, this could be investigated in the future. EM analysis favors Ib boutons (AZs are typically used from large boutons with prominent postsynaptic SSR). However, it cannot be excluded that sometimes Is connections are also sampled (because they are difficult to distinguish at this level).

We agree with the reviewer that mapping possible differences in function, composition and SV distribution might be interesting. We also agree that this will be very challenging, likely requiring the genetic manipulation of one vs. the other. This again could perturb the system in unexpected ways in addition to desired effects, so specificity could be a problem. Depending on what comparative analyses reveal, this could lead to a refinement of the model to the two bouton types (and how each of these additionally contributes to heterogeneity). However, this is beyond the scope of the current study. As it stands, the current model is a compound model describing overall synaptic transmission to this postsynaptic connection, which we want to publish as such.

3.5) I think wild-type animals were investigated in the STED and electrophysiology experiments, but the genotypes are not noted. All genotypes should be clearly labeled in figures. Additionally, since the data is being re-used here, this manuscript should provide all relevant methods rather than referring readers to the earlier publication. The STED results appear to be based on three animals from a single previously published experiment. Have any technical replicates of this experiment been performed to control for experimental variability?

We now provide all details on genotypes in a dedicated section of the Materials and methods where genotypes are listed by figures, making this information immediately accessible. Electrophysiological recordings were performed in animals expressing an Ok6-Gal4 driver: *Ok6-GAL4/+ (Ok6-Gal4/II* crossed to *w1118*). The reason that Gal4 driver was introduced into the wildtype strain was such that these animals could function as a control group for a motor neuron specific knockdown of Syt-7 (which was suggested to function as a facilitation sensor). However, in the end we did not include these data (where the Ok6-Gal4 is combined with a UAS-Syt7-RNAi).

Technically, there could be a potential effect of the Ok6-Gal4 driver on synaptic transmission, its Ca^2+^ dependence and/or its short-term plasticity. To alleviate this concern, we performed electrophysiological recordings in *Drosophila* larva expressing Ok6-Gal4 and compared these to wildtype animals (+/+: *w1118*) in parallel. These data revealed no difference between the two groups and are now shown in the new Figure 2—figure supplement 3.

We found no difference in the amplitude of AP-evoked eEJCs at 0.75 mM [Ca^2+^]_ext_: wildtype eEJC amplitude: -35.13 ± 24.94 nA (n = 6 cells from 3 animals, mean ± SD) and Ok6-Gal4/+ eEJC amplitude: -32.36 ± 9 nA, (n = 8 cells from 4 animals, mean ± SD), P = 0.77; Student´s t – test. Similar responses were also seen at 1.5 mM extracellular Ca^2+^, arguing for a similar Ca^2+^ sensitivity: wildtype eEJC amplitude -98.24 ± 10.73 nA (mean ± SD, n = 5 cells from 3 animals) and Ok6-Gal4/+ eEJC amplitudes: -93.22 ± 12.16 nA (mean ± SD, n = 5 cells from 3 animals); P = 0.55; Student´s t – test.

Finally, both groups showed indistinguishable PPR values at both concentrations (0.75 mM Ca^2+^: wildtype: 1.02 ± 0.51 (n = 6 cells from 3 animals, mean ± SD) and Ok6-Gal4/+: 1.15 ± 0.26 (n = 8 cells from 4 animals, mean ± SD); P = 0.55; Student´s t – test.; 1.5 mM Ca^2+^: wildtype: 0.65 ± 0.12 (mean ± SD, n = 5 cells from 3 animals) and Ok6-Gal4/+: 0.58 ± 0.08 (mean ± SD, n = 5 cells from 3 animals); P = 0.39; Student´s t – test):

Together, these observations rule out that the presence of the Ok6-Gal4 has a severe effect on the magnitude- or Ca^2+^ dependence of AP-evoked synaptic transmission, or that it markedly alters the synapse’ short-term plasticity characteristics in a way that could affect the conclusions of our study.

We furthermore included a description of the full procedure of larval preparation, fixation, immunohistochemistry, image acquisition and the averaging of the STED images in the Materials and methods section. And we also now include another biological replicate of the approach (completely different animals, different staining, imaged separately) based on 583 individual AZ images from 3 wildtype larva for comparison. This analysis essentially reproduces the distribution Unc13A distribution found before. These data are depicted a new figure, Figure 1—figure supplement 1.

3.6) There are very few references to other *Drosophila* labs working in this area. Multiple labs have conducted relevant work on the distribution of synaptic vesicles, Ca^2+^ channels, and heterogeneous release properties at this synapse that should be cited.

We apologize that this was missed and now provide additional references to these important works.